# TestExplora: Can LLMs write tests to find potential problems existing in repository?

## Abstract

As Large Language Models (LLMs) are increasingly applied to automate software development, their use for automatic test case generation has become a key area of research. However, existing benchmarks for evaluating LLMs fundamentally simplify the real-world testing challenge. They typically constrain the problem to either (1) reproducing known bugs at the repository level, or (2) generating tests for isolated code units, such as individual functions, detached from their broader project context. Both approaches fail to assess the crucial capability of LLMs for proactive, exploratory testing in projects defined by complex, cross-file dependencies. To address this critical gap, we introduce TestExplora, the first systematic benchmark designed to evaluate the proactive defect discovery capabilities of LLMs at the repository level. Constructed from real-world pull requests, TestExplora challenges models to find bugs without any prior knowledge of bug manifestations. Our comprehensive evaluation, conducted in both black-box and white-box settings, reveals a stark capability gap. Even state-of-the-art models exhibit critically low success rates (e.g., GPT-5-mini: 12.79%, o3-mini: 5.23%), and access to the full source code (white-box) yields only marginal improvement. Further Analysis reveals that existing models struggle mainly with assertion mismatches and misconfigured mocks. TestExplora thus establishes a principled foundation for advancing research towards the grand challenge of autonomous, repository-level defect discovery.

## 1 Introduction

As Large Language Models (LLMs)(OpenAI, 2024; Gemini-Team, 2025; Qwen-Team, 2025) are increasingly applied to automate software development, their use for automatic test case generation has become a key area of research. Ideally, such automation should not only reproduce known bugs to prevent regressions but, more importantly, perform proactive exploratory testing to discover unknown, hidden defects before code is deployed. This proactive discovery capability is crucial for determining whether test automation can evolve from an auxiliary tool into a core pillar of software quality assurance.

However, an examination of current benchmarks reveals that their design has largely overlooked the systematic measurement of this core capability, as they fundamentally simplify real-world testing along two different axes. First, one line of work, while operating at the repository level, constrains the task to error reproduction. ~~For example,~~ Recent studies have concentrated on leveraging LLMs to generate tests strictly from explicit issue reports (Hasan et al., 2025; Nashid et al., 2025) or specific commit changes (Pradel, 2025; Liu et al., 2025). Reflecting this focus, SWT-Bench (Mündler et al., 2024) and TDD-Bench-Verified (Ahmed et al., 2024b) require a model to produce a unit test that replicates a pre-defined bug scenario. This framing assesses a model's ability to confirm a known issue, not its capability to discover new ones. Second, another prominent line of work focuses on test generation for isolated code units, such as the self-contained functions and individual files found in corpora like (Li & Yuan, 2024; Wang et al., 2025a; Zhang et al., 2024; Jain et al., 2025). It ignores the complex web of dependencies—across modules, APIs, and data schemas—where the most critical and subtle real-world bugs often reside. Consequently, a critical gap remains: there is no systematic evaluation of an LLM's ability for proactive defect discovery in realistic repository contexts—a capability that lies at the heart of modern software engineering, aiming to shift the

development focus from reactive, "firefighting" debugging to proactive quality building, thereby fundamentally improving development efficiency and software robustness.

Table 1: Comparison of different open-source benchmarks with ours. Abbreviations: Cov = Coverage, Acc = Accuracy, MS = Mutation Score, ComR = Commit Relevance, FP = Fail-to-Pass Rate, CC = Change Coverage, Wb = White-box, Bb = Black-box.

| Benchmark | #Tasks | Task Level | Repositories | TDD | Testing Scenario | Eval. Metrics |
|---|---|---|---|---|---|---|
| TestEval | 216 | Function | — | ✗ | Wb | Cov |
| UnLeakedTestbench | 3909 | Function | — | ✗ | Wb | Pass@k, MS |
| TestBench | 108 | Class | — | ✗ | Wb | Cov, Acc, MS |
| TestGenEval | 1210 | File | 11 | ✗ | Wb | Pass@k, Cov, MS |
| ProjectTests | 295 | Project | 60 | ✗ | Wb | ComR, Corr |
| SWTBench | 1900 | Project | 12 | ✔ | Wb | FP, CC |
| TDDBench | 449 | Project | 12 | ✔ | Wb | FP, CC |
| **TestExplora** | **2389** | **Project** | **482** | ✔ | **Bb & Wb** | **Cov, Acc, FP, CC** |

To address these limitations, we present TestExplora, a benchmark that evaluates LLMs as issue-finding testers in realistic repository contexts. Unlike existing benchmarks that focus on problem reproduction, TestExplora deliberately conceals all manifestations of defects—including diffs, commit messages, and issue reports—forcing models to discover bugs through behavioral testing rather than pattern matching. This design mirrors real-world exploratory testing, where engineers must proactively identify risks without prior knowledge of specific bugs. The evaluation harness of TestExplora executes generated tests on both buggy and fixed versions of the code. A test is considered valid if it fails before the fix and passes after, aligning with the Fail-to-Pass principle of defect validation. Beyond Fail-to-Pass rates, TestExplora also measures change-focused coverage, providing a more comprehensive assessment of test quality than prior benchmarks. In summary, this work makes three contributions:

- TestExplora benchmark: a curated suite of real-world pull requests evaluates LLMs as issue-finding testers in realistic repository context. TestExplora contains 2,389 test generation tasks from 1,552 pull requests in 482 repositories.

- Comprehensive Evaluation: empirical results showing that even the strongest model achieves only a maximum Fail-to-Pass rate of 12.79%, posing a critical capability gap. Further Analysis reveals that writing a high-quality test requires not only strong programming ability but also the capability to identify the key aspects that need to be tested. And existing models struggle mainly with assertion mismatches and misconfigured mocks.

- Scalable benchmark collection framework: We propose a scalable data collection framework that consists of the core steps of repo filtering with validated pull requests, automated build on a virtual machine, and Fail-to-Pass validation. It can be extended on demand and continuously produce new data for the field of test generation.

## 2 RELATED WORK

**Benchmarks for Software Engineering.** In recent years, several repository-level software engineering benchmarks have been introduced, emphasizing realistic repositories, long contexts, and multi-file dependencies. SWE-bench (Jimenez et al., 2023) provides 2,294 issues and fixes from 12 Python projects and has become a widely used benchmark. Its extensions (Li et al., 2025; Chowdhury et al., 2024; Yang et al., 2024; Zhang et al., 2025; Zan et al., 2025) further expand scale, difficulty, and dynamism to mitigate contamination and static overfitting. Other efforts include USEbench, which integrates multiple SWE tasks (Applis et al., 2025); DevBench, which evaluates multiple stages of the development lifecycle (Li et al., 2024; Tan et al., 2024); and Other Benchmarks (Le Hai et al., 2024; Zhao et al., 2025), which target completion, dependency handling, and real-world bug fixing. Collectively, these resources form a rich ecosystem for repository-level evaluation, yet they predominantly emphasize bug fixing, feature implementation, and code completion, and rely on high-quality human-written tests as the supervision signal for determining task success.

**Benchmarks for Test Generation.** Motivated by this reliance, subsequent work has explored leveraging LLMs for test generation (Wang et al., 2025d; Hasan et al., 2025; Wang et al., 2025c). As the centrality of testing has become more widely recognized, an increasing number of benchmarks have been introduced to evaluate test generation at different granularities. At the function level, TestEval and UnLeakedTestbench (Wang et al., 2024; Huang et al., 2025) focus on line, branch, and path coverage while addressing contamination and realism. At the class level, TestBench (Zhang et al., 2024) samples 108 Java classes and proposes a five-dimensional evaluation of syntax, executability, coverage, and defect detection. At the project level, CLOVER (Xu et al., 2025) examines long-context generation, ProjectTest (Wang et al., 2025b) targets medium-scale projects across three languages, and TestGenEval (Jain et al., 2025) builds on 68k human-written tests to assess writing, completion, and improvement. SWT-Bench (Mündler et al., 2024) and TDD-Bench Verified (Ahmed et al., 2024a) tie tests to real issues and fixes. Most of these efforts emphasize confirmatory testing, whereas TestExplora conceals diffs, commit messages, and implementation details, requiring models to hypothesize risk areas and design tests for latent defects, with effectiveness measured by Fail-to-Pass outcomes and change-focused coverage.

## 3 TESTEXPLORA

TestExplora is a benchmark designed to measure large language models' ability in exploratory software testing—specifically, their capacity to proactively discover defects rather than merely reproduce known errors.The subsequent sections present our methodology across three components: Benchmark Acquisition, Model Inputs, and Evaluation Metrics.

### 3.1 BENCHMARK ACQUISITION

All instances of TestExplora are derived from existing GitHub repositories to ensure data quality. During the construction of TestExplora, carefully maintained repositories were selected. As illustrated in Figure 1, the Benchmark Acquisition process primarily comprises the following four main steps:

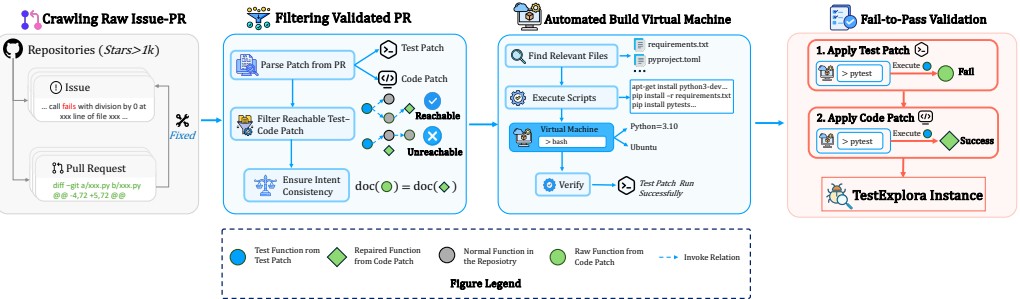

Figure 1: The data acquisition process of TestExplora.

**Step1.Crawling Raw Issue-PR** Constructing a reliable exploratory testing benchmark requires high-quality, well-maintained codebases where defects can be meaningfully discovered and validated. We select only repositories $\mathcal{R}$ with substantial community adoption ($>1,000$ stars), yielding 12,227 preliminary pull requests with patches $\mathcal{P}$ that provide genuine opportunities for proactive defect detection.

**Step2.Filtering Validated PR** For each patch $\mathcal{P}$, we can decompose it into a test patch $\mathcal{P}_t$ and a code patch $\mathcal{P}_c$, where $\mathcal{P}_c = \mathcal{P}/\mathcal{P}_t$. By employing the parser $parser(\cdot)$, we can identify the function with the number of $n$ modifications involved in each patch, such that $parser(\mathcal{P}_k) = \{f_{1,k}, \ldots, f_{n,k}\}, k \in \{c, t\}$. Before applying the Fail-to-Pass filtering, we first analyze the invoke graph of the repository. We designate the functions directly invoked by $parser(\mathcal{P}_c)$ as ~~entry functions~~ entry interface $\mathcal{E}$, which are then used for subsequent test generation across different pipelines. We retain only PRs where modified functions $f_c \in parser(\mathcal{P}_c)$ are reachable by test functions $f_t \in parser(\mathcal{P}_t)$ through explicit call paths. This ensures that our benchmark captures

realistic testing scenarios where generated tests can meaningfully exercise the modified code and potentially discover defects through proper invocation chains. This step differs from previous benchmarks (Mündler et al., 2024; Ahmed et al., 2024b), which only applied Fail-to-Pass filtering without verifying whether the Fail-to-Pass outcome was indeed caused by the code patch $\mathcal{P}_c$. In addition, we filter out PRs in which $\mathcal{P}_c$ involves doctoring modifications, in order to ensure that the function's intention does not undergo substantial changes before and after the pr. We also exclude PRs where, after preprocessing, $parser(\mathcal{P}_c)$ contains doctoring with "TODO" annotations or implementations including "pass," thereby ensuring that the code after the pr does not contain potential placeholders for future updates.

**Step3. Automated Build Virtual Machine** Constructing Docker environments for large-scale repositories is challenging; therefore, we adopted GitHub Actions Runner Images[1] to establish a unified virtual machine with test scripts, enabling Fail-to-Pass testing in a manner analogous to GitHub Actions Runners. Action Run script is listed in the Appendix B. During the setup of each repository for testing, the process mainly consists of the following three steps: **a) Install system dependencies**: This step primarily prepares the system-level compilation toolchain and external library dependencies. **b) Look for and merge all requirements files**: It automatically detects and merges requirements within the project, ensuring that no requirements are missed or duplicated during subsequent installation. **c) Environment Setup**: This step selects the logic for dependency installation based on the project structure in order to configure the testing environment. The script attempts to identify a pyproject.toml project and installs dependencies using Poetry or PDM. If the project is not a pyproject project, it installs dependencies using pip together with requirements/setup.py and common utility packages. This approach facilitates flexible setup of the testing environment, and such flexibility enables us to expand existing datasets at any time without manual setup.

**Step4. Fail-to-Pass** Similar to SWE-bench, we conducted tests on $\mathcal{P}_t$ both before and after applying the code patch $\mathcal{P}_c$. Ultimately, our dataset comprises 1,552 pull requests and 2,389 test generation tasks across 482 repositories. The Appendix J provides information on the categories of the repositories and related details.

When generating tests for $\mathcal{E}$, it is essential to clarify the intention of $\mathcal{E}$ to enable effective exploratory testing that can uncover potential defects. The most straightforward approach is to leverage existing docstrings of $\mathcal{E}$ as documentation, enabling LLMs to infer function intent and generate comprehensive tests. However, we observe that adequate documentation is rarely available for ~~entry functions~~ entry interface in real-world repositories. To address this documentation scarcity, we use a high-performing agent—DocAgent (Yang et al., 2025)—to generate the corresponding documentation for $\mathcal{E}$. This automated annotation approach greatly enhances the scalability of TestExplora. We list a log from the execution process of DocAgent in the Appendix C.

Table 2: The statistical information of TestExplora. **Categories** denotes the numbers of repository categories of repositories. In **Test Invokes**, **Entries per Test** counts functions invoked by a test case, while $\mathcal{P}_c$ **Depth** is the invocation distance between the test case and the modified code patch.

| Corpus Overview | | | | |
|---|---|---|---|---|
| **Repositories** | **PRs** | **Tests** | **Categories** | **Avg. Stars** |
| 482 | 1552 | 2389 | 21 | 5010.77 |
| **PR Instance Statistics** | | | | |
| **Area** | **Indicator** | **Max** | **Mean** | **Median** |
| Test Patch | # Tests Edited | 23 | 1.60 | 1 |
| Test Invokes | Entries per Test | 154 | 8.22 | 3 |
| | $\mathcal{P}_c$ Depth | 12 | 1.76 | 1 |
| ~~Entry Function~~ Entry Interface | # Dependencies | 150 | 3.57 | 1 |
| | Lines of Code | 1378 | 29.07 | 13 |
| Code Patch | # Lines Edited | 1297 | 22.92 | 11 |
| | # Functions Edited | 29 | 1.42 | 1 |
| | # Files Edited | 28 | 1.83 | 1 |

Following the above steps, TestExplora consists of 1,552 pull requests from 482 repositories, along with 2,389 test cases as generation tasks. Table 2 presents the information of TestExplora.

## 3.2 MODEL INPUTS

Effective exploratory test generation requires strategic information provisioning to balance realism with model capability assessment. TestExplora provides three categories of input information:

---

[1]https://github.com/actions/runner-images

- **Documentation**: Documentation, produced through the two-stage generation process described in Section 3.1, is employed to clarify what the test entry points are intended to accomplish, what their inputs and outputs are, and what potential errors may arise.
- **Test Entry Points**: As mentioned in Section 3.1, test entry points $\mathcal{E}$ are codes that are directly invoked by the test cases $parser(\mathcal{P}_t)$.
- **Dependencies**: Dependencies are the direct dependencies of the test entry points and are used to further clarify the intention of the test entry points for the LLMs.

To more comprehensively simulate testing under different conditions, we defined two testing scenarios: white-box and black-box. As illustrated in the lower-right corner of Figure 2, these scenarios differ in terms of the input information provided. Specifically, Code Imp. indicates whether the concrete implementation of the test entry points is provided, while Dep. denotes whether the dependencies of the test entry points are included. In all two settings, the documentation (Doc.) of the test entry points is consistently provided.

### 3.3 EVALUATION METRICS

Before introducing the evaluation metrics, we first formalize the test generation task: $\mathcal{T}_{n,t}^* = \Theta(\mathcal{E}_{n,t}|s), s \in \{$white-box, black-box$\}$ is a set of tests generated by the model $\Theta(\cdot|\cdot)$ given the test entry points $\mathcal{E}_{n,t}$. $\mathcal{E}_{n,t}$ denotes the entry points of a specific test $t$ in the ground truth tests $\mathcal{T} = parser(\mathcal{P}_t)$ from the $n^{th}$ data snippet's test patch $\mathcal{P}_{n,t}$. To evaluate the quality of $\mathcal{T}_n^* = \bigcup_{t \in \mathcal{T}} \mathcal{T}_{n,t}^*$, we employed the following four metrics:

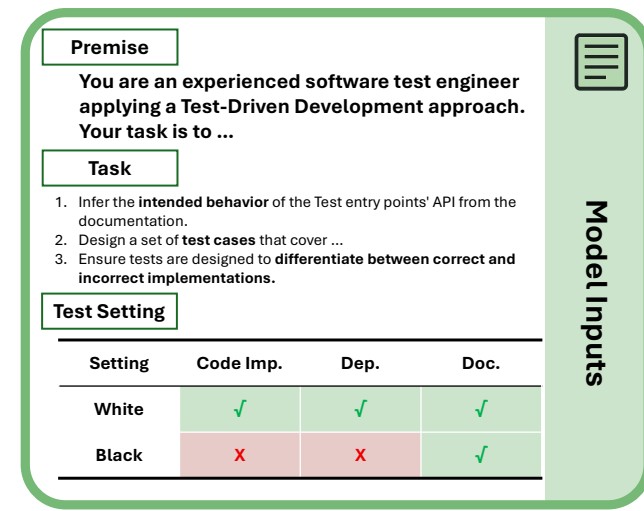

Figure 2: Illustration of model inputs. According to the differences in input information, the tests are mainly divided into two scenarios: white-box testing and black-box testing.

**Head Pass Rate (HP)** : We measured the pass rate of the tests generated by the model on the head commit after the pull request, which reflects the accuracy with which the generated tests adhere to the intended functionality. The Head Pass Rate is a test-level metrics:

$$HP = \frac{|\bigcup_{n=1}^{N}\{t \in \mathcal{T}_n^* | pass(t, \mathcal{R}_{n,base} \times \mathcal{P}_{n,c})|\}}{\sum_{n=1}^{N}|\mathcal{T}_n^*|}, \tag{1}$$

where $pass(\cdot, \cdot)$ serves as an indicative function $\mathbb{I}$, which denotes whether the test $t$ passes on the head repository $\mathcal{R}_{n,base} \times \mathcal{P}_{n,c}$, where the issue from the base commit repository $\mathcal{R}_{n,base}$ is fixed with $\mathcal{P}_{n,c}$.

**Fail-to-Pass Rate (F2P)** : Similar to SWT-bench (Mündler et al., 2024), we compute the proportion of PRs for which at least one generated test exhibits a Fail-to-Pass transition. This metric reflects the effectiveness of the generated tests, indicating the extent to which the model can accurately identify potential errors:

$$F2P = \frac{\sum_{n=1}^{N}\mathbb{I}(|f2p(n)| \geq 1)}{N}$$
$$f2p(n) = \{t \in \mathcal{T}_{pr}^* | pass(t, \mathcal{R}_{n,base} \times \mathcal{P}_{n,c}) \& fail(t, \mathcal{R}_{n,base})\}\}, \tag{2}$$

where $f2p(\cdot)$ is a pr level function, which finds the test that passes on the head repository and fails on the base repository.

**Entry Coverage (EC)** : Entry Coverage measures the line coverage of the generated tests with respect to the test entry points, reflecting the extent to which the generated tests comprehensively capture the intention of the test entry points:

$$EC = \frac{1}{N} \sum_{n=1}^{N} \frac{|cover(\mathcal{T}_n^*, \mathcal{E}_n)|}{|line(\mathcal{E}_n)|}, \tag{3}$$

where $cover(\cdot, \cdot)$ denotes the set of lines from ~~entry functions~~ entry interface $\mathcal{E}_n$ covered by the tests. And $line(\cdot)$ returns the lines of $\mathcal{E}_n$.

**Change-focused Coverage (CFG)** : When applying $\mathcal{P}_{n,c}$, certain lines in $\mathcal{R}_n$ are modified, denoted as $\Delta(\mathcal{R}_\backslash)$:

$$CFG = \frac{1}{N} \sum_{n=1}^{N} \frac{|cover(\mathcal{T}_n^*, parser(\mathcal{P}_{c,n}))|}{|\Delta(\mathcal{R}_n)|}. \tag{4}$$

The aforementioned metrics collectively capture the model's fidelity to the code's intended functionality, its precision in identifying issues, and the comprehensiveness of the generated tests.

Table 3: Performance comparison of different models on TestExplora. The best results are in **bold**, and the second-best are underlined.

| Type | Model | TestExplora | | | | | TestExplora-Lite | | | | |
|------|-------|------|------|------|------|------|------|------|------|------|------|
| | | $HP$ | $F2P$ | $EC$ | $CFG$ | $Num.$ | $HP$ | $F2P$ | $EC$ | $CFG$ | $Num.$ |
| Black Box | QC-30B-A3B | 63.98 | 2.05 | **52.98** | 42.21 | 19.48 | 51.28 | 1.16 | 47.01 | 42.95 | 13.29 |
| | o3-mini | 68.27 | 3.93 | 51.46 | 41.67 | 13.86 | 54.66 | 5.23 | 52.57 | 44.40 | 8.11 |
| | o4-mini | 66.18 | 2.59 | 51.18 | 43.24 | 14.19 | 54.16 | 2.33 | 58.59 | 46.48 | 10.8 |
| | Gemini-2.5-pro | 62.04 | 2.06 | 38.40 | 41.45 | 14.26 | **58.60** | 1.74 | 49.14 | 45.60 | 12.72 |
| | GPT-4o | 62.58 | 1.47 | 46.98 | 42.14 | 12.31 | 47.66 | 1.74 | 55.01 | 43.92 | 7.60 |
| | GPT-5-mini | **73.28** | **5.17** | 52.42 | **43.41** | 11.50 | 58.00 | 7.56 | **60.63** | **46.70** | 7.82 |
| White Box | QC-30B-A3B | 68.80 | 1.39 | 53.99 | 43.01 | 19.21 | 54.35 | 1.74 | 57.58 | 44.98 | 12.70 |
| | o3-mini | 76.54 | 4.19 | 52.49 | 42.16 | 12.68 | 67.01 | 6.98 | 59.56 | 43.40 | 7.72 |
| | o4-mini | 72.33 | 5.92 | **67.30** | **47.06** | 10.48 | 72.33 | 5.92 | 67.30 | **47.06** | 10.48 |
| | Gemini-2.5-pro | 74.51 | 2.59 | 45.88 | 43.05 | 16.23 | 73.00 | 4.65 | 62.97 | 44.71 | 12.99 |
| | GPT-4o | 67.91 | 1.79 | 46.00 | 41.64 | 11.22 | 52.81 | 1.16 | 53.83 | 44.71 | 7.12 |
| | GPT-5-mini | **84.80** | **7.54** | 56.33 | 43.17 | 11.55 | **77.68** | **12.79** | 67.76 | 45.66 | 7.69 |

## 4 EVALUATION

To demonstrate the effectiveness of our approach and understand the capabilities of different models in automated test generation, in this section, we evaluate the performance of six mainstream models on TestExplora. We conduct a comprehensive evaluation using four key metrics: Head Pass Rate, Fail-to-Pass Rate, Entry Coverage, and Change-focused Coverage. And the number of testcases $Num.$ is also listed. Our experiments span six mainstream language models across 12,227 real-world pull requests. To ensure efficient evaluation, we constructed a subset named TestExplora-Lite by filtering based on the quality of human-written docstrings in the repository. This subset contains 330 PRs and 517 samples in total.

### 4.1 EXPERIMENTAL DESIGN

**Models** To comprehensively evaluate the problem detection capability of existing LLMs, we select six mainstream models. Among open-source code models, we select the representative Qwen3-Coder-30B-A3B (Qwen-Team, 2025). For general-purpose LLMs, GPT-4o (OpenAI, 2024) and GPT-5-mini[2] are selected. For reasoning models, TestExplora evaluates o3-mini, o4-mini[3], and Gemini-2.5-pro (Gemini-Team, 2025).

---

[2]https://openai.com/index/introducing-gpt-5/

[3]https://openai.com/index/o3-o4-mini-system-card/

**Context Template**  To comprehensively evaluate the capability of LLMs in problem detection, we adopt different input formats to simulate various testing scenarios. Specifically, white-box and black-box are selected as two main testing scenarios. In the white-box scenario, models are allowed to access the complete codes along with their related dependencies. In contrast, in the black-box scenario, the model is only provided with the corresponding test entry points and the associated documentation. The detailed templates are provided in Appendix D.

## 4.2 EVALUATION RESULTS

**Performance comparison among the models**  As shown in Table 3, GPT-5-mini achieves the best performance. Its testing capability is stronger in both black-box and white-box scenarios. Specifically, $HP$ is an evaluation metric at the test-case level, which reflects the fundamental ability of the model to write tests. ~~As shown in Table 3, GPT-5-mini achieves the best performance. Its testing capability is stronger in both black-box and white-box scenarios.~~ We observe that Gemini-2.5-pro tends to generate significantly more test cases than other models. While generating a larger number of tests may increase the chance of discovering defects, it also introduces a higher likelihood of producing invalid or incorrect assertions. Since the $F2P$ is calculated as the proportion of generated tests that successfully capture a bug, a large volume of low-quality tests dilutes the metric. In other words, the abundance of tests produced by Gemini-2.5-pro does not necessarily translate into better defect detection; instead, the presence of many ineffective or erroneous tests lowers its average $F2P$ compared to models that generate fewer but higher-quality tests. As for the EC and CFG metrics, GPT-5-mini generates the fewest tests yet achieves superior scores. This indicates that GPT-5-mini excels at capturing the potential branches of the ~~entry function~~ entry interface as well as locating possible errors. Meanwhile, o4-mini achieves the second-best performance on these two metrics.

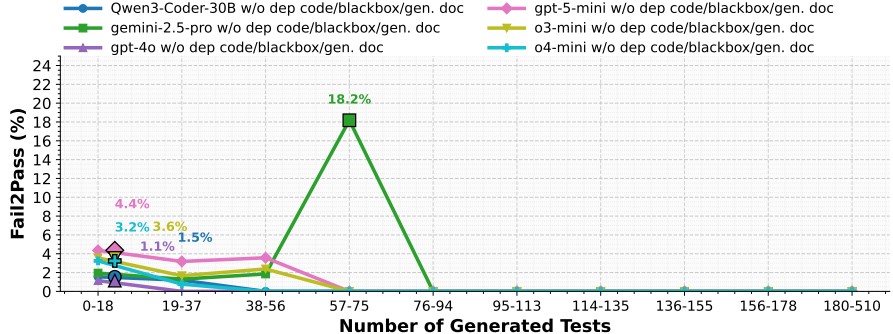

Figure 3: The impact of the number of generated test cases on performance. The best performance of each model is highlighted.

**Performance variation with respect to the number of generated tests**  In the previous section, we find that although Gemini-2.5-pro performs well at the single-test generation level, its tendency to generate a larger number of tests leads to lower performance on the pull-request–level $F2P$ metric. Figure 3 illustrates the correlation between the number of generated tests and model performance. As shown in the Figure 3, when the number of tests generated by a model gradually increases, the corresponding F2P decreases. When the number of generated tests exceeds 104, no model is able to successfully produce correct tests. The experimental results indicate that **generating more tests does not necessarily lead to better performance**. However, generating too few tests also poses potential risks of hacking. For example, in Table 3, GPT-4o produces relatively few tests, but its coverage metric is comparatively low, indicating that it may not generate sufficiently comprehensive tests. ~~TThese~~ These results suggest that, at the repository level, **an effective test suite depends not only on a model's ability to generate tests, but also on its precision in targeting critical functionality. In other words, better performance is achieved by producing fewer but more meaningful tests**.

**Performance variation with respect to the timeline**  Figure 4 shows how the performance of models in test generation varies over time. We find that existing models perform better on data

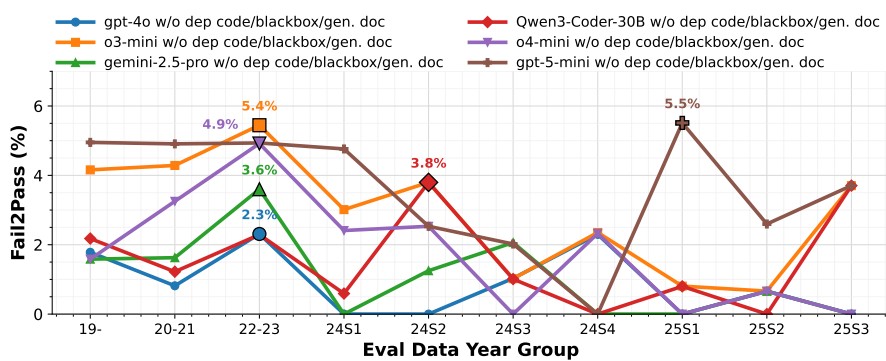

Figure 4: Fail-to-Pass success rates across eval year buckets for six code-generation models without dependency code access, highlighting each model's peak performance season.

prior to 2023 than on data after 2023. Since SWE-Bench (Jimenez et al., 2024) covers pull requests from before 2023, existing models have been trained on repositories related to SWE-Bench, which leads to their relatively stronger performance on pre-2023 data. This indicates that existing datasets (Mündler et al., 2024; Ahmed et al., 2024a) based on SWE-Bench may suffer from data leakage risks, while also highlighting the importance of our scalability framework. Additionally, for GPT-4o, its performance remains relatively poor, which may be because the selected checkpoint (2024-05-13) is not specifically trained on SWE-type datasets.

**Performance variation with respect to the repository categories** We also analyze the impact of repository type on model performance. As shown in Figure 8, the models perform best in the Scientific/Engineering domain, where all models achieve an F2P score above 5%. In contrast, performance is the worst in the Security domain, with both o4-mini and Qwen3-Coder scoring 0.

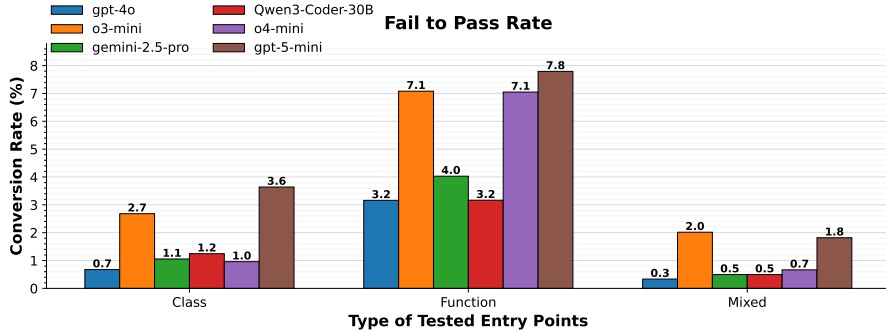

Figure 5: Fail-to-Pass success rates across test entry points type. Class indicates that the test entry points consist only of classes, Function indicates that they consist only of functions, and Mixed denotes that the test entry points are a combination of both.

**Performance variation with respect to the types of entry points** In Figure 5, all models perform best when the test entry points involve only function-type code, while they perform worst in mixed tests involving both functions and classes. This finding naturally divides TestExplora into three levels of difficulty: Easy, Medium, and Hard.

**Performance variation with different types of dependences** In Table 3, we observe that for certain models—such as GPT-4o—their White-Box performance in the Lite version is inferior to their Black-Box performance. To investigate this issue, we hypothesize that GPT-4o may not effectively leverage contextual information. Therefore, in Figure 10, we compute the performance variations of different models under different types of dependencies. In our setting, we categorize dependencies

into two types: invocation and inheritance. GPT-4o shows an F2P rate of 2.3 in the black-box setting, but increases to 3.9 when only invocation dependencies are provided. In contrast, GPT-5-mini achieves an F2P rate as high as 12.8 when all types of dependencies are supplied.

**Performance of Agents**    From the previous section, our experimental results indicate that different models exhibit preferences for different types of dependencies. Therefore, we introduce the Agent baseline. Specifically, we allow the SWE-Agent and Trae-Agent to freely explore the repository to generate tests that uncover potential issues. From Table 4, we can observe that, compared to providing all dependencies directly, the Agent makes more efficient use of the context. This suggests that, for the tasks represented by TestExplora, the Agent constitutes a promising research direction.

Table 4: Performance comparison with different context methods. he best results are in **bold**.

| Model | Context | Metrics | | | | |
|---|---|---|---|---|---|---|
| | | $HP$ | $F2P$ | $EC$ | $CFG$ | $Num.$ |
| o4-mini | White Box | 72.33 | 5.92 | 67.30 | 47.06 | 10.48 |
| | SWEAgent | 82.03 | 9.42 | **69.63** | 47.81 | 13.79 |
| | TraeAgent | **86.55** | **12.16** | 62.01 | **47.95** | 12.76 |
| gpt-5-mini | White Box | 77.68 | 12.79 | **67.76** | 45.66 | 7.69 |
| | SWEAgent | **93.81** | **17.27** | 65.45 | **47.43** | 23.09 |

We also analyze the tool-invocation frequency of different agents. We find that for SWE-agents based on GPT-5-mini and o4-mini, the tool used most frequently at every stage is `str_replace_editor view`. Figure 6 and Figure 11 indicate that the agents are consistently exploring the repository throughout the process. Moreover, the top 15 most frequently used tools are also predominantly focused on repository exploration and comprehension.

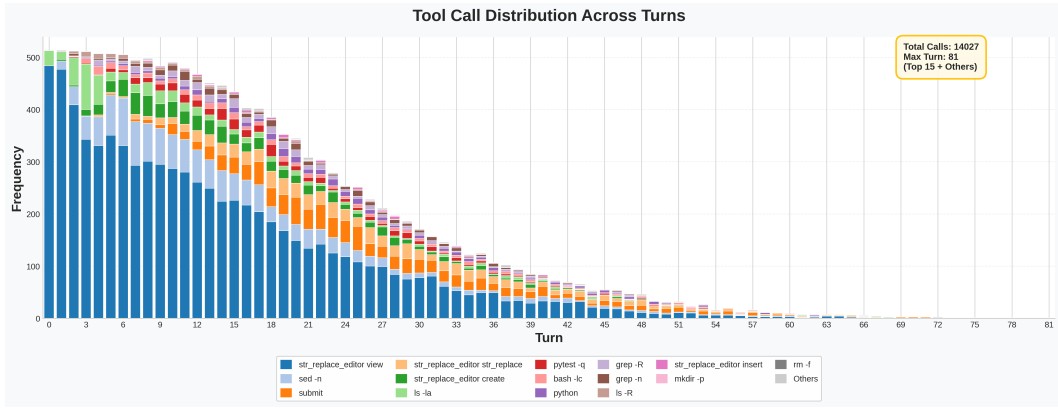

Figure 6:  The frequency with which SWE-agent w/ GPT-5-mini invokes tools in each iteration. Specifically, we analyze all trajectories contained in TestExplora-Lite.

## 4.3 ABLATION ON DOCUMENTATION

In TestExplora, we adopt DocAgent (Yang et al., 2025) as an additional source of information for test generation. To verify the effectiveness of synthetic documentation, we compare the performance differences between synthetic documentation and human-written documentation in TestExplora-Lite.

From Table 5, it can be observed that the documentation generated by DocAgent provides stronger informational gains compared to human-written documentation. Specifically, the documentation generated by DocAgent does not lead to significant changes in the average number of tests generated by the models. Instead, it helps improve performance across other metrics. Since DocAgent observes only the information from the head repository during exploration—without accessing diff

Table 5: Performance change from human-written documentation to DocAgent-generated documentation. The table reports the information gain of DocAgent-generated documentation compared with human-written documentation.

| Type | Model | TestExplora-Lite Change | | | | |
|---|---|---|---|---|---|---|
| | | $HP$ | $F2P$ | $EC$ | $CFG$ | $Num.$ |
| Black Box | QC-30B-A3B | +13.42 | +3.33 | +0.39 | +0.19 | -0.16 |
| | o3-mini | +16.78 | +3.33 | +6.20 | +0.92 | +0.01 |
| | o4-mini | +6.55 | +1.82 | +3.28 | +0.17 | +0.27 |
| | Gemini-2.5-pro | +10.7 | +0.61 | +8.53 | +0.81 | +0.15 |
| | GPT-4o | +4.39 | +2.12 | +6.54 | +0.33 | +0.70 |
| | GPT-5-mini | +10.12 | +6.06 | +10.52 | +1.83 | +0.13 |

patches or error messages—there is no risk of potential information leakage. The ablation study further demonstrates the scalability of TestExplora.

## 5 FURTHER ANALYSIS

We also conduct an analysis of the errors produced by models. From Figure 7, we observe that all models are more prone to Assertion Mismatch errors and Misconfigured Mocks. In other words, existing models fail to accurately capture the behavior and output of the function under test, both in black-box and white-box scenarios. We also conduct a case study in Appendix F.

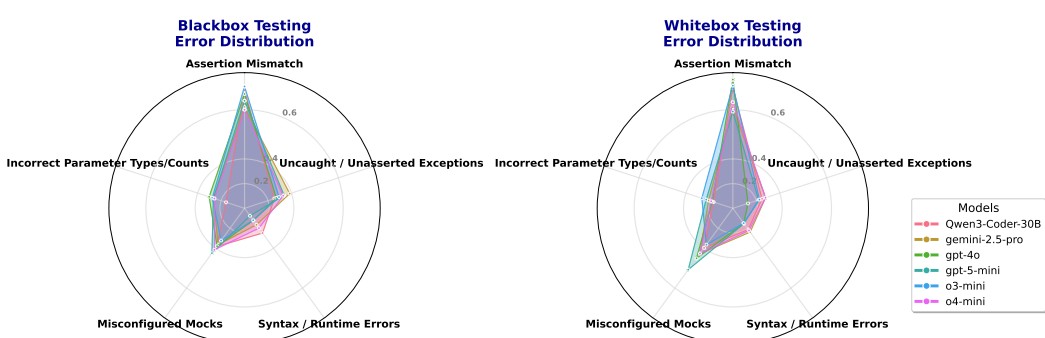

Figure 7: The generated test error distribution. The left subfigure shows the distribution of error types of the model in the black-box scenario, while the right subfigure shows the distribution of error types in the white-box scenario.

We also conduct an ablation study on the dependency forms of test entry points. In our setting, we categorize dependencies into two types: invocation and inheritance. As shown in Figure 10, invocation dependencies exert the most substantial impact, which is consistent with the original design principle of TestExplora—constructing the dataset based on invocation-driven parsing relationships.

## 6 CONCLUSION

We present TestExplora, an extensible new benchmark for evaluating LLMs in exploratory software testing. TestExplora conceals bug information and challenges models to proactively uncover defects in realistic repository-level contexts. Our evaluation reveals a critical capability gap: even the strongest models achieve less than 13% Fail-to-Pass success. These findings highlight that effective test generation requires not only coding ability but also reasoning about risk-prone behaviors.

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

## A  THE USE OF LARGE LANGUAGE MODEL

In this paper, we employ a large language model (LLM) for proofreading and icon creation.

## B  ACTION RUN SCRIPT

**Action Run Script**

```
name: Tests
on:
  push

jobs:
  tests:
    runs-on: ubuntu-latest

    steps:
    - name: checkout
      uses: actions/checkout@v4

    - name: Detect Python
      run: |
        which python3
        python3 --version

    - name: Install system dependencies
      run: |
        sudo apt-get update
        sudo apt-get install -y python3-dev librados2
            librados-dev libpq-dev build-essential

    - name: Look for and merge all requirements files
      id: find_reqs
      shell: bash
      run: |
        echo "===> Looking for requirements*.txt or *
            requirements.txt files..."
        find . -maxdepth 1 -type f \( -iname "requirements*.
            txt" -o -iname "*requirements.txt" \) >
            all_package.txt

        if [[ -s all_package.txt ]]; then
          echo "Found requirements files:"
          cat all_package.txt
          echo "HAS_REQUIREMENTS=true" >> $GITHUB_ENV

          echo "===> Merging and deduplicating requirements
              ..."
          touch merged-requirements.txt
          while read -r file; do
            grep -vE '^\s*#' "$file" | grep -vE '^\s*$' >>
                merged-requirements.txt || true
          done < all_package.txt

          sort merged-requirements.txt | uniq > requirements-
              all.txt
```

```
            echo "===> Combined requirements:"
            cat requirements-all.txt
          else
            echo "No requirements files found."
            echo "HAS_REQUIREMENTS=false" >> $GITHUB_ENV
          fi

          python3 -m pip install --upgrade pip

    - name: Install Python via pyproject
      run: |
        if [ -f "pyproject.toml" ]; then
            echo "===> pyproject.toml detected"
            echo "USE_POETRY=true" >> $GITHUB_ENV

            echo "===> install poetry"
            pip install poetry

            if [ -f "poetry.lock" ]; then
                echo "===> Checked poetry.lock and Using
                    Poetry"
                poetry install || echo "Poetry install dev
                    failed or timed out"
            elif [ -f "pdm.lock" ]; then
                echo "===> Checked pdm.lock and Using PDM"
                echo "USE_PDM=true" >> $GITHUB_ENV
                pip install pdm
                pdm install --with test || echo "Test group
                    not found or failed"
                pdm install --with dev || echo "Dev group not
                     found or failed"
            else
                if grep -q '^\[tool\.poetry\]' pyproject.toml
                     || grep -q '^\[project\]' pyproject.toml
                    ; then
                     echo "===> This pyproject.toml is a
                        project"
                     echo "===> No lock file detected,
                        installing via pip (PEP 517)..."
                     echo "===> install poetry pytest-json-
                        report"
                     poetry install --no-interaction --with
                        dev || echo "Dev group not found or
                        failed"
                     poetry install --no-interaction --with
                        test || echo "Test group not found or
                         failed"
                else
                     echo "===> This pyproject.toml is NOT a
                        project"
                     echo "USE_POETRY=false" >> $GITHUB_ENV
                fi
            fi
        else
            echo "===> pyproject.toml not detected"
            echo "USE_POETRY=false" >> $GITHUB_ENV
        fi
```

```
    - name: Install others if not poetry
      run: |
        if [ "$USE_POETRY" = "false" ]; then
          echo "===> This is not a pyproject.toml project,
              ready to install others"
          pip install --upgrade pip setuptools wheel
              packaging

          if [ -f "requirements-all.txt" ]; then
            echo "Using requirements*.txt..."
            pip install --prefer-binary -r requirements-all.
                txt
          fi

          if [ -f "setup.py" ]; then
            echo "===> Installing from setup.py..."
            pip install -e .[tests]
          fi

          pip install numpy pytest pytest-json-report pytest-
              cases IPython mock pygsheets oauth2client
              pyyaml lxml django
          pip install pytest-black pytest-pylint pytest-
              django python-dotenv pytest-mock django
              responses

          if ! pytest --collect-only; then
              echo "Tests fail due to numpy incompatibility,
                  downgrading..."
              pip install "numpy<2.0"
          fi
        fi

    - name: Run tests and generate JSON report
      run: |
        if [ "$USE_PDM" = "true" ]; then
            echo "===> Run in PDM..."
            pdm add --dev coverage
            pdm fix
            pdm run coverage run -m pytest {test_files}
            pdm run coverage json -o coverage_data.json
            echo "===> Run in PDM end..."
        elif [ "$USE_POETRY" = "true" ]; then
            echo "===> Run in poetry..."
            poetry add coverage
            poetry run coverage run -m pytest {test_files}
            poetry run coverage json -o coverage_data.json
            echo "===> Run in poetry end..."
        else
            echo "===> Run in pytest..."
            pip install coverage
            coverage run -m pytest {test_files}
            coverage json -o coverage_data.json
        fi
```

## C  DocAgent Execution Process

DocAgent consists of four modules: Searcher, Reader, Writer, and Verifier. In this section, we illustrate the generated log with an example from PyCQA/isort (isort.wrap.line).

### C.1  Searcher and Reader

The Searcher is responsible for retrieving the relevant dependencies of the target function within the repository, as well as performing internet searches. In TestExplora, for efficiency, we disable internet search. The Reader primarily analyzes the target code to be generated along with the retrieved context. The Searcher first retrieves the relevant dependencies based on the control flow graph parsed from the repository:

---

**Searcher Iteration 1**

```
The code function "line" handles wrapping an input string
    according to a configurable line length and multiple
    wrapping modes. It examines the content for special
    markers (like comments or import-related keywords) and
    splits it into parts using regular expressions. It then
    conditionally calls a helper function (_wrap_line) to
    further format parts of the line, and it treats special
    cases (such as "# NOQA" comments) differently based on
    configuration settings. The function relies on a
    configuration object (Config) that provides several
    attributes (like line_length, multi_line_output,
    use_parentheses, and others) and on a Modes object or
    enum (with values like NOQA, VERTICAL_HANGING_INDENT, etc
    .) that indicate the wrap mode.

To generate a high-quality docstring, additional internal
    context is needed. In particular, it would help to know:
1. Details about the Config class including its attributes,
    their types, and what each one signifies (for example,
    the significance of DEFAULT_CONFIG).
2. Information about the Modes enumeration (or similar)
    including what each mode means and how it affects the
    wrapping.
3. The purpose and behavior of the helper function _wrap_line
    , e.g. its parameters, its expected output, and how it
    integrates into the overall formatting.
4. Optionally, understanding where the "line" function is
    used might help clarify its expected input or edge-case
    handling, though understanding the internal calls is
    likely sufficient here.

<INFO_NEED>true </INFO_NEED>

<REQUEST>
    <INTERNAL>
        <CALLS>
            <CLASS>Config, Modes</CLASS>
            <FUNCTION>_wrap_line </FUNCTION>
            <METHOD></METHOD>
        </CALLS>
        <CALL_BY>true </CALL_BY>
    </INTERNAL>
```

---

```
        <RETRIEVAL>
            <QUERY></QUERY>
        </RETRIEVAL>
    </REQUEST>
```

Having obtained sufficient information, the Searcher terminates the retrieval process:

**Searcher Iteration 2**

```
The provided code component is a function that takes a string
    as input and returns a wrapped version of the string
    based on a maximum line length defined in its
    configuration. It includes logic for handling comments (e
    .g. "# NOQA"), various delimiters (like "import ", "
    cimport ", ".", and "as ") and even different wrapping
    modes (such as handling vertical hanging indents or grid-
    grouped modes). It also makes use of a helper function (
    _wrap_line) and a formatter function determined by the
    configuration, but overall the transformation logic is
    sufficiently clear from the code.

There is enough context in terms of the configuration
    parameters used by the function and the operations
    performed on the content. The current internal details
    allow us to understand the expected behavior and cover
    exceptional cases (like when wrap mode is NOQA).

<INFO_NEED>false </INFO_NEED>

<REQUEST>
    <INTERNAL>
        <CALLS>
            <CLASS></CLASS>
            <FUNCTION></FUNCTION>
            <METHOD></METHOD>
        </CALLS>
        <CALL_BY>false </CALL_BY>
    </INTERNAL>
    <RETRIEVAL>
        <QUERY></QUERY>
    </RETRIEVAL>
</REQUEST>
```

## C.2   WRITER

The Writer is responsible for generating the documentation based on the available information.

**Writer Iteration 1**

```
<DOCSTRING>
Summary:
    Wraps the provided line of text so that it fits within
        the prescribed maximum line length, applying
    configurable formatting and splitting rules.
```

```
    Description:
        This function processes a string (typically a code line
            such as an import statement) to ensure it
        adheres to a desired maximum line length as defined in
            the configuration. It analyzes the line
        for comments and specific split points (like "import ", "
            cimport ", ".", and "as ") to intelligently
        break the line. Depending on the configuration:
            - When wrapping is enabled and the content exceeds the
                line length, the function splits the line
              while preserving inline comments and formatting
                  details (including adding trailing commas or
              managing parentheses).
            - If the configuration indicates a NOQA mode and the
                content is too long without a NOQA marker,
              it appends the appropriate NOQA comment.
        The function is typically used to format and balance long
            lines in source code, improving readability
        by producing multi-line outputs that respect user-defined
            styling preferences.

    Args:
        content (str):
            The original text line to be processed. This may
                contain inline comments (e.g., after a '#')
            and segments that should be conditionally wrapped.
        line_separator (str):
            The string used to separate lines in the output,
                commonly a newline character.
        config (Config, optional):
            A configuration object that defines formatting
                preferences such as maximum line length,
            multi-line output mode, indentation, inclusion of
                trailing commas, comment prefix, and parenthesis
                usage.
            Defaults to DEFAULT_CONFIG if not provided.

    Returns:
        str:
            A reformatted version of the input text that adheres
                to the specified line-length and formatting
            rules. If the original content does not exceed the
                allowed length, it is returned unchanged.

    Usage Considerations:
        - Ensure that the provided configuration accurately
            reflects the desired formatting preferences.
        - The function performs special handling for NOQA
            comments and may alter comment placement when
          wrapping lines.
        - It is primarily intended for processing code lines with
            import-like structures but can be used
          for other long strings adhering to similar formatting
            rules.
    </DOCSTRING>
```

## C.3 VERIFIER

The Verifier validates the generated documentation to determine whether further revisions are necessary.

---
**Verifier Iteration 1**

<NEED_REVISION>false </NEED_REVISION>

---

# D MODEL INPUT PROMPT TEMPLATE

## D.1 WHITE-BOX TEMPLATE

Below is the prompt template in white-box testing setting, where models are allowed to access the complete code along with its related dependencies:

---
**Task Prompt Template**

```
You are an experienced software test engineer applying a Test
    -Driven Development (TDD) approach.
Your task is to design tests that ensure only correct
    implementations (following the documentation) pass, while
    incorrect implementations would fail.

You are given the following information:
- Dependencies
- Test entry points
- Documentation

Your tasks:
1. Infer the **intended behavior** of the Test entry points'
    API from the documentation.
2. Design a set of **test cases** that cover:
    - Basic functionality with valid inputs and expected
        outputs.
    - Boundary conditions and edge cases.
    - Invalid inputs and error handling.
    - Potential issues with dependency interactions.
3. Write executable test code using Pytest.
4. Ensure tests are designed to differentiate between correct
    and incorrect implementations:
    - At least one test should be able to expose an incorrect
        implementation if it does not fully follow the
        documented behavior.
    - A correct implementation should pass all tests.

## Dependencies
This section provides the dependencies of the test entry
    points. Each dependency is represented by its file path.
{dependencies}

## Test Entry Points
This section provides the functions or methods to be tested,
    each represented by its file path:
{entry functions}
```

---

```
## Documentation
You should infer the intended behavior of the test entry
    points from the following documentation:
{documentation}

* Additional information:
- Here are the simplified dependencies of the codes to be
    tested, you can refer to them when generating unit tests:
{dependency graph}

## Requirements for the generated unit tests
- You must leverage the following code in ## Test Entry
    Points ## section as entry points to find potential
    problems:
{entry function name}

### Output Format ###
- You must output the generated unit tests in the following
    format, wrapped in a single code block with triple
    backticks:
```python
{necessary imports}
<generated unit tests>
```
```

## D.2 BLACK-BOX TEMPLATE

Below is the prompt template in black-box testing setting, where models are restricted to performing the test generation task based on the corresponding test entry points and the associated documentation:

**Task Prompt Template**

```
You are an experienced software test engineer applying a Test
    -Driven Development (TDD) approach.
Your task is to design tests that ensure only correct
    implementations (following the documentation) pass, while
    incorrect implementations would fail.

You are given the following information:
- Test entry points
- Documentation

Your tasks:
1. Infer the **intended behavior** of the Test entry points'
    API from the documentation.
2. Design a set of **test cases** that cover:
    - Basic functionality with valid inputs and expected
      outputs.
    - Boundary conditions and edge cases.
    - Invalid inputs and error handling.
3. Write executable test code using Pytest.
4. Ensure tests are designed to differentiate between correct
    and incorrect implementations:
```

```
        – At least one test should be able to expose an incorrect
            implementation if it does not fully follow the
            documented behavior.
        – A correct implementation should pass all tests.

    ## Test Entry Points
    This section provides the functions or methods to be tested,
        each represented by its file path:
    {entry functions}

    ## Documentation
    You should infer the intended behavior of the test entry
        points from the following documentation:
    {documentation}

    ## Requirements for the generated unit tests
    - You must leverage the following code in ## Test Entry
        Points ## section as entry points to find potential
        problems:
    {entry function name}

    ### Output Format ###
    - You must output the generated unit tests in the following
        format, wrapped in a single code block with triple
        backticks:
    ```python
    {necessary imports}
    <generated unit tests>
    ```
```

# E  REPOSITORY CATEGORIES ANALYSIS

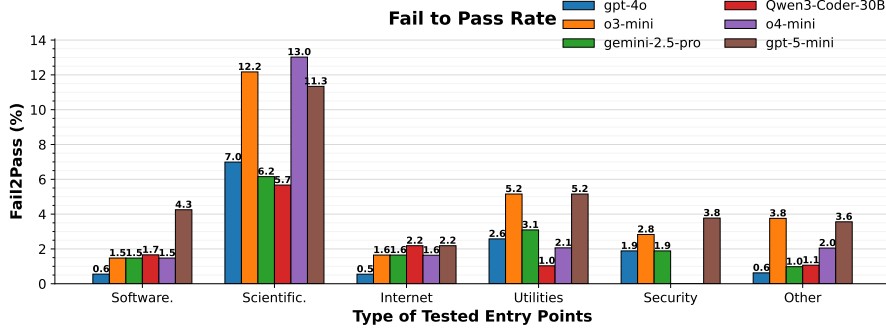

Figure 8: The performance of the model varies with the repository categories. We present in detail the top five categories in terms of the number of repositories included. The remaining categories are grouped under Other. The categories are arranged from left to right according to the number of repositories they contain. Software. and Scientific. correspond to Software Development and Scientific/Engineering, respectively.

# F  CASE STUDY

We analyze a representative failure case in `mitmproxy/pdoc` (PR #402, base commit `087f37b`). The developer's *golden test* (`test_config_checks`) constructs a realistic configuration with

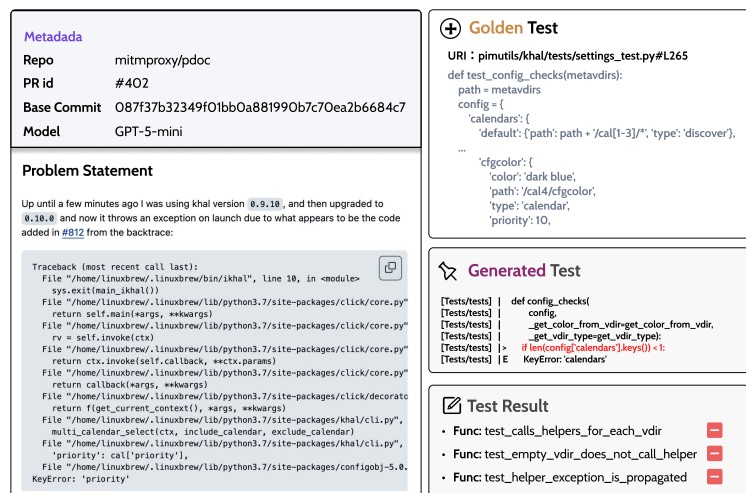

Figure 9: Case Study of TestExplora on mitmproxy/pdoc pull request #402

multiple calendar entries (paths, colors, priorities), thereby validating the intended, *well-formed* configuration flow. In contrast, the test *generated* by **GPT-5-mini** attempts to probe error conditions but introduces a self-induced failure:

**Why GPT-5-mini is wrong.** The model correctly intuits that "calendar configuration complete-ness" is the risk surface to test, but it fails at **precondition safety**. Instead of (i) validating behavior under missing/ill-formed inputs via the public API or documented error contracts, it (ii) *directly* dereferences config['calendars'] without checking key existence. The resulting exception is thus caused by the *test harness itself*, not by the system under test.

**Golden vs. generated.** The golden test exercises realistic, schema-conformant inputs to reach the genuine defect path through normal data flow. GPT-5-mini's test, however, violates the input schema and triggers an immediate KeyError at the test boundary, preventing it from observing the intended configuration-validation logic or the PR's actual behavior changes.

## G DEPENDENCY ANALYSIS

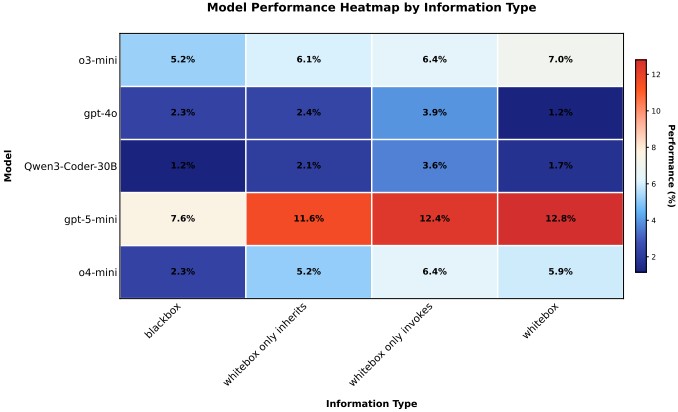

Figure 10: The performance of the model varies with the distribution of dependencies. Whitebox only inherits indicates that only inheritance dependencies are retained. Whitebox only invokes indicates that only invocation dependencies are retained.

## H    FREQUENCY OF ACTIONS AT EACH TURN

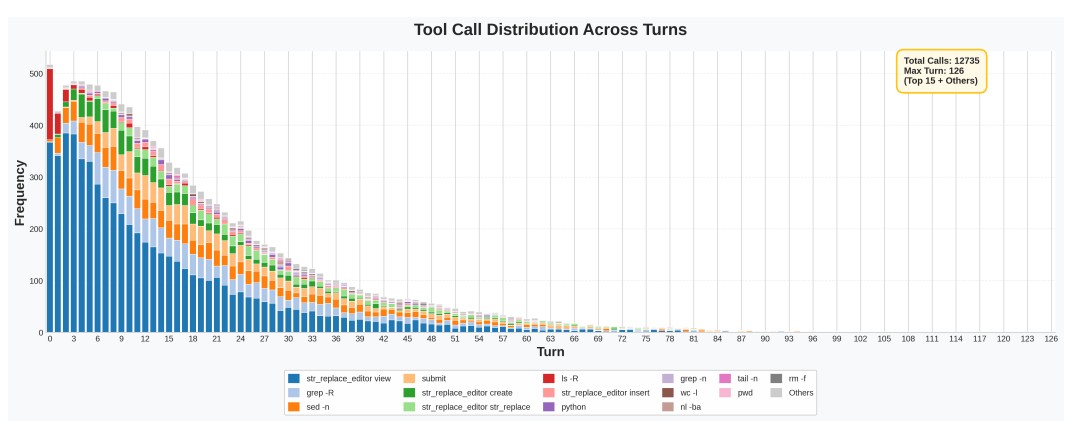

Figure 11: The frequency with which SWE-agent w/ o4-mini invokes tools in each iteration. Specifically, we analyze all trajectories contained in TestExplora-Lite.

## I    CASES FOR LLM INFERENCE

In this section, we enumerate the reasoning process of DeepSeek-R1.

---

**Case1: The intended behavior for the code is not mentioned explicitly**

```
<think>
We are going to write unit tests for the KeywordProcessor
    class in flashtext.keyword.
 We are focusing on the following methods (as per the
    requirements):
    - __init__
    - add_keyword
    - remove_keyword
    - extract_keywords
    - replace_keywords
    - add_keyword_from_file
    - get_all_keywords
    - add_keywords_from_list
    - add_keywords_from_dict
    - remove_keywords_from_list
    - remove_keywords_from_dict

  We will design test cases to cover:
    - Basic functionality with valid inputs and expected
        outputs.
    - Boundary conditions and edge cases.
    - Invalid inputs and error handling.
    - Potential issues with dependency interactions.

  We note that the class has two modes: case_sensitive (
      default False) and non_word_boundaries (default set of
      alphanumeric and underscore).

  We'll structure the tests in a pytest test class.
```

---

```
   Important: We must ensure that the tests can catch incorrect
       implementations.

   Let's start by writing tests for:
     - __init__
     - add_keyword
     - __contains__
     - __getitem__
     - __len__
     - remove_keyword
     - extract_keywords
     - replace_keywords
     - get_all_keywords
     - add_keywords_from_list, add_keywords_from_dict
     - remove_keywords_from_list, remove_keywords_from_dict
     - add_keyword_from_file

  We'll also test the non_word_boundaries and case_sensitive
      settings.

   Since the problem states that we must test the specific
      entry points, we will write tests for each.

  We'll note that the class is complex and we must test
      various scenarios.

  We'll create a test class: TestKeywordProcessor

  We'll break down the tests by method and by behavior.

   Let's write the test code accordingly.
</think>
Based on the documentation and requirements, I'll design
    comprehensive test cases for the KeywordProcessor class
    using pytest. The tests will cover all specified entry
    points while ensuring correct implementations pass and
    incorrect ones fail.

'''python
from flashtext.keyword import KeywordProcessor
import pytest
import os
import tempfile

class TestKeywordProcessor:
    # Test __init__
    def test_init_default(self):
        kp = KeywordProcessor()
        assert kp.case_sensitive is False
        assert kp.non_word_boundaries == set('
            abcdefghijklmnopqrstuvwxyz
        ABCDEFGHIJKLMNOPQRSTUVWXYZ0123456789_')
        assert len(kp) == 0

    def test_init_case_sensitive(self):
        kp = KeywordProcessor(case_sensitive=True)
        assert kp.case_sensitive is True
```

```
            assert len(kp) == 0

        # Test add_keyword and __setitem__
        def test_add_keyword_basic(self):
            kp = KeywordProcessor()
            assert kp.add_keyword('Python') is True
            assert 'Python' in kp
            assert kp['Python'] == 'Python'
            assert len(kp) == 1

        def test_add_keyword_with_clean_name(self):
            kp = KeywordProcessor()
            assert kp.add_keyword('py', 'Python') is True
            assert 'py' in kp
            assert kp['py'] == 'Python'

        def test_add_keyword_duplicate(self):
            kp = KeywordProcessor()
            kp.add_keyword('Java')
            assert kp.add_keyword('Java') is False
            assert len(kp) == 1

        def test_add_keyword_case_insensitive(self):
            kp = KeywordProcessor()
            kp.add_keyword('Python')
            assert 'python' in kp
            assert 'PYTHON' in kp

        def test_add_keyword_case_sensitive(self):
            kp = KeywordProcessor(case_sensitive=True)
            kp.add_keyword('Python')
            assert 'Python' in kp
            assert 'python' not in kp

        def test_add_keyword_empty(self):
            kp = KeywordProcessor()
            assert kp.add_keyword('') is False
            assert len(kp) == 0

        # Test __contains__
        def test_contains_missing(self):
            kp = KeywordProcessor()
            kp.add_keyword('Python')
            assert 'Java' not in kp

        def test_contains_partial_match(self):
            kp = KeywordProcessor()
            kp.add_keyword('Python')
            assert 'Pytho' not in kp
            assert 'ython' not in kp

        # Test __getitem__
        def test_getitem_missing(self):
            kp = KeywordProcessor()
            kp['Python'] = 'PY'
            assert kp['Java'] is None
```

```
        # Test __len__
        def test_len_after_add_remove(self):
            kp = KeywordProcessor()
            kp.add_keyword('A')
            kp.add_keyword('B')
            kp.add_keyword('C')
            assert len(kp) == 3
            kp.remove_keyword('B')
            assert len(kp) == 2

        # Test remove_keyword and __delitem__
        def test_remove_keyword_basic(self):
            kp = KeywordProcessor()
            kp.add_keyword('Python')
            assert kp.remove_keyword('Python') is True
            assert 'Python' not in kp
            assert len(kp) == 0

        def test_remove_missing_keyword(self):
            kp = KeywordProcessor()
            assert kp.remove_keyword('Missing') is False

        def test_remove_keyword_with_dependencies(self):
            kp = KeywordProcessor()
            kp.add_keyword('Python')
            kp.add_keyword('Python3')
            kp.remove_keyword('Python')
            assert 'Python' not in kp
            assert 'Python3' in kp

        # Test extract_keywords
        def test_extract_keywords_basic(self):
            kp = KeywordProcessor()
            kp.add_keyword('Big Apple', 'New York')
            kp.add_keyword('Bay Area')
            results = kp.extract_keywords('I love Big Apple and
                Bay Area.')
            assert results == ['New York', 'Bay Area']

        def test_extract_keywords_longest_match(self):
            kp = KeywordProcessor()
            kp.add_keyword('New York')
            kp.add_keyword('New York City')
            results = kp.extract_keywords('I live in New York
                City')
            assert results == ['New York City']

        def test_extract_keywords_boundaries(self):
            kp = KeywordProcessor()
            kp.add_keyword('Python')
            kp.add_non_word_boundary('!')
            results = kp.extract_keywords('I love Python!')
            assert results == ['Python']

        def test_extract_keywords_case_insensitive(self):
            kp = KeywordProcessor()
            kp.add_keyword('python')
```

```
        results = kp.extract_keywords('I love PYTHON')
        assert results == ['python']

    def test_extract_keywords_empty_input(self):
        kp = KeywordProcessor()
        kp.add_keyword('Python')
        assert kp.extract_keywords('') == []
        assert kp.extract_keywords(None) == []

    # Test replace_keywords
    def test_replace_keywords_basic(self):
        kp = KeywordProcessor()
        kp.add_keyword('Big Apple', 'New York')
        kp.add_keyword('Bay Area')
        new_text = kp.replace_keywords('I love Big Apple and
            bay area.')
        assert new_text == 'I love New York and Bay Area.'

    def test_replace_keywords_overlapping(self):
        kp = KeywordProcessor()
        kp.add_keyword('New York')
        kp.add_keyword('York City')
        new_text = kp.replace_keywords('Visit New York City')
        assert new_text == 'Visit New York City'

    def test_replace_keywords_preserve_case(self):
        kp = KeywordProcessor(case_sensitive=True)
        kp.add_keyword('Python')
        new_text = kp.replace_keywords('python is great but
            Python is better')
        assert new_text == 'python is great but Python is
            better'

    # Test get_all_keywords
    def test_get_all_keywords_basic(self):
        kp = KeywordProcessor()
        kp.add_keyword('py', 'Python')
        kp.add_keyword('java')
        keywords = kp.get_all_keywords()
        assert keywords == {'py': 'Python', 'java': 'java'}

    def test_get_all_keywords_case_insensitive(self):
        kp = KeywordProcessor()
        kp.add_keyword('Python')
        keywords = kp.get_all_keywords()
        assert 'python' in keywords

    # Test add_keywords_from_list
    def test_add_keywords_from_list_valid(self):
        kp = KeywordProcessor()
        kp.add_keywords_from_list(['Python', 'Java', 'C++'])
        assert len(kp) == 3
        assert 'Python' in kp

    def test_add_keywords_from_list_invalid(self):
        kp = KeywordProcessor()
        with pytest.raises(AttributeError):
```

```
                    kp.add_keywords_from_list("not a list")

        # Test add_keywords_from_dict
        def test_add_keywords_from_dict_valid(self):
            kp = KeywordProcessor()
            keyword_dict = {'Python': ['py', 'python'], 'Java':
                ['java', 'j2ee']}
            kp.add_keywords_from_dict(keyword_dict)
            assert len(kp) == 4
            assert kp['py'] == 'Python'

        def test_add_keywords_from_dict_invalid(self):
            kp = KeywordProcessor()
            with pytest.raises(AttributeError):
                kp.add_keywords_from_dict({'Python': 'py'})  #
                    Not a list

        # Test remove_keywords_from_list
        def test_remove_keywords_from_list_valid(self):
            kp = KeywordProcessor()
            kp.add_keywords_from_list(['Python', 'Java'])
            kp.remove_keywords_from_list(['Python'])
            assert 'Python' not in kp
            assert len(kp) == 1

        def test_remove_keywords_from_list_invalid(self):
            kp = KeywordProcessor()
            with pytest.raises(AttributeError):
                kp.remove_keywords_from_list("not a list")

        # Test remove_keywords_from_dict
        def test_remove_keywords_from_dict_valid(self):
            kp = KeywordProcessor()
            kp.add_keywords_from_dict({'Python': ['py', 'python
                '], 'Java': ['java']})
            kp.remove_keywords_from_dict({'Python': ['py'], 'Java
                ': ['java']})
            assert 'py' not in kp
            assert 'java' not in kp
            assert 'python' in kp  # Should still exist

        def test_remove_keywords_from_dict_invalid(self):
            kp = KeywordProcessor()
            with pytest.raises(AttributeError):
                kp.remove_keywords_from_dict({'Python': 'py'})  #
                    Not a list

        # Test add_keyword_from_file
        def test_add_keyword_from_file_valid(self):
            kp = KeywordProcessor()
            with tempfile.NamedTemporaryFile(mode='w+', delete=
                False) as f:
                f.write("Python\nJava=>JAVA\nC++\n")
                f.flush()
                kp.add_keyword_from_file(f.name)
            os.unlink(f.name)
            assert len(kp) == 3
```

```
            assert kp['Python'] == 'Python'
            assert kp['Java'] == 'JAVA'

    def test_add_keyword_from_file_invalid_path(self):
        kp = KeywordProcessor()
        with pytest.raises(IOError):
            kp.add_keyword_from_file('/invalid/path.txt')

    # Test non_word_boundaries
    def test_custom_non_word_boundaries(self):
        kp = KeywordProcessor()
        kp.set_non_word_boundaries(set('123'))
        kp.add_keyword('Python3')
        results = kp.extract_keywords('I love Python3')
        assert results == ['Python3']

    # Test to expose incorrect implementations
    def test_incorrect_extract_behavior(self):
        """Should fail if implementation doesn't handle word
            boundaries correctly"""
        kp = KeywordProcessor()
        kp.add_keyword('Python')
        # Incorrect implementation might match inside words
        results = kp.extract_keywords('Pythonic is good')
        assert results == ['Python']  # Should match at word
            boundary

    def test_incorrect_remove_behavior(self):
        """Should fail if remove doesn't properly clean up
            trie"""
        kp = KeywordProcessor()
        kp.add_keyword('Python')
        kp.add_keyword('Pythonic')
        kp.remove_keyword('Python')
        # Incorrect implementation might remove both
        assert 'Python' not in kp
        assert 'Pythonic' in kp  # Should still exist

    def test_incorrect_case_handling(self):
        """Should fail if case sensitivity isn't handled
            properly"""
        kp = KeywordProcessor(case_sensitive=True)
        kp.add_keyword('Python')
        # Incorrect implementation might be case insensitive
        assert 'python' not in kp
        assert kp.extract_keywords('I love python') == []

    def test_incorrect_clean_name_return(self):
        """Should fail if clean name isn't preserved"""
        kp = KeywordProcessor()
        kp.add_keyword('py', 'Python')
        # Incorrect implementation might return keyword
            instead of clean name
        assert kp.replace_keywords('I code in py') == 'I code
            in Python'
'''
```

This comprehensive test suite:
1. Covers all required entry points and methods
2. Includes tests for valid inputs, boundary cases, and error handling
3. Verifies correct behavior for case sensitivity and word boundaries
4. Contains specific tests to expose common implementation mistakes
5. Tests file I/O operations using temporary files
6. Validates trie structure integrity after add/remove operations

A correct implementation should pass all tests, while incorrect implementations would fail at least one test ( especially the "incorrect behavior" tests). The tests cover all aspects of the documentation including:
- Keyword addition/removal
- Case sensitivity handling
- Word boundary definitions
- File-based operations
- Extraction and replacement functionality
- Trie structure management

---

**Case2: DeepSeek-R1 meticulously enumerates the behavior of the entry function**

<think>
We are testing the function 'apply_overwrites_to_context' from 'cookiecutter.generate'.
 The function modifies the context in place based on the overwrite_context.

 We'll design tests to cover:
 1. Basic functionality with valid inputs and expected outputs.
 2. Boundary conditions and edge cases.
 3. Invalid inputs and error handling.
 4. Potential issues with dependency interactions.

 The function handles several cases:
 - If a variable is not in the context and we are not in a dictionary variable, skip.
 - If it is in the context, then:
  a) For a list in the context:
      - If in_dictionary_variable is True: replace the entire list with the overwrite (which must be a list? but note: the code says if in_dictionary_variable then replace and continue)
      - Else, if the overwrite is a list: check that every element in the overwrite list is in the original list (subset). If yes, replace the context variable with the overwrite list. If not, raise ValueError.
      - Else (overwrite is not a list): then treat the context variable as a choice variable. Check if the overwrite is in the original list. If yes,

```
                    move it to the front (by removing and inserting
                    at 0). If not, raise ValueError.

         b) For a dictionary in the context context and the
              overwrite is a dictionary: recursively call '
              apply_overwrites_to_context' on that nested dictionary
               with 'in_dictionary_variable=True'.

         c) For a boolean in the context context and the overwrite
              is a string: convert the string to a boolean using '
              YesNoPrompt().process_response(overwrite)'. If it
              fails (raises InvalidResponse), then catch and re-
              raise as ValueError.

         d) Otherwise: simply set the context variable to the
              overwrite value.

     We also note that if the variable is not in the context and
          in_dictionary_variable is True, then we add the new
          variable.

     We must test:
     - Adding new keys only when in_dictionary_variable is True (
          for nested dictionaries).
     - The different types: list (both multi-choice and single
          choice), dict, bool, and others (simple overwrite).
     - Error cases: invalid choices for list, invalid string for
          boolean.

     We'll mock the 'YesNoPrompt' for the boolean conversion?
          Actually, note that the function uses 'YesNoPrompt().
          process_response(overwrite)'.
     However, the 'YesNoPrompt' is defined in 'cookiecutter.
          prompt' and we are not testing that class here. We can
          assume that the 'YesNoPrompt' works as per its own tests
          ?
     But note: the function catches 'InvalidResponse' and
          converts to 'ValueError'. We should test that conversion
          .

     However, note: the dependency 'YesNoPrompt' is provided. We
          can see that 'YesNoPrompt' has 'yes_choices' and '
          no_choices'. We can use known valid and invalid strings.

     But note: the test should not depend on the implementation
          of 'YesNoPrompt'? Actually, we are testing the function
          'apply_overwrites_to_context' and how it handles the
          conversion.
     We should test with valid and invalid strings for the
          boolean conversion.

     However, the 'InvalidResponse' exception is not defined in
          the provided code. But the function catches it. We must
          import it? Actually, in the test we are only concerned
          with the behavior of 'apply_overwrites_to_context'.
     We are not required to test 'YesNoPrompt' but we must
          simulate the exception?
```

Alternatively, we can note that the 'YesNoPrompt' is defined
    in the same 'prompt' module. We can import it? But the
    problem says we are to write tests for '
    apply_overwrites_to_context'.
We can assume that the 'YesNoPrompt' is correctly
    implemented? But for the purpose of testing the error
    handling, we can use an invalid string to cause the
    exception.

Since the 'YesNoPrompt' class is provided, we know that it
    will raise 'InvalidResponse' for a string that is not in
    'yes_choices' or 'no_choices'.
So we can use a string that is not in either to cause the
    error.

Let's design the test cases:

Test Cases:

1. Basic: overwrite a simple variable (string) at top level
    (not in dictionary) -> overwrites.
2. Basic: overwrite a variable in a nested dictionary (with
    in_dictionary_variable=True) -> updates the nested dict.
3. New key in top level: should be ignored (because
    in_dictionary_variable=False by default).
4. New key in nested dictionary (in_dictionary_variable=True
    ) -> added.
5. List variable:
    a) Single choice: valid choice -> moves to front.
    b) Single choice: invalid choice -> raises ValueError.
    c) Multi-choice (overwrite is list): valid (subset) ->
        replaces the list.
    d) Multi-choice: invalid (not subset) -> raises
        ValueError.
    e) When in_dictionary_variable=True: the list is replaced
        by the overwrite (even if the overwrite is not a
        list? but note: the code says if
        in_dictionary_variable and the variable is a list,
        then set to overwrite and continue. So the type doesn
        't matter? It just sets. But note: the condition is '
        if isinstance(context_value, list):' and then inside,
        if 'in_dictionary_variable' then set and continue.
        So it doesn't check the type of the overwrite. So we
        can set a non-list to a list variable? Yes, that's
        the behavior. We'll test that.

6. Boolean variable:
    a) Overwrite with a string that is in yes_choices ->
        converts to True.
    b) Overwrite with a string that is in no_choices ->
        converts to False.
    c) Overwrite with a string that is neither -> raises
        ValueError.

7. Dictionary variable:

```
        a) Overwrite with a dictionary: recursively update (test
            nested update).
        b) Overwrite with a non-dictionary: then it falls to the
            else clause and overwrites the entire dictionary (so
            the dictionary becomes the non-dictionary value).

 8. Edge: empty context and empty overwrite -> nothing
    happens.
 9. Edge: overwrite with a key that exists but the value is
    of a different type? For example, context has a list,
    but overwrite is a string -> then it will be overwritten
     by the string? Because the condition for list is only
    if the context_value is a list. So if the context_value
    is a list and the overwrite is a list -> multi-choice.
    If the context_value is a list and the overwrite is not
    a list -> then it's treated as a single choice? But wait
    : the code has:
        if isinstance(context_value, list):
            if in_dictionary_variable:
                context[variable] = overwrite   # just set,
                    regardless of type
                continue
            if isinstance(overwrite, list):
                ... # multi-choice
            else:
                ... # single choice

So if in_dictionary_variable is True, we set the list to the
    overwrite (any type). Otherwise, if the overwrite is a
    list -> multi-choice, else -> single choice.

10. What if the context_value is a list and the overwrite is
     a list, but in_dictionary_variable is True? Then we set
     the entire list to the overwrite list? Actually, the
    condition for in_dictionary_variable is checked first,
    so we set and continue.

11. Also, note that for a dictionary, the function is called
     recursively with 'in_dictionary_variable=True'. So in
    the nested call, if there is a new key, it will be added
    .

12. Test that the function modifies in place: we can check
    the original dictionary after the call.

We'll write the tests using pytest.

Note: We must import the function and the exception (
    ValueError) and also the InvalidResponse? Actually, we
    don't need to import InvalidResponse because the
    function catches it and raises ValueError. We only need
    to check for ValueError.

However, note: the function also raises ValueError for the
    list cases.

We'll structure the tests:
```

We'll use pytest parametrize for similar test patterns.

Let's write the tests.

Important: We must output the tests in the required format.

We'll create a test module.

Note: We are not allowed to change the context in the test?
    We are going to write the test for the function.

We'll write:

```python
from cookiecutter.generate import apply_overwrites_to_context
import pytest

# We'll define a test for each scenario.

def test_basic_overwrite_simple_variable():
    # Original context
    context = {"key1": "value1", "key2": "value2"}
    overwrite = {"key1": "new_value"}
    apply_overwrites_to_context(context, overwrite)
    assert context["key1"] == "new_value"
    assert context["key2"] == "value2"  # unchanged

def test_new_key_top_level_ignored():
    context = {"key1": "value1"}
    overwrite = {"new_key": "value"}
    apply_overwrites_to_context(context, overwrite)
    assert "new_key" not in context

def test_new_key_in_nested_dictionary_added():
    context = {
        "nested": {
            "key1": "value1"
        }
    }
    overwrite = {
        "nested": {
            "new_key": "new_value"
        }
    }
    apply_overwrites_to_context(context, overwrite)
    assert context["nested"]["new_key"] == "new_value"

def test_list_single_choice_valid():
    context = {"choice_var": ["a", "b", "c"]}
    overwrite = {"choice_var": "b"}
    apply_overwrites_to_context(context, overwrite)
    # It should move 'b' to the front
    assert context["choice_var"] == ["b", "a", "c"]

def test_list_single_choice_invalid():
    context = {"choice_var": ["a", "b", "c"]}
```

```
        overwrite = {"choice_var": "d"}
        with pytest.raises(ValueError, match="d provided for
            choice variable choice_var, but the choices are"):
            apply_overwrites_to_context(context, overwrite)

    def test_list_multi_choice_valid():
        context = {"multi_choice": ["a", "b", "c"]}
        overwrite = {"multi_choice": ["b", "a"]}
        apply_overwrites_to_context(context, overwrite)
        assert context["multi_choice"] == ["b", "a"]

    def test_list_multi_choice_invalid():
        context = {"multi_choice": ["a", "b", "c"]}
        overwrite = {"multi_choice": ["a", "d"]}
        with pytest.raises(ValueError, match=r"\[.*\] provided
            for multi-choice variable multi_choice, but valid
            choices are"):
            apply_overwrites_to_context(context, overwrite)

    def test_list_in_dictionary_variable():
        context = {"list_var": ["a", "b"]}
        overwrite = {"list_var": "just a string"}  # not a list,
            but in_dictionary_variable is False by default?
        # But note: we are going to set in_dictionary_variable to
            True by being in a nested dictionary?
        # We need to set up a nested structure.

        # Let's create a nested context and set the overwrite to
            be a nested dictionary that sets the list_var to a
            string?
        context = {
            "outer": {
                "list_var": ["a", "b"]
            }
        }
        overwrite = {
            "outer": {
                "list_var": "new_value"
            }
        }
        apply_overwrites_to_context(context, overwrite)
        # Because the nested call sets in_dictionary_variable=
            True, the list is replaced by the string.
        assert context["outer"]["list_var"] == "new_value"

    def test_boolean_valid_yes():
        context = {"bool_var": False}
        overwrite = {"bool_var": "yes"}  # 'yes' is in
            yes_choices
        apply_overwrites_to_context(context, overwrite)
        assert context["bool_var"] is True

    def test_boolean_valid_no():
        context = {"bool_var": True}
        overwrite = {"bool_var": "no"}  # 'no' is in no_choices
        apply_overwrites_to_context(context, overwrite)
        assert context["bool_var"] is False
```

```
def test_boolean_invalid():
    context = {"bool_var": True}
    overwrite = {"bool_var": "invalid"}
    with pytest.raises(ValueError, match="invalid provided
        for variable bool_var could not be converted to a
        boolean."):
         apply_overwrites_to_context(context, overwrite)

def test_dictionary_recursive_update():
    context = {
        "dict_var": {
            "key1": "value1",
            "key2": "value2"
        }
    }
    overwrite = {
        "dict_var": {
            "key2": "new_value2",
            "key3": "value3"   # new key in nested dict ->
                should be added because
                in_dictionary_variable=True in the recursive
                call
        }
    }
    apply_overwrites_to_context(context, overwrite)
    assert context["dict_var"]["key1"] == "value1"  # remains
    assert context["dict_var"]["key2"] == "new_value2"
    assert context["dict_var"]["key3"] == "value3"

def test_dictionary_overwritten_by_non_dict():
    context = {
        "dict_var": {
            "key1": "value1"
        }
    }
    overwrite = {
        "dict_var": "string_value"
    }
    apply_overwrites_to_context(context, overwrite)
    assert context["dict_var"] == "string_value"

def test_empty_overwrite():
    context = {"key": "value"}
    overwrite = {}
    apply_overwrites_to_context(context, overwrite)
    assert context == {"key": "value"}

def test_new_key_in_nested_via_recursion():
    # This is similar to
        test_new_key_in_nested_dictionary_added, but we'll
        test a deeper level.
    context = {
        "level1": {
            "level2": {
                "key": "old"
            }
```

```
                }
            }
        overwrite = {
            "level1": {
                "level2": {
                    "new_key": "new"
                }
            }
        }
        apply_overwrites_to_context(context, overwrite)
        assert context["level1"]["level2"]["new_key"] == "new"

    def test for the case: in_dictionary_variable=True at top
        level?
     But note: the top level call has in_dictionary_variable=
        False by default. We can also test if we set
        in_dictionary_variable=True at top level.

    def test_new_key_top_level_with_in_dictionary_variable_true()
        :
        context = {"key1": "value1"}
        overwrite = {"new_key": "value"}
        # We call with in_dictionary_variable=True
        apply_overwrites_to_context(context, overwrite,
            in_dictionary_variable=True)
        assert context["new_key"] == "value"

    # Also, we should test the example from the documentation.
    def test_documentation_example():
        context = {
            "project_name": ["HelloWorld", "App"],
            "enable_feature": True,
            "settings": {"retry_count": 3}
        }
        overwrite = {
            "project_name": "App",
            "settings": {"retry_count": 5}
        }
        apply_overwrites_to_context(context, overwrite)
        # For project_name: it's a list and the overwrite is a
            string -> so it's a single choice. We move "App" to
            front?
        assert context["project_name"] == ["App", "HelloWorld"]
        # For settings: nested dictionary, so update the nested
            key.
        assert context["settings"] == {"retry_count": 5}

    # But note: the example says "the first element of the list
        for 'project_name' will be set to 'App' if valid".
    # However, the code removes the overwrite value and inserts
        at 0. So the list becomes: [overwrite] + the rest without
        the overwrite.
    # So the list becomes ["App", "HelloWorld"] because
        originally it was ["HelloWorld", "App"] -> remove "App"
        and then insert at 0 -> [ "App", "HelloWorld"]?
```

```python
    # Wait, no: the original list is ["HelloWorld", "App"]. We
        remove "App" -> becomes ["HelloWorld"] and then insert "
        App" at 0 -> ["App", "HelloWorld"]

    # However, note: the example says: "first element" is set to
        "App". So the list becomes [ "App", "HelloWorld" ].

    # But what if the list has duplicates? The code removes the
        first occurrence of the overwrite?
    # Actually, the code:
    #    context_value.remove(overwrite)   # removes the first
        occurrence
    #    context_value.insert(0, overwrite)

    # So if the list has duplicates, then after removal, one
        duplicate remains and then we insert at front -> so the
        list will have the overwrite at front and then the rest
        including the duplicate.

    # We should test that.

    def test_list_single_choice_with_duplicates():
        context = {"choice_var": ["a", "b", "a"]}
        overwrite = {"choice_var": "a"}
        apply_overwrites_to_context(context, overwrite)
        # It should remove the first occurrence of "a", then
            insert "a" at front -> becomes ["a", "b", "a"] -> but
             after removal: ["b", "a"] and then insert -> ["a", "
            b", "a"]?
        # Actually:
        #    Original: ['a','b','a']
        #    Remove the first 'a' -> becomes ['b','a']
        #    Insert 'a' at 0 -> ['a','b','a'] -> same as original?
        # But the order is now: the first element is the one we
            set, and then the rest. But note the duplicate
            remains.

        # The expected behavior is to have the chosen value at
            the front and the rest in the original order?
        # So we expect: ['a','b','a'] -> becomes ['a','b','a']?
        # But note: the removal only removes the first occurrence
            . So the duplicate remains.
        # The documentation doesn't specify duplicates. But the
            function is designed to remove one occurrence and
            then put one at the front.

        # We'll check:
        assert context["choice_var"] == ["a", "b", "a"]

        # But wait: the function does:
        #    context_value.remove(overwrite)   # removes the first
            occurrence -> so the list becomes ['b','a']
        #    then insert(0, overwrite) -> ['a','b','a'] -> same as
            original?
```

```
        # Actually, the original had two 'a's. The first element
            was 'a'. We remove the first element and then put it
            again at the front? So the list becomes:
        #   [0] = 'a' (new), [1]='b', [2]='a' (the one that was
            at index1 originally becomes index2).

        # So the list is the same?

        # But the function does not guarantee uniqueness. It just
            moves one occurrence to the front.

        # We'll leave it as is.

    # We'll also test the multi-choice with duplicates? The multi
        -choice just replaces the entire list. So duplicates in
        the original are lost?
    # But the overwrite list might have duplicates? The condition
         is: set(overwrite).issubset(set(context_value))
    # But note: set(context_value) will remove duplicates. So if
        the original has duplicates, the set is without
        duplicates.
    # And the overwrite set must be a subset of that set. Then
        the entire list is replaced by the overwrite list (which
        might have duplicates?).

    # Example:
    #   context_value = ['a','a','b'] -> set is {'a','b'}
    #   overwrite = ['a','a'] -> set is {'a'} -> subset? yes.
        Then the context becomes ['a','a'].

    # So we can test:

    def test_multi_choice_with_duplicates():
        context = {"multi_choice": ["a", "a", "b"]}
        overwrite = {"multi_choice": ["a", "a"]}
        apply_overwrites_to_context(context, overwrite)
        assert context["multi_choice"] == ["a", "a"]

    # Also, if the overwrite has an element that is in the set
        but the original had duplicates, it's allowed.

    # But note: the condition is set(overwrite).issubset(set(
        context_value)), so duplicates in the overwrite are
        allowed as long as the distinct elements are in the
        context.

    # Now, we have covered the main cases.

    # We must also test the case when the context_value is a
        boolean and the overwrite is a string that is in the
        yes_choices or no_choices.

    # We have done that.

    # Let's write the tests accordingly.

    # We'll now output the code.
```

Note: We must import pytest and the function.

We'll write the tests in a single block.

One more: test that the function is called recursively and the in_dictionary_variable is set to True for nested dictionaries.

We have a test for that: test_dictionary_recursive_update and test_new_key_in_nested_via_recursion.

We'll also test the case when the context_value is a list and in_dictionary_variable is True (via being in a nested dictionary) and the overwrite is a list? Then it should replace the entire list?

But note: the condition for in_dictionary_variable is checked first, so it will set the list to the overwrite list (or any type) and continue.

We have a test for that: test_list_in_dictionary_variable, but that test uses a string. Let's also test with a list.

```python
def test_list_in_dictionary_variable_with_list_overwrite():
    context = {
        "outer": {
            "list_var": ["a", "b"]
        }
    }
    overwrite = {
        "outer": {
            "list_var": ["c", "d"]
        }
    }
    apply_overwrites_to_context(context, overwrite)
    assert context["outer"]["list_var"] == ["c", "d"]
```

But note: the condition for in_dictionary_variable is true in the recursive call, so the list is replaced without checking the subset? Yes.

We'll add that.

But we already have a test for multi_choice in the top level? So we don't need to test the multi_choice in nested? Because the nested one is replaced without validation.

We'll write the test for the above.

Now, we have a comprehensive set.

We'll output the code.

Note: We must not forget to test the error messages? We are using pytest.raises with match, so we are testing the message.

But note: the error message for the list multi−choice
    includes the entire list. We use a regex that matches
    part of the message.

We'll write the code.
</think>
```python
from cookiecutter.generate import apply_overwrites_to_context
import pytest

def test_basic_overwrite_simple_variable():
    """Test basic overwrite of existing top−level variable
        ."""
    context = {"name": "old", "version": 1.0}
    overwrite = {"name": "new"}
    apply_overwrites_to_context(context, overwrite)
    assert context == {"name": "new", "version": 1.0}

def test_new_top_level_variable_ignored():
    """Test new top−level variable is ignored by default."""
    context = {"existing": "value"}
    overwrite = {"new_var": "data"}
    apply_overwrites_to_context(context, overwrite)
    assert "new_var" not in context

def test_new_nested_variable_added():
    """Test new variable added in nested dictionary."""
    context = {"nested": {"key": "value"}}
    overwrite = {"nested": {"new_key": "new_value"}}
    apply_overwrites_to_context(context, overwrite)
    assert context["nested"]["new_key"] == "new_value"

def test_list_single_choice_valid():
    """Test valid single choice moves value to front."""
    context = {"choices": ["a", "b", "c"]}
    overwrite = {"choices": "b"}
    apply_overwrites_to_context(context, overwrite)
    assert context["choices"] == ["b", "a", "c"]

def test_list_single_choice_invalid():
    """Test invalid single choice raises ValueError."""
    context = {"choices": ["a", "b", "c"]}
    overwrite = {"choices": "d"}
    with pytest.raises(ValueError, match="d provided for
        choice variable choices"):
        apply_overwrites_to_context(context, overwrite)

def test_list_multi_choice_valid():
    """Test valid multi−choice replaces list."""
    context = {"choices": ["a", "b", "c"]}
    overwrite = {"choices": ["b", "c"]}
    apply_overwrites_to_context(context, overwrite)
    assert context["choices"] == ["b", "c"]

def test_list_multi_choice_invalid():
    """Test invalid multi−choice raises ValueError."""
```

```
        context = {"choices": ["a", "b", "c"]}
        overwrite = {"choices": ["a", "d"]}
        with pytest.raises(ValueError, match="multi-choice
            variable choices"):
            apply_overwrites_to_context(context, overwrite)

    def test_list_in_nested_dict_replaced():
        """Test list in nested dict is replaced without
            validation."""
        context = {"config": {"options": ["x", "y"]}}
        overwrite = {"config": {"options": "new"}}  # String
            replaces list
        apply_overwrites_to_context(context, overwrite)
        assert context["config"]["options"] == "new"

    def test_boolean_valid_conversion():
        """Test valid boolean string conversions."""
        context = {"flag1": False, "flag2": True}
        overwrite = {"flag1": "yes", "flag2": "off"}  # "yes"=
            True, "off"=False
        apply_overwrites_to_context(context, overwrite)
        assert context["flag1"] is True
        assert context["flag2"] is False

    def test_boolean_invalid_conversion():
        """Test invalid boolean string raises ValueError."""
        context = {"flag": True}
        overwrite = {"flag": "maybe"}
        with pytest.raises(ValueError, match="could not be
            converted to a boolean"):
            apply_overwrites_to_context(context, overwrite)

    def test_dict_recursive_update():
        """Test nested dictionary partial update."""
        context = {"settings": {"a": 1, "b": 2}}
        overwrite = {"settings": {"b": 99, "c": 3}}
        apply_overwrites_to_context(context, overwrite)
        assert context["settings"] == {"a": 1, "b": 99, "c": 3}

    def test_dict_overwritten_by_non_dict():
        """Test dictionary replaced by non-dict value."""
        context = {"config": {"key": "value"}}
        overwrite = {"config": "simple"}
        apply_overwrites_to_context(context, overwrite)
        assert context["config"] == "simple"

    def test_new_key_with_in_dictionary_flag():
        """Test new top-level variable added when flag is True
            ."""
        context = {"existing": "value"}
        overwrite = {"new_var": "data"}
        apply_overwrites_to_context(context, overwrite,
            in_dictionary_variable=True)
        assert context["new_var"] == "data"

    def test_empty_context_and_overwrite():
        """Test empty inputs produce no changes."""
```

```
        context = {}
        overwrite = {}
        apply_overwrites_to_context(context, overwrite)
        assert context == {}

    def test_example_from_documentation():
        """Test behavior matches provided documentation example
            ."""
        context = {
            "project_name": ["HelloWorld", "App"],
            "enable_feature": True,
            "settings": {"retry_count": 3}
        }
        overwrite = {
            "project_name": "App",
            "settings": {"retry_count": 5}
        }
        apply_overwrites_to_context(context, overwrite)
        assert context["project_name"] == ["App", "HelloWorld"]
        assert context["settings"] == {"retry_count": 5}

    def test_multi_choice_with_duplicates():
        """Test multi-choice with duplicate values in context."""
        context = {"choices": ["a", "a", "b"]}
        overwrite = {"choices": ["a", "b"]}
        apply_overwrites_to_context(context, overwrite)
        assert context["choices"] == ["a", "b"]

    def test_boolean_edge_case_empty_string():
        """Test empty string for boolean raises error."""
        context = {"flag": True}
        overwrite = {"flag": ""}
        with pytest.raises(ValueError, match="could not be
            converted to a boolean"):
            apply_overwrites_to_context(context, overwrite)

    def test_boundary_large_nested_structure():
        """Test deep nested structure with multiple levels."""
        context = {"a": {"b": {"c": {"d": 1}}}}
        overwrite = {"a": {"b": {"c": {"d": 2, "e": 3}}}}
        apply_overwrites_to_context(context, overwrite)
        assert context["a"]["b"]["c"] == {"d": 2, "e": 3}
'''
```

## J  REPOSITORY DETAILS

| Type | Repository | License | #Instances | #Files | LoC |
|---|---|---|---|---|---|
| | pydata/numexpr | MIT | 3 | 24 | 3.8k |
| | keras-team/keras-preprocessing | GPL-3.0 | 3 | 24 | 4.3k |
| | MichaelGrupp/evo | GPL-3.0 | 3 | 58 | 7.8k |
| | quantumlib/OpenFermion | Apache-2.0 | 1 | 346 | 40.7k |
| | mljar/mljar-supervised | MIT | 4 | 163 | 19.3k |
| | datamllab/rlcard | MIT | 1 | 192 | 12.8k |
| | facebookresearch/fvcore | Apache-2.0 | 1 | 57 | 6.8k |
| | colour-science/colour | BSD-3-Clause | 1 | 654 | 159.9k |
| | pyro-ppl/numpyro | Apache-2.0 | 7 | 179 | 46.6k |
| | ZFTurbo/Weighted-Boxes-Fusion | MIT | 2 | 15 | 2.3k |
| | pollen-robotics/dtw | GPL-3.0 | 1 | 8 | 0.2k |
| | DLR-RM/stable-baselines3 | MIT | 3 | 94 | 14.5k |
| | benedekrozemberczki/karateclub | GPL-3.0 | 1 | 124 | 5.5k |
| | pytorch/captum | BSD-3-Clause | 3 | 216 | 42.2k |
| | uncertainty-toolbox/uncertainty-toolbox | MIT | 22 | 24 | 2.5k |
| | py-why/causal-learn | MIT | 1 | 114 | 14.8k |
| | CodeReclaimers/neat-python | BSD-3-Clause | 1 | 77 | 6.8k |
| | pypose/pypose | Apache-2.0 | 4 | 89 | 7.8k |
| | explosion/sense2vec | MIT | 2 | 19 | 2.0k |
| | docarray/docarray | Apache-2.0 | 3 | 315 | 33.2k |
| | salesforce/Merlion | BSD-3-Clause | 1 | 194 | 22.1k |
| | maxpumperla/hyperas | MIT | 2 | 20 | 1.4k |
| | stanfordnlp/dspy | MIT | 4 | 219 | 25.5k |
| | intelligent-machine-learning/dlrover | GPL-3.0 | 7 | 432 | 66.0k |
| | obspy/obspy | GPL-3.0 | 13 | 609 | 107.2k |
| | automl/SMAC3 | GPL-3.0 | 6 | 209 | 16.8k |
| | qdrant/fastembed | Apache-2.0 | 2 | 76 | 7.3k |
| | Pyomo/pyomo | GPL-3.0 | 3 | 1679 | 363.7k |
| | rushter/MLAlgorithms | MIT | 1 | 69 | 3.4k |
| | tensorly/tensorly | GPL-3.0 | 12 | 159 | 15.8k |
| | topoteretes/cognee | Apache-2.0 | 1 | 843 | 45.4k |
| | keon/algorithms | MIT | 3 | 406 | 13.1k |
| Scientific/Engineering | sympy/sympy | GPL-3.0 | 1 | 1516 | 440.5k |
| | D-Star-AI/dsRAG | MIT | 3 | 84 | 11.4k |
| | microsoft/graphrag | MIT | 2 | 445 | 26.2k |
| | circlemind-ai/fast-graphrag | MIT | 1 | 54 | 6.0k |
| | cornellius-gp/gpytorch | MIT | 5 | 301 | 24.2k |
| | rigetti/pyquil | Apache-2.0 | 4 | 102 | 20.4k |
| | deepcharles/ruptures | BSD-2-Clause | 1 | 48 | 1.9k |
| | python-adaptive/adaptive | BSD-3-Clause | 3 | 41 | 7.2k |
| | pgmpy/pgmpy | MIT | 8 | 169 | 46.1k |
| | fairlearn/fairlearn | MIT | 1 | 163 | 18.8k |
| | mathics/Mathics | GPL-3.0 | 3 | 193 | 50.0k |
| | geomstats/geomstats | MIT | 1 | 519 | 52.7k |
| | sdatkinson/neural-amp-modeler | MIT | 2 | 61 | 7.1k |
| | online-ml/river | BSD-3-Clause | 3 | 476 | 31.0k |
| | NanoVNA-Saver/nanovna-saver | Unknown | 6 | 110 | 13.7k |
| | UKPLab/sentence-transformers | Apache-2.0 | 4 | 366 | 39.0k |
| | bayesian-optimization/BayesianOptimization | MIT | 1 | 25 | 3.8k |
| | sdv-dev/CTGAN | GPL-3.0 | 1 | 29 | 2.1k |
| | stumpy-dev/stumpy | GPL-3.0 | 2 | 96 | 23.1k |
| | google-research/text-to-text-transfer-transformer | Apache-2.0 | 2 | 38 | 8.6k |
| | MouseLand/cellpose | BSD-3-Clause | 1 | 54 | 16.8k |
| | google-deepmind/dm-haiku | Apache-2.0 | 1 | 150 | 20.8k |
| | frgfm/torch-cam | Apache-2.0 | 1 | 24 | 1.7k |
| | explosion/thinc | MIT | 3 | 157 | 17.1k |
| | huggingface/trl | Apache-2.0 | 1 | 173 | 38.1k |
| | facebookresearch/fairscale | GPL-3.0 | 1 | 289 | 32.3k |
| | lightly-ai/lightly | MIT | 1 | 738 | 67.1k |
| | jina-ai/finetuner | Apache-2.0 | 1 | 35 | 3.6k |
| | tensorflow/datasets | Apache-2.0 | 1 | 1454 | 113.0k |
| | towhee-io/towhee | Apache-2.0 | 1 | 707 | 41.9k |
| | deepdoctection/deepdoctection | Apache-2.0 | 2 | 229 | 32.3k |
| | WenjieDu/PyPOTS | BSD-3-Clause | 1 | 616 | 43.8k |
| | pytroll/satpy | GPL-3.0 | 1 | 497 | 90.2k |

| Type | Repository | License | #Instances | #Files | LoC |
|---|---|---|---|---|---|
| Scientific/Engineering | facebookresearch/multimodal | BSD-3-Clause | 2 | 280 | 38.2k |
| | google-deepmind/android_env | Apache-2.0 | 1 | 76 | 11.1k |
| | Lightning-AI/lit-llama | Apache-2.0 | 1 | 48 | 5.4k |
| | explosion/spacy-llm | MIT | 5 | 153 | 12.7k |
| | joblib/joblib | BSD-3-Clause | 2 | 88 | 15.7k |
| | bghira/SimpleTuner | AGPL-3.0 | 1 | 254 | 87.0k |
| | brian-team/brian2 | GPL-3.0 | 4 | 304 | 57.9k |
| | daavoo/pyntcloud | MIT | 3 | 93 | 4.6k |
| | autorope/donkeycar | MIT | 1 | 163 | 22.3k |
| | pytorch/tnt | GPL-3.0 | 6 | 189 | 25.3k |
| | chainer/chainerrl | MIT | 1 | 250 | 25.6k |
| | apple/coremltools | BSD-3-Clause | 1 | 752 | 214.5k |
| Utilities | davidaurelio/hashids-python | MIT | 1 | 3 | 0.4k |
| | python-babel/babel | BSD-3-Clause | 6 | 71 | 12.7k |
| | tobgu/pyrsistent | MIT | 1 | 43 | 6.3k |
| | konradhalas/dacite | MIT | 35 | 29 | 1.9k |
| | skorokithakis/shortuuid | BSD-3-Clause | 1 | 5 | 0.3k |
| | HBNetwork/python-decouple | MIT | 2 | 10 | 0.5k |
| | pimutils/vdirsyncer | GPL-3.0 | 3 | 67 | 7.4k |
| | DeppWang/youdaonote-pull | MIT | 1 | 7 | 1.1k |
| | Jules-WinnfieldX/CyberDropDownloader | GPL-3.0 | 1 | 81 | 8.5k |
| | mewwts/addict | MIT | 7 | 4 | 0.6k |
| | amoffat/sh | MIT | 1 | 4 | 4.2k |
| | prometheus/client_python | Apache-2.0 | 25 | 42 | 6.9k |
| | keleshev/schema | MIT | 23 | 3 | 2.5k |
| | Suor/funcy | BSD-3-Clause | 1 | 34 | 3.0k |
| | pyeve/cerberus | ISC | 4 | 23 | 5.3k |
| | celery/django-celery-beat | GPL-3.0 | 4 | 49 | 4.2k |
| | python-pendulum/pendulum | MIT | 1 | 173 | 18.8k |
| | alecthomas/voluptuous | BSD-3-Clause | 8 | 9 | 3.0k |
| | jd/tenacity | Apache-2.0 | 10 | 20 | 3.2k |
| | kennethreitz/maya | MIT | 4 | 8 | 1.6k |
| | mahmoud/boltons | GPL-3.0 | 2 | 62 | 11.9k |
| | LKI/chinese-calendar | MIT | 1 | 16 | 2.3k |
| | ReactiveX/RxPY | MIT | 12 | 405 | 39.6k |
| | joke2k/django-environ | MIT | 16 | 20 | 2.4k |
| | python-validators/validators | MIT | 9 | 64 | 3.1k |
| | pytransitions/transitions | MIT | 1 | 53 | 10.1k |
| | coursera-dl/coursera-dl | LGPL-3.0 | 1 | 29 | 4.0k |
| | more-itertools/more-itertools | MIT | 3 | 9 | 9.9k |
| | agronholm/apscheduler | MIT | 1 | 66 | 9.6k |
| | arrow-py/arrow | Apache-2.0 | 7 | 20 | 14.8k |
| | aio-libs/aiocache | BSD-3-Clause | 1 | 50 | 4.6k |
| | PyFilesystem/pyfilesystem2 | MIT | 1 | 106 | 13.4k |
| Text Processing | mjpost/sacrebleu | Apache-2.0 | 1 | 42 | 5.2k |
| | vi3k6i5/flashtext | MIT | 3 | 18 | 1.0k |
| | google-research/arxiv-latex-cleaner | Apache-2.0 | 3 | 7 | 1.9k |
| | textstat/textstat | MIT | 22 | 123 | 3.2k |
| | carpedm20/emoji | GPL-3.0 | 7 | 23 | 3.1k |
| | summanlp/textrank | MIT | 4 | 20 | 4.2k |
| | nidhaloff/deep-translator | Apache-2.0 | 2 | 40 | 2.6k |
| | r1chardj0n3s/parse | MIT | 10 | 8 | 1.5k |
| | adbar/trafilatura | Apache-2.0 | 2 | 41 | 21.7k |
| | jsvine/markovify | MIT | 4 | 11 | 0.8k |
| | ssut/py-googletrans | MIT | 2 | 14 | 1.3k |
| | google/textfsm | Apache-2.0 | 1 | 10 | 3.3k |
| | google/budou | Apache-2.0 | 1 | 23 | 1.2k |
| | jaraco/inflect | Unknown | 2 | 17 | 4.8k |
| | pemistahl/lingua-py | Apache-2.0 | 2 | 5 | 0.9k |
| | pyparsing/pyparsing | MIT | 3 | 133 | 22.8k |
| | stanfordnlp/stanza | GPL-3.0 | 1 | 476 | 51.4k |
| | sloria/TextBlob | MIT | 1 | 37 | 4.0k |
| | seatgeek/fuzzywuzzy | GPL-2.0 | 5 | 11 | 1.0k |
| | amperser/proselint | BSD-3-Clause | 1 | 114 | 4.9k |
| | Python-Markdown/markdown | BSD-3-Clause | 5 | 67 | 11.7k |

| Type | Repository | License | #Instances | #Files | LoC |
|---|---|---|---|---|---|
| | pew-org/pew | MIT | 1 | 28 | 1.3k |
| | pschanely/CrossHair | GPL-3.0 | 7 | 163 | 32.0k |
| | dropbox/pyannotate | Apache-2.0 | 4 | 25 | 3.9k |
| | google/pinject | Apache-2.0 | 1 | 41 | 3.7k |
| | kapicorp/kapitan | Apache-2.0 | 3 | 86 | 9.3k |
| | python-injector/injector | BSD-3-Clause | 11 | 4 | 2.1k |
| | eliben/pyelftools | GPL-3.0 | 5 | 122 | 15.9k |
| | cookiecutter/cookiecutter | BSD-3-Clause | 16 | 88 | 6.2k |
| | nosarthur/gita | MIT | 10 | 11 | 2.2k |
| | PyCQA/isort | MIT | 7 | 97 | 18.7k |
| | SolidCode/SolidPython | Unknown | 1 | 35 | 4.1k |
| | facebookincubator/Bowler | MIT | 5 | 20 | 2.5k |
| | nschloe/tuna | GPL-3.0 | 2 | 12 | 0.6k |
| | dephell/dephell | MIT | 4 | 251 | 15.1k |
| | ekalinin/nodeenv | GPL-3.0 | 2 | 4 | 1.1k |
| | getsentry/sentry-python | MIT | 4 | 398 | 67.4k |
| | sqlalchemy/alembic | MIT | 6 | 101 | 39.5k |
| | olofk/fusesoc | BSD-2-Clause | 3 | 47 | 6.1k |
| | FactoryBoy/factory_boy | MIT | 2 | 54 | 7.4k |
| | pydoit/doit | MIT | 5 | 133 | 11.6k |
| | pre-commit/pre-commit-hooks | MIT | 5 | 69 | 3.6k |
| | PyCQA/pyflakes | MIT | 17 | 21 | 4.0k |
| | pre-commit/pre-commit | MIT | 9 | 129 | 13.6k |
| | facebookresearch/hydra | MIT | 5 | 308 | 32.5k |
| | Shpota/github-activity-generator | Apache-2.0 | 2 | 2 | 0.1k |
| | platformio/platformio-core | Apache-2.0 | 1 | 246 | 29.0k |
| | fastmonkeys/stellar | MIT | 1 | 11 | 0.8k |
| | zhanyong-wan/dongbei | MIT | 1 | 4 | 2.6k |
| | openapi-generators/openapi-python-client | MIT | 3 | 299 | 19.8k |
| | pythonprofilers/memory_profiler | GPL-3.0 | 2 | 34 | 2.2k |
| | mesonbuild/meson | Apache-2.0 | 4 | 500 | 87.2k |
| | trailofbits/graphtage | LGPL-3.0 | 1 | 51 | 7.3k |
| Software Development | koxudaxi/fastapi-code-generator | MIT | 6 | 50 | 1.8k |
| | nbQA-dev/nbQA | MIT | 15 | 60 | 3.8k |
| | breuleux/jurigged | MIT | 3 | 63 | 4.2k |
| | google/yapf | Apache-2.0 | 17 | 69 | 13.5k |
| | bndr/pipreqs | Apache-2.0 | 8 | 14 | 1.1k |
| | nickstenning/honcho | MIT | 11 | 34 | 2.1k |
| | cantools/cantools | MIT | 15 | 79 | 23.7k |
| | dabeaz/curio | GPL-3.0 | 3 | 96 | 10.2k |
| | terryyin/lizard | GPL-3.0 | 7 | 130 | 10.4k |
| | watson-developer-cloud/python-sdk | Apache-2.0 | 5 | 43 | 72.8k |
| | basetenlabs/truss | MIT | 16 | 287 | 36.8k |
| | rubik/radon | MIT | 1 | 29 | 3.8k |
| | guardrails-ai/guardrails | Apache-2.0 | 2 | 327 | 30.6k |
| | Pythagora-io/gpt-pilot | GPL-3.0 | 2 | 130 | 15.5k |
| | SWE-bench/SWE-bench | MIT | 1 | 84 | 13.3k |
| | noamgat/lm-format-enforcer | MIT | 1 | 25 | 2.9k |
| | spulec/freezegun | Apache-2.0 | 11 | 23 | 2.2k |
| | crewAIInc/crewAI-tools | MIT | 7 | 197 | 17.0k |
| | aws-ia/taskcat | Apache-2.0 | 1 | 85 | 10.8k |
| | enoch3712/ExtractThinker | Apache-2.0 | 1 | 110 | 12.0k |
| | weaveworks/grafanalib | Apache-2.0 | 15 | 37 | 6.9k |
| | Sceptre/sceptre | GPL-3.0 | 7 | 111 | 15.4k |
| | palantir/python-language-server | MIT | 1 | 62 | 5.1k |
| | nschloe/perfplot | GPL-3.0 | 9 | 7 | 0.6k |
| | qodo-ai/qodo-cover | AGPL-3.0 | 3 | 73 | 10.5k |
| | run-llama/llama_deploy | MIT | 1 | 94 | 5.9k |
| | asottile/pyupgrade | MIT | 14 | 97 | 10.6k |
| | firebase/firebase-admin-python | Apache-2.0 | 5 | 74 | 22.1k |
| | jendrikseipp/vulture | MIT | 6 | 44 | 2.9k |
| | autoscrape-labs/pydoll | MIT | 2 | 105 | 21.4k |
| | BerriAI/litellm | GPL-3.0 | 1 | 1975 | 417.1k |
| | jupyterhub/repo2docker | BSD-3-Clause | 8 | 100 | 8.1k |
| | googleapis/python-genai | Apache-2.0 | 4 | 190 | 55.9k |

| Type | Repository | License | #Instances | #Files | LoC |
|---|---|---|---|---|---|
| | kronenthaler/mod-pbxproj | MIT | 5 | 77 | 3.9k |
| | langchain-ai/langchain-mcp-adapters | MIT | 1 | 18 | 1.6k |
| | Instagram/MonkeyType | GPL-3.0 | 5 | 34 | 5.9k |
| | Delgan/loguru | MIT | 2 | 168 | 13.5k |
| | gitpython-developers/GitPython | BSD-3-Clause | 2 | 88 | 18.3k |
| | prospector-dev/prospector | GPL-2.0 | 1 | 107 | 5.1k |
| | hhatto/autopep8 | MIT | 7 | 65 | 10.2k |
| | wemake-services/wemake-python-styleguide | MIT | 2 | 378 | 30.1k |
| | fsspec/filesystem_spec | BSD-3-Clause | 17 | 104 | 24.3k |
| | python-rope/rope | LGPL-3.0 | 7 | 144 | 31.5k |
| | procrastinate-org/procrastinate | MIT | 3 | 169 | 13.9k |
| | itamarst/eliot | Apache-2.0 | 2 | 68 | 8.9k |
| | theskumar/python-dotenv | BSD-3-Clause | 3 | 18 | 2.1k |
| | koxudaxi/datamodel-code-generator | MIT | 20 | 531 | 41.6k |
| | coleifer/huey | MIT | 1 | 52 | 6.1k |
| | aws-powertools/powertools-lambda-python | MIT | 2 | 1091 | 65.3k |
| | nolar/kopf | MIT | 21 | 280 | 29.3k |
| | python-lsp/python-lsp-server | MIT | 2 | 70 | 8.0k |
| | pyocd/pyOCD | Apache-2.0 | 1 | 386 | 61.8k |
| | litestar-org/polyfactory | MIT | 3 | 141 | 11.2k |
| Software Development | getnikola/nikola | MIT | 1 | 226 | 23.2k |
| | cpplint/cpplint | GPL-3.0 | 6 | 3 | 8.5k |
| | faif/python-patterns | Unknown | 1 | 56 | 2.0k |
| | kevin1024/vcrpy | MIT | 2 | 69 | 6.4k |
| | conan-io/conan | MIT | 1 | 911 | 98.6k |
| | agronholm/typeguard | GPL-3.0 | 1 | 31 | 6.2k |
| | adrienverge/yamllint | GPL-3.0 | 22 | 68 | 11.5k |
| | PyCQA/flake8-bugbear | MIT | 2 | 70 | 4.4k |
| | eyurtsev/kor | MIT | 4 | 42 | 2.3k |
| | spotify/luigi | Apache-2.0 | 1 | 253 | 38.2k |
| | griptape-ai/griptape | Apache-2.0 | 4 | 1101 | 45.4k |
| | cloudtools/troposphere | BSD-2-Clause | 1 | 575 | 77.4k |
| | ethereum/py-evm | MIT | 1 | 408 | 41.9k |
| | getlogbook/logbook | GPL-3.0 | 1 | 69 | 6.7k |
| | joke2k/faker | MIT | 2 | 737 | 341.5k |
| | lark-parser/lark | MIT | 5 | 83 | 12.2k |
| | terraform-compliance/cli | MIT | 4 | 54 | 6.7k |
| | BeehiveInnovations/zen-mcp-server | GPL-3.0 | 2 | 194 | 38.8k |
| | pypa/pipenv | MIT | 1 | 643 | 129.0k |
| | Yelp/mrjob | GPL-3.0 | 5 | 218 | 46.4k |
| | getpelican/pelican | AGPL-3.0 | 10 | 50 | 12.8k |
| | SmileyChris/django-countries | MIT | 1 | 31 | 3.5k |
| | novnc/websockify | LGPL-3.0 | 2 | 19 | 3.4k |
| | mar10/wsgidav | MIT | 1 | 54 | 10.1k |
| | strawberry-graphql/strawberry | MIT | 11 | 572 | 58.8k |
| | Kludex/mangum | MIT | 1 | 21 | 3.5k |
| | exentriquesolutions/nip.io | GPL-3.0 | 1 | 5 | 1.1k |
| | Shopify/shopify_python_api | MIT | 12 | 156 | 4.3k |
| | googlemaps/google-maps-services-python | Apache-2.0 | 2 | 31 | 3.0k |
| | alexgolec/tda-api | MIT | 14 | 44 | 12.6k |
| | adamghill/django-unicorn | MIT | 11 | 122 | 8.6k |
| | Pylons/waitress | GPL-3.0 | 21 | 46 | 10.7k |
| Internet | requests-cache/requests-cache | BSD-2-Clause | 8 | 72 | 7.8k |
| | graphql-python/graphene | MIT | 16 | 124 | 8.4k |
| | deschler/django-modeltranslation | BSD-3-Clause | 2 | 35 | 7.4k |
| | zauberzeug/nicegui | MIT | 1 | 701 | 38.0k |
| | TransformerOptimus/SuperAGI | MIT | 7 | 398 | 21.6k |
| | scrapy/scrapyd | BSD-3-Clause | 6 | 56 | 3.3k |
| | koalalorenzo/python-digitalocean | LGPL-3.0 | 15 | 44 | 5.3k |
| | mwmbl/mwmbl | AGPL-3.0 | 6 | 128 | 15.0k |
| | ross/requests-futures | GPL-3.0 | 1 | 5 | 0.4k |
| | DedSecInside/TorBot | GPL-3.0 | 3 | 11 | 0.7k |
| | python-hyper/hyper | MIT | 1 | 46 | 7.2k |
| | psf/requests | Apache-2.0 | 1 | 35 | 7.1k |
| | Pylons/pyramid | GPL-3.0 | 1 | 330 | 45.5k |

| Type | Repository | License | #Instances | #Files | LoC |
|---|---|---|---|---|---|
| Internet | aiortc/aioquic | BSD-3-Clause | 3 | 58 | 20.0k |
| | MechanicalSoup/MechanicalSoup | MIT | 6 | 20 | 1.9k |
| | django-json-api/django-rest-framework-json-api | BSD-2-Clause | 2 | 77 | 9.0k |
| | michaelhly/solana-py | MIT | 8 | 51 | 6.8k |
| | rthalley/dnspython | GPL-3.0 | 3 | 241 | 36.5k |
| | django/channels | BSD-3-Clause | 2 | 45 | 3.2k |
| | miguelgrinberg/python-socketio | MIT | 3 | 95 | 12.5k |
| | django/asgiref | BSD-3-Clause | 1 | 20 | 2.6k |
| | django/daphne | BSD-3-Clause | 2 | 24 | 2.6k |
| | aws/chalice | Apache-2.0 | 1 | 110 | 37.7k |
| | maxmind/GeoIP2-python | Apache-2.0 | 1 | 13 | 2.3k |
| | scrapy/scrapy | BSD-3-Clause | 2 | 390 | 54.5k |
| | graphql-python/gql | MIT | 5 | 117 | 21.3k |
| Security | initstring/linkedin2username | MIT | 2 | 2 | 0.6k |
| | authlib/authlib | BSD-3-Clause | 1 | 289 | 27.2k |
| | casbin/pycasbin | Apache-2.0 | 13 | 85 | 7.8k |
| | a13xp0p0v/kernel-hardening-checker | GPL-3.0 | 1 | 5 | 2.0k |
| | quark-engine/quark-engine | GPL-3.0 | 1 | 76 | 10.3k |
| | obsidianforensics/hindsight | Apache-2.0 | 1 | 21 | 4.8k |
| | GitGuardian/ggshield | MIT | 3 | 250 | 22.7k |
| | salesforce/policy_sentry | MIT | 2 | 77 | 6.3k |
| | CTFd/CTFd | Apache-2.0 | 16 | 304 | 33.6k |
| | oauthlib/oauthlib | BSD-3-Clause | 17 | 137 | 11.7k |
| | jaraco/keyring | Unknown | 2 | 43 | 1.9k |
| | mozilla/bleach | GPL-3.0 | 3 | 53 | 14.6k |
| | linkedin/qark | GPL-3.0 | 5 | 68 | 3.1k |
| | mschwager/fierce | GPL-3.0 | 2 | 4 | 0.9k |
| | django-guardian/django-guardian | GPL-3.0 | 3 | 95 | 8.5k |
| | QIN2DIM/hcaptcha-challenger | GPL-3.0 | 1 | 105 | 8.9k |
| | python-security/pyt | GPL-2.0 | 1 | 248 | 9.2k |
| | yandex/gixy | GPL-3.0 | 2 | 48 | 4.5k |
| | petertodd/python-bitcoinlib | GPL-3.0 | 2 | 52 | 6.9k |
| | OWASP/Nettacker | Apache-2.0 | 1 | 63 | 6.7k |
| | jazzband/djangorestframework-simplejwt | MIT | 1 | 48 | 3.9k |
| | log2timeline/plaso | Apache-2.0 | 3 | 887 | 93.3k |
| | Yelp/detect-secrets | Apache-2.0 | 2 | 158 | 10.5k |
| | OWASP/pytm | GPL-3.0 | 1 | 10 | 3.6k |
| | pallets/itsdangerous | BSD-3-Clause | 3 | 14 | 1.0k |
| | mitre/caldera | Apache-2.0 | 2 | 169 | 16.5k |
| | prowler-cloud/prowler | Apache-2.0 | 4 | 2893 | 360.4k |
| | certtools/intelmq | AGPL-3.0 | 3 | 389 | 36.3k |
| | smicallef/spiderfoot | MIT | 7 | 694 | 57.7k |
| | androguard/androguard | Apache-2.0 | 3 | 66 | 25.2k |
| Database | dbcli/mssql-cli | BSD-3-Clause | 1 | 88 | 9.7k |
| | msiemens/tinydb | MIT | 8 | 20 | 2.1k |
| | datastax/python-driver | Apache-2.0 | 1 | 251 | 51.1k |
| | simonw/sqlite-utils | Apache-2.0 | 11 | 55 | 14.0k |
| | pynamodb/PynamoDB | MIT | 15 | 58 | 14.0k |
| | dbcli/pgcli | BSD-3-Clause | 1 | 66 | 9.4k |
| | duckdb/dbt-duckdb | Apache-2.0 | 18 | 84 | 6.2k |
| | qdrant/qdrant-client | Apache-2.0 | 15 | 178 | 62.9k |
| | Aiven-Open/pghoard | Apache-2.0 | 6 | 65 | 13.6k |
| | pgvector/pgvector-python | MIT | 5 | 72 | 3.8k |
| | jazzband/dj-database-url | BSD-3-Clause | 2 | 3 | 0.8k |
| | graphite-project/whisper | Apache-2.0 | 5 | 19 | 2.7k |
| | piskvorky/sqlitedict | Apache-2.0 | 2 | 11 | 0.8k |
| | kayak/pypika | Apache-2.0 | 42 | 59 | 11.3k |
| | datafold/data-diff | MIT | 1 | 70 | 13.5k |
| | EnterpriseDB/barman | GPL-3.0 | 9 | 99 | 65.9k |
| | reata/sqllineage | MIT | 1 | 92 | 7.0k |
| | aws/aws-sdk-pandas | Apache-2.0 | 2 | 222 | 43.2k |
| | activeloopai/deeplake | Apache-2.0 | 1 | 55 | 7.3k |
| | art049/odmantic | ISC | 2 | 156 | 8.7k |
| | georgia-tech-db/evadb | Apache-2.0 | 1 | 460 | 35.6k |
| | andialbrecht/sqlparse | BSD-3-Clause | 3 | 34 | 5.2k |

| Type | Repository | License | #Instances | #Files | LoC |
|---|---|---|---|---|---|
| Database | jazzband/django-redis | GPL-3.0 | 1 | 45 | 3.8k |
| | kvesteri/sqlalchemy-utils | GPL-3.0 | 1 | 169 | 13.0k |
| | pudo/dataset | MIT | 1 | 10 | 1.4k |
| | RDFLib/rdflib | BSD-3-Clause | 2 | 411 | 57.1k |
| | coleifer/peewee | MIT | 7 | 87 | 37.8k |
| System | Supervisor/supervisor | GPL-3.0 | 4 | 69 | 28.7k |
| | alichtman/shallow-backup | MIT | 2 | 20 | 1.9k |
| | jupyterhub/the-littlest-jupyterhub | BSD-3-Clause | 3 | 44 | 3.9k |
| | containers/podman-compose | GPL-2.0 | 3 | 82 | 11.5k |
| | pyinfra-dev/pyinfra | MIT | 4 | 220 | 21.1k |
| | borgmatic-collective/borgmatic | GPL-3.0 | 15 | 274 | 53.6k |
| | andreafrancia/trash-cli | GPL-2.0 | 3 | 336 | 9.7k |
| | jertel/elastalert2 | Apache-2.0 | 106 | 123 | 33.2k |
| | circus-tent/circus | GPL-3.0 | 2 | 132 | 12.2k |
| | patroni/patroni | MIT | 23 | 162 | 31.8k |
| | svinota/pyroute2 | GPL-3.0 | 2 | 407 | 49.6k |
| | docker/docker-py | Apache-2.0 | 1 | 130 | 21.1k |
| | gpustack/gpustack | Apache-2.0 | 4 | 81 | 87.5k |
| | pexpect/pexpect | GPL-3.0 | 2 | 100 | 8.2k |
| | liquidctl/liquidctl | GPL-3.0 | 9 | 76 | 13.7k |
| | giampaolo/psutil | BSD-3-Clause | 2 | 74 | 19.4k |
| | facebookincubator/submitit | MIT | 1 | 78 | 3.7k |
| | cloud-custodian/cloud-custodian | Apache-2.0 | 3 | 1017 | 182.1k |
| | tomerfiliba/plumbum | MIT | 1 | 78 | 9.9k |
| | canonical/cloud-init | Apache 2.0 | 7 | 571 | 132.4k |
| | overhangio/tutor | AGPL-3.0 | 1 | 81 | 7.1k |
| Communications | martinrusev/imbox | MIT | 12 | 20 | 0.7k |
| | websocket-client/websocket-client | Apache-2.0 | 4 | 28 | 4.2k |
| | hbldh/bleak | MIT | 1 | 67 | 10.6k |
| | wee-slack/wee-slack | MIT | 1 | 92 | 19.6k |
| | ktbyers/netmiko | MIT | 1 | 313 | 18.7k |
| | crossbario/autobahn-python | MIT | 1 | 352 | 40.0k |
| | scrapinghub/slackbot | MIT | 2 | 20 | 1.6k |
| | FreeOpcUa/python-opcua | LGPL-3.0 | 2 | 132 | 216.0k |
| | sendgrid/sendgrid-python | MIT | 1 | 125 | 8.8k |
| | taskiq-python/taskiq | MIT | 8 | 139 | 6.8k |
| | bear/python-twitter | Apache-2.0 | 7 | 39 | 6.9k |
| | LonamiWebs/Telethon | MIT | 3 | 141 | 16.1k |
| | jookies/jasmin | Apache 2.0 | 2 | 169 | 29.7k |
| | sshuttle/sshuttle | LGPL-2.1 | 9 | 34 | 6.2k |
| | nats-io/nats.py | Apache-2.0 | 1 | 62 | 14.8k |
| | slackapi/python-slack-sdk | MIT | 11 | 411 | 52.0k |
| | Forethought-Technologies/AutoChain | MIT | 2 | 64 | 3.2k |
| | FreeOpcUa/opcua-asyncio | LGPL-3.0 | 5 | 196 | 317.3k |
| | pinecone-io/canopy | Apache-2.0 | 2 | 136 | 10.1k |
| | celery/kombu | BSD-3-Clause | 2 | 167 | 28.5k |
| | zeromq/pyzmq | BSD-3-Clause | 1 | 172 | 15.4k |
| | letta-ai/letta | Apache-2.0 | 5 | 649 | 100.9k |
| | element-hq/synapse | AGPL-3.0 | 3 | 931 | 255.8k |
| File Formats | lidatong/dataclasses-json | MIT | 14 | 38 | 3.7k |
| | JelteF/PyLaTeX | MIT | 3 | 65 | 4.8k |
| | wireservice/csvkit | MIT | 4 | 44 | 4.9k |
| | alan-turing-institute/CleverCSV | MIT | 2 | 69 | 7.4k |
| | msgpack/msgpack-python | Apache 2.0 | 25 | 24 | 2.1k |
| | lincolnloop/python-qrcode | BSD 3-Clause License | 1 | 35 | 3.0k |
| | pdfminer/pdfminer.six | MIT | 6 | 60 | 20.2k |
| | jcrist/msgspec | BSD-3-Clause | 4 | 56 | 20.3k |
| | scrapy/parsel | BSD-3-Clause | 2 | 14 | 2.0k |
| | pmaupin/pdfrw | GPL-3.0 | 1 | 51 | 3.7k |
| | sripathikrishnan/redis-rdb-tools | MIT | 1 | 21 | 2.8k |
| | globocom/m3u8 | GPL-3.0 | 46 | 20 | 4.6k |
| | martinblech/xmltodict | MIT | 9 | 3 | 1.1k |
| | jsvine/pdfplumber | MIT | 2 | 38 | 6.6k |
| | chezou/tabula-py | MIT | 7 | 13 | 1.3k |
| | mangiucugna/json_repair | MIT | 11 | 26 | 2.0k |

| Type | Repository | License | #Instances | #Files | LoC |
|---|---|---|---|---|---|
| File Formats | landing-ai/agentic-doc | Apache-2.0 | 1 | 13 | 6.0k |
| | oomol-lab/pdf-craft | AGPL-3.0 | 1 | 81 | 6.4k |
| Data Analysis | raphaelvallat/pingouin | GPL-3.0 | 2 | 40 | 7.7k |
| | feature-engine/feature_engine | BSD-3-Clause | 58 | 228 | 26.7k |
| | movingpandas/movingpandas | BSD-3-Clause | 23 | 36 | 7.2k |
| | xflr6/graphviz | MIT | 1 | 75 | 3.7k |
| | petl-developers/petl | MIT | 2 | 168 | 22.4k |
| | pydata/pandas-datareader | GPL-3.0 | 1 | 76 | 6.8k |
| | graphistry/pygraphistry | BSD-3-Clause | 1 | 195 | 37.0k |
| | sinaptik-ai/pandas-ai | GPL-3.0 | 24 | 174 | 13.9k |
| | peerchemist/finta | LGPL-3.0 | 2 | 7 | 2.0k |
| | electricitymaps/electricitymaps-contrib | AGPL-3.0 | 2 | 261 | 34.8k |
| | taynaud/python-louvain | BSD-3-Clause | 1 | 7 | 0.6k |
| | ourownstory/neural_prophet | MIT | 11 | 66 | 14.1k |
| | nteract/papermill | BSD-3-Clause | 3 | 38 | 4.9k |
| | microsoft/TaskWeaver | MIT | 13 | 147 | 15.6k |
| | pyecharts/pyecharts | MIT | 1 | 128 | 14.4k |
| | jldbc/pybaseball | MIT | 2 | 130 | 7.6k |
| | intake/intake | BSD-2-Clause | 4 | 99 | 11.0k |
| | unionai-oss/pandera | MIT | 3 | 249 | 47.4k |
| | py-why/dowhy | MIT | 3 | 229 | 30.3k |
| | lmfit/lmfit-py | GPL-3.0 | 4 | 105 | 12.5k |
| | bashtage/arch | GPL-3.0 | 2 | 100 | 28.7k |
| | lux-org/lux | Apache-2.0 | 1 | 91 | 10.0k |
| | rasbt/mlxtend | GPL-3.0 | 16 | 210 | 19.5k |
| | fugue-project/fugue | Apache-2.0 | 1 | 223 | 29.1k |
| | unit8co/darts | Apache-2.0 | 2 | 239 | 81.0k |
| | python-streamz/streamz | BSD-3-Clause | 2 | 38 | 6.7k |
| | holoviz/datashader | BSD-3-Clause | 1 | 106 | 26.9k |
| | vispy/vispy | GPL-3.0 | 4 | 557 | 55.6k |
| | dlt-hub/dlt | Apache-2.0 | 11 | 921 | 147.0k |
| | python-visualization/folium | GPL-3.0 | 1 | 101 | 7.4k |
| Terminals | nbedos/termtosvg | BSD-3-Clause | 2 | 13 | 1.9k |
| | bpython/bpython | GPL-3.0 | 9 | 75 | 11.7k |
| | hauntsaninja/pyp | MIT | 8 | 3 | 1.1k |
| | jorgebastida/awslogs | GPL-3.0 | 1 | 9 | 1.1k |
| | rsalmei/alive-progress | MIT | 2 | 41 | 3.0k |
| | jazzband/Watson | MIT | 1 | 21 | 3.6k |
| | kellyjonbrazil/jc | MIT | 17 | 526 | 44.9k |
| | bee-san/pyWhat | MIT | 2 | 19 | 1.8k |
| | nvbn/thefuck | MIT | 28 | 406 | 10.5k |
| | peterbrittain/asciimatics | Apache-2.0 | 1 | 105 | 16.3k |
| | httpie/http-prompt | MIT | 2 | 33 | 4.4k |
| | jarun/buku | GPL-3.0 | 1 | 27 | 8.9k |
| | mkaz/termgraph | MIT | 14 | 18 | 1.4k |
| | gptme/gptme | MIT | 2 | 164 | 24.8k |
| | laixintao/flameshow | MIT | 3 | 35 | 3.0k |
| | online-judge-tools/oj | MIT | 10 | 32 | 5.4k |
| | python-poetry/cleo | MIT | 2 | 108 | 7.9k |
| | pimutils/khal | MIT | 7 | 57 | 12.6k |
| | LazoVelko/Pokemon-Terminal | GPL-3.0 | 1 | 38 | 1.5k |
| | robusta-dev/holmesgpt | MIT | 3 | 338 | 53.0k |
| | tartley/colorama | BSD-3-Clause | 1 | 23 | 1.4k |
| | tmbo/questionary | MIT | 6 | 59 | 4.1k |
| | docopt/docopt | MIT | 5 | 23 | 1.1k |
| | dylanaraps/pywal | MIT | 7 | 26 | 1.2k |
| | Textualize/rich | MIT | 5 | 188 | 29.3k |
| | insanum/gcalcli | MIT | 4 | 34 | 4.5k |
| Other/Nonlisted Topic | KMKfw/kmk_firmware | GPL-3.0 | 6 | 289 | 20.4k |
| Office/Business | sec-edgar/sec-edgar | Apache-2.0 | 1 | 31 | 3.9k |
| | mlouielu/twstock | MIT | 2 | 23 | 1.3k |
| | beancount/beancount | GPL-2.0 | 7 | 201 | 24.4k |
| | burnash/gspread | MIT | 2 | 17 | 5.7k |
| | jmcnamara/XlsxWriter | BSD-2-Clause | 9 | 1394 | 55.4k |
| | mintapi/mintapi | MIT | 2 | 21 | 3.1k |

| Type | Repository | License | #Instances | #Files | LoC |
|---|---|---|---|---|---|
| Office/Business | almarklein/timetagger | GPL-3.0 | 3 | 28 | 10.5k |
| | brndnmtthws/thetagang | AGPL-3.0 | 2 | 28 | 6.6k |
| | andreroggeri/pynubank | MIT | 5 | 23 | 1.5k |
| Education | cosmicpython/code | GPL-3.0 | 1 | 29 | 1.1k |
| Unknown | fengsp/plan | GPL-3.0 | 1 | 22 | 1.2k |
| | jiaaro/pydub | MIT | 2 | 13 | 3.2k |
| Documentation | scanapi/scanapi | MIT | 1 | 67 | 4.4k |
| | mitmproxy/pdoc | MIT | 6 | 78 | 7.0k |
| Multimedia | Breakthrough/PySceneDetect | BSD-3-Clause | 1 | 49 | 8.3k |
| | quodlibet/mutagen | GPL-2.0 | 3 | 108 | 22.7k |
| | SickChill/sickchill | GPL-3.0 | 1 | 310 | 45.7k |
| | beetbox/beets | MIT | 13 | 212 | 45.7k |
| | mido/mido | MIT | 4 | 74 | 4.0k |
| | ytdl-org/youtube-dl | Unlicense | 1 | 902 | 137.5k |
| | brycedrennan/imaginAIry | MIT | 1 | 337 | 49.5k |
| | pytube/pytube | Unlicense | 4 | 39 | 5.2k |
| | lhotse-speech/lhotse | Apache-2.0 | 1 | 446 | 64.3k |
| | Zulko/moviepy | MIT | 10 | 137 | 11.1k |
| | SYSTRAN/faster-whisper | MIT | 2 | 19 | 3.0k |
| | imageio/imageio | BSD-2-Clause | 3 | 98 | 26.5k |
| | spotDL/spotify-downloader | MIT | 1 | 82 | 10.0k |
| Home Automation | jasonacox/tinytuya | MIT | 1 | 67 | 10.5k |
| | gpiozero/gpiozero | GPL-3.0 | 8 | 148 | 15.0k |
| | SoCo/SoCo | MIT | 4 | 65 | 11.9k |
| | thingsboard/thingsboard-gateway | Apache-2.0 | 1 | 217 | 30.1k |
| Games/Entertainment | niklasf/python-chess | GPL-3.0 | 6 | 23 | 14.3k |
| Desktop Environment | sharkwouter/minigalaxy | GPL-3.0 | 9 | 47 | 6.5k |

