# OpenReview forum: "TestExplora: Can LLMs Write Tests to Find Potential Problems Existing in Repository?"
_ICLR.cc/2026/Conference — Submitted to ICLR 2026_

### Official Review · Reviewer_aTns · 2025-10-30

**Soundness:** 3
**Presentation:** 3
**Contribution:** 2
**Rating:** 4
**Confidence:** 4

**Summary:**

The paper introduces TestExplora, a new test case generation benchmark for proactive, exploratory testing at repository level. The authors construct the benchmark by taking a valid PR (after careful filtering and selection process) and extract 3 pieces of information: documentation about the intended functionality, entry point, and their dependencies. The overall task is to let the LLMs generate test cases for a piece of function given larger (repo-level) context without access to information pointing to bugs (e.g., issues, fixing commits).

The authors evaluate the LLMs with several metrics capturing different scopes, under two scenarios: white-box and black-box, being different in terms of whether the actual code is visible to LLMs or not. The paper's key finding is that even state-of-the-art models perform very poorly on this task, with a low success rate (e.g., 17.56% for GPT-5-mini). The analysis suggests models primarily fail due to assertion mismatches and misconfigured mocks, showing that there still exist a significant gap toward finding defects proactively.

**Strengths:**

* A solid contribution of large amount of highly curated data instances & PRs
* Good findings about the SOTA LLMs not being robust wrt finding the potential bugs without access to the actual errors. Proactively assessing the bug attracts interest in real-life cases where running the test suites are expensive. So I think it is a nice direction to pursue.

**Weaknesses:**

* The novelty is limited. SWT-bench is a repository-level test generation benchmark, but this is given the bug report. It sounds to me that the proposed benchmark only differs conceptually from SWT-bench in that the input is intended behavior instead of the bug report. Though still nice to see the limitations of SOTA models, the claimed novelty is somewhat incremental.
* Somewhat shallow analysis. Experimental settings are great, but the analysis doesn't fully dissect the results. For example, whitebox and blackbox testing show similar F2P scores. I understand both settings have similar error patterns, but what parts are different? If not so different, does that mean the models are simply not utilizing the provided context?
* DocAgent provides better documentation – is it longer, more structured, comprehensive? It might be expected if Human-written versions tend to have only brief documentations. If we want to simulate a real-world setting, should we modulate the DocAgent output to mimic human-written documentations?

—

Other notes:

* Sec 4.2 starts with a sentence repetition.
* TestExplora-Lite is only briefly mentioned in Sec 4.3. It'd be nice to define it if it also appears in the main results table.

**Questions:**

Please see above.

---

> ### Author Response · Authors · 2025-11-20
> **Response to reviewer aTns (Part I)**
>
> We thank the reviewer for the positive assessment of our dataset curation and experimental settings, as well as the constructive feedback regarding the novelty and analysis depth. We address your concerns point-by-point below.
>
> ## Novelty and Comparison to SWT-Bench
>
> > W1: The novelty is limited... the proposed benchmark only differs conceptually from SWT-bench in that the input is intended behavior instead of the bug report.
> >
>
> We respectfully clarify that TestExplora and SWT-Bench represent complementary, rather than conflicting, lines of research. While SWT-Bench focuses on Fault Verification (reproducing existing, known issues to support regression testing), TestExplora targets Fault Discovery (identifying potential, unknown issues proactively).
>
> To clearly delineate our contribution, we categorize the landscape of issue-related test generation below based on Earl T. Barr et al. [1]. We view TestExplora and SWT-Bench as addressing distinct branches of test generation. SWT-Bench operates in the "Bug Known" domain, serving as a benchmark for Issue Reproduction and simulating the post-reporting workflow. In contrast, TestExplora targets the "Bug Unknown" domain. Unlike fuzzing (implicit oracles) or regression testing (derived oracles), TestExplora establishes a benchmark for Specified Oracles (Functional Correctness):
>
> ```
> - Test Generation Scenarios
>   |
>   |-- A. Bug Known (Goal: Reproduction)
>   |      |
>   |      |-- Issue Reproduction
>   |             -> Focus: Replicability (Triggering the specific reported failure)
>   |             -> Methods: Automated Issue Reproduction (e.g., SWT-Bench)
>   |
>   |-- B. Bug Unknown (Goal: Discovery)
>          |
>          |-- 1. Rely on Implicit Oracle (Generic Reliability)
>          |      -> Focus: Robustness (Crashes, Memory Errors)
>          |      -> Methods: Coverage-guided Fuzzing (e.g., AFL, libFuzzer, Pynguin)
>          |
>          |-- 2. Rely on Derived Oracle (Behavioral Consistency)
>          |      -> Focus: Regression (Unintended Changes)
>          |      -> Methods: Differential Testing, Regression Suites (e.g., TestTora)
>          |
>          |-- 3. Rely on Specified Oracles (Functional Correctness)  <== TestExplora
>                 -> Focus: Logic Verification (Does code match intent?)
>                 -> Methods: TestExplora (Semantic-aware generation)
> ```
>
> More broadly, TestExplora targets the pre-reporting stage of development, simulating how developers uncover bugs through usage. SWT-Bench, by contrast, operates in the post-reporting stage, simulating the task of reproducing an issue that has already been documented.

---

> ### Author Response · Authors · 2025-11-20
> **Response to reviewer aTns (Part II)**
>
> ## Analysis of White-box vs. Black-box and Model Context
>
> > W2: Somewhat shallow analysis... whitebox and blackbox testing show similar F2P scores. I understand both settings have similar error patterns, but what parts are different? If not so different, does that mean the models are simply not utilizing the provided context?
> >
>
> Your observation is accurate. The similarity in performance arises because the models struggle to effectively utilize the additional context provided in the White-box setting—specifically, the code dependencies. As discussed in our Dependency Analysis (Appendix G), some models lack the robust reasoning required to leverage these dependencies fully. For instance, GPT-4o performs best (3.9 F2P) when provided only with the code invoked by the test entry function, whereas GPT-5-mini achieves its peak performance (12.8 F2P) when given the full set of dependencies.
>
> The observation that different models exhibit preferences over dependency scopes motivated us to include the following agent-based experiments. Specifically, we deployed both SWE-Agent and Trae-Agent:
> | Model      | Context    | HP | F2P   | EC | CFG | Num.  |
> |------------|-----------|-----------|-------|-------------|--------------|-------|
> | o4-mini    | White Box  | 72.33     | 5.92  | 67.30       | 47.06        | 10.48 |
> |     | SWE-Agent  | 82.03     | 9.42  | **69.63**       | 47.81        | 13.79 |
> |     | Trae-Agent | **86.55**     | **12.16** | 62.01       | **47.95**       | 12.76 |
> | gpt-5-mini | White Box  | 77.68     | 12.79 | **67.76**       | 45.66        | 7.69  |
> |     | SWE-Agent  | **93.81**     | **17.27** | 65.45       | **47.43**        | 23.09 |
> |     | Trae-Agent | 88.50     | 14.55 | 62.31       | 46.84       | 12.79 |
>
> The significant improvement in the Agent Setting highlights that models utilize context more effectively when they can actively explore it. This establishes agent-driven exploratory test generation as a promising direction for TestExplora. Furthermore, we evaluated the pass@k metrics for SWE-Agent, indicating substantial room for improvement in current agents:
>
>
> | Model     | pass@1    | pass@2 | pass@3   | pass@4 | pass@5 |
> |------------|-----------|-----------|-------|-------------|--------------|
> |SWE-agent (o4-mini)|9.4|14.8|17.0|20.0|21.2|
> |SWE-agent (gpt-5-mini)|17.3|23.6|26.4|28.2|29.7|
>
> **Importantly, these extended agent-based evaluations are part of our continuous research roadmap. We view TestExplora as a long-term foundation for automated testing research and are committed to its sustained development, ensuring deep and rigorous investigation rather than shallow exploration.**

---

> ### Author Response · Authors · 2025-11-20
> **Response to reviewer aTns (Part III)**
>
> ## DocAgent and Documentation Quality
>
> > W3: DocAgent provides better documentation... If we want to simulate a real-world setting, should we modulate the DocAgent output to mimic human-written documentations?
> >
>
> You are correct that DocAgent generates more detailed and structured documentation than typical human-written stubs. However, we justify this design choice on two grounds:
> * **Necessity for Verification**: Overly simplistic documentation often lacks the behavioral specification required for a model to distinguish between intended logic and a bug. Detailed documentation is a prerequisite for resolving "Fail-to-Pass" scenarios effectively.
> * **Simulating High-Quality Standards**: Well-maintained repositories (e.g., Scikit-Learn) enforce high documentation standards. Moreover, as LLM-based documentation tools (e.g., RepoAgent [2], GitHub Copilot [3], Sourcegraph Cody [4]) become standard, we believe TestExplora’s synthetic documentation accurately reflects the future of "real-world" development environments.
>
> To verify that DocAgent improves performance via better specification rather than information leakage, we conducted an ablation study. We compared the standard SWE-Agent against a version where the ground-truth patch was "leaked" into the context (simulating perfect knowledge vs. derived knowledge):
> | Model      | Method    | HP | F2P   | EC | CFG | Num.  |
> |------------|-----------|-----------|-------|-------------|--------------|-------|
> | gpt-5-mini | SWEAgent  | 93.81     | 17.27 | 65.45       | 47.43        | 23.09 |
> |  | SWEAgent (patch leakage)  | 98.26     | 73.94 | 59.13       | 53.73        | 34.33 |
>
> The vast gap between the standard setup (17.27 F2P) and the leakage setup (73.94 F2P) confirms that DocAgent does not trivialize the task. The challenge remains reasoning about the code to find defects.
>
> To rigorously verify whether there is any leakage, we conducted a annotation of 1,552 PRs using GPT-5 and a manual review of 200 random samples by an external senior engineer:
>
> | Checker | Total Instance | Patch/bug Leakage |
> | ------- | -------------- | ----------------- |
> | GPT-5   | 1,552          | 16                |
> | Human   | 200            | 0                 |
>
> The results confirm negligible leakage.
>
> ## Typo
>
> > W4 & 5: TestExplora-Lite is only briefly mentioned in Sec 4.3. It'd be nice to define it if it also appears in the main results table. & Sec 4.2 starts with a sentence repetition.
> >
>
> Thank you for your careful observation; we have addressed and corrected this issue in the revised version of the paper.
>
> $Ref$:
>
> [1] The Oracle Problem in Software Testing: A Survey, TSE 2015.
>
> [2] RepoAgent: An LLM-Powered Open-Source Framework for Repository-level Code Documentation Generation, Arxiv 2024.
>
> [3] https://github.com/features/copilot
>
> [4] https://sourcegraph.com/docs/cody

---

> ### Author Response · Authors · 2025-11-25
>
> Dear Reviewer aTns,
>
> We are grateful for your insightful comments and the time you have dedicated to reviewing our work. With about one week remaining in the discussion phase, we wanted to check in and confirm that our recent clarifications have addressed the issues you previously noted.
>
> If any additional information is needed, please feel free to let us know—we would be delighted to assist immediately.
>
> Thank you very much for your continued consideration.
>
> Sincerely, Authors

---

### Official Review · Reviewer_qmdw · 2025-10-31

**Soundness:** 3
**Presentation:** 4
**Contribution:** 4
**Rating:** 6
**Confidence:** 4

**Summary:**

TestExplora is a new benchmark set to assess LLM’s abilities to proactively detect bugs through tests generated. The benchmark is composed of 2389 test generation tasks from 482 repositories. LLMs are provided test entry points along with their human and LLM generated documentation and prompted to generate tests. The performance of the LLMs are assessed using various metrics, notably with fail-to-pass rates which checks if generated tests exhibit a Fail-to-Pass transition with selected PRs. The best LLM in this task displays 17% success rate, demonstrating challenging aspects of this benchmark.

**Strengths:**

- The paper is well written and organized.
- The work is well motivated. Proactive detection of bugs through LLM-based test generation would be of interest to the field.
- The benchmark is a novel contribution motivating further advancements for coding performance of LLMs.

**Weaknesses:**

- My understanding is that the benchmark is only composed of python examples, which might be limiting.
- The evaluations are lacking error bars. As LLMs are not deterministic, I’d love to understand the variability in the results. Also lacking one of the strongest coding LLM from Claude line of models.

**Questions:**

- Curious to hear if the authors considered repeating the experiments where they provide a hint to the LLM that there is a bug and would love to see if it would help LLMs to be more critical and observe how the results change.
- In Table 3:
   - What is the “Num” column?
   - There are several cases where the results are lower in White Box cases as compared to Black Box cases (e.g. GPT4-o for EC and CFG). It felt counterintuitive to me, can you elaborate on this behavior?
- In the main discussion as well as Figure 2, it is indicated that Black box case does not include code implementation. However looking at the prompts in the appendix, I only see dependencies as the delta since test entry functions are included for both. Could you confirm what the delta is between White and Black Box?
- Typo in line 372: “TThese”

---

> ### Author Response · Authors · 2025-11-20
> **Response to reviewer qmdw (Part I)**
>
> We thank the reviewer for recognizing TestExplora as a novel contribution that motivates further advancements in LLM coding performance. We appreciate the constructive feedback regarding evaluation rigor and model diversity. Below, we address the specific weaknesses and questions raised. To ensure logical coherence, we reordered some reviewer questions.
>
> ## The benchmark is only composed of python examples, which might be limiting.
>
> > W1: My understanding is that the benchmark is only composed of python examples, which might be limiting.
> >
>
> Thank you for the suggestion. Our benchmark is built around Python primarily because it is currently one of the most widely used programming languages. Moreover, Python-based projects exhibit strong consistency, which makes it easier to construct a unified framework for automated data collection and evaluation. We also ensured diversity by collecting as many repositories as possible. Additionally, we plan to extend the benchmark to include multi-language projects in future work to further advance research on test generation.
>
> ## Evaluation Rigor and Missing SOTA Models
>
> > W2.1: The evaluations are lacking error bars. As LLMs are not deterministic, I’d love to understand the variability in the results.
> >
>
>
> We agree that understanding variability is crucial for non-deterministic LLMs. To address this, we conducted repeated runs on TestExplora-Lite for all models for three times. As shown in the table below, we report the mean and variance (in parentheses). The low variance indicates that the performance on TestExplora is relatively stable.
>
> |Context | Model | HP (var) | F2P (var) | EC (var) | CFG (var) | Num. (var) |
> |-------|-------|----------|-----------|----------|-----------|------------|
> | Black Box | QC-30B-A3B | 54.38 (4.83) | 2.30 (0.67) | 49.98 (4.40) | 42.21 (0.37) | 13.40 (0.05) |
> |  | o3-mini | 58.37 (6.90) | 5.58 (0.06) | 56.43 (7.50) | 44.64 (0.03) | 8.39 (0.05) |
> |  | o4-mini | 56.97 (7.92) | 3.74 (1.99) | 59.44 (0.72) | **46.29** (0.04) | 10.82 (0.00) |
> |  | Gemini 2.5-Pro | 58.74 (0.01) | 2.60 (0.43) | 53.24 (8.57) | 44.91 (0.64) | 12.89 (0.01) |
> |  | gpt-4o | 51.38 (7.00) | 3.61 (1.80) | 54.69 (0.64) | 43.55 (0.07) | 7.75 (0.01) |
> |  | GPT-5 mini | **60.99** (4.64) | **7.67** (0.02) | **59.72** (1.83) | 45.59 (0.70) | 7.96 (0.01) |
> |  White Box | QC-30B-A3B | 60.90 (21.74) | 2.09 (0.08) | 59.78 (2.84) | 44.05 (0.47) | 13.21 (0.13) |
> |  | o3-mini | 71.53 (10.45) | 7.07 (0.75) | 61.84 (4.93) | 43.78 (0.07) | 8.00 (0.04) |
> |  | o4-mini | 74.70 (5.64) | 6.45 (0.28) | **67.16** (0.02) | **46.32** (0.55) | 10.36 (0.01) |
> |  | Gemini 2.5-Pro | 74.91 (2.05) | 5.08 (0.11) | 61.47 (15.75) | 44.25 (0.32) | 12.84 (0.13) |
> |  | gpt-4o | 57.92 (13.15) | 2.20 (0.61) | 56.29 (6.19) | 44.15 (0.18) | 7.42 (0.05) |
> |  | GPT-5 mini | **80.25** (3.30) | **11.84** (1.20) | 65.78 (2.29) | 44.37 (0.84) | 7.78 (0.01) |
>
> Note: The HP of some models might be slightly higher. Therefore, we list the results from the three runs as follows:
>
> **o3-mini (White Box)** - HP values across runs:
>   - Run 1: 67.01
>   - Run 2: 73.20
>   - Run 3: 74.38
>
> **gpt-4o (White Box)** - HP values across runs:
>   - Run 1: 52.81
>   - Run 2: 60.10
>   - Run 3: 60.85
>
> **QC Coder (White Box)** - HP values across runs:
>   - Run 1: 54.35
>   - Run 2: 63.52
>   - Run 3: 64.83
>
> > W2.2: Also lacking one of the strongest coding LLM from Claude line of models.
> >
>
> Furthermore, to address the concern regarding model coverage, we evaluated the latest SOTA models, including Claude-4.5-Sonnet, GPT-5, and DeepSeek-R1.
>
> | Model      | Context    | HP | F2P   | EC | CFG | Num.  |
> |------------|-----------|-----------|-------|-------------|--------------|-------|
> |Claude-Sonnet-4.5| White Box  | 87.52     | 16.06  | 71.96       | 47.37        | 23.94 |
> | GPT-5 | White Box  | 84.88     | 15.15 | 69.44    | 45.59        | 9.59  |
> |DeepSeek-R1| White Box  | 63.72     | 3.03  | 47.16       | 34.77        | 10.22 |
>
> These results confirm that TestExplora remains a challenging benchmark with significant room for improvement, even for the most advanced models.

---

> ### Author Response · Authors · 2025-11-20
> **Response to reviewer qmdw (Part II)**
>
> ## White Box vs. Black Box Performance & Agent-Based Exploration
>
> > Q2.2: There are several cases where the results are lower in White Box cases as compared to Black Box cases (e.g. GPT4-o for EC and CFG). It felt counterintuitive to me, can you elaborate on this behavior?
> >
>
> This performance drop is attributable to the challenges associated with dependency handling in the White Box setting. We observed a similar trend during our analysis and have provided a full breakdown of how dependencies impact generation in Appendix G: Dependency Analysis. The primary reason is that different models exhibit distinct sensitivities to context overload. For instance, GPT-4o's F2P rate improves from 2.3 (Black Box) to 3.9 when limited to invocation dependencies, whereas GPT-5-mini thrives with full context (12.8 F2P).
>
> The observation that different models exhibit preferences over dependency scopes motivated us to include the following agent-based experiments. **It is worth noting that this extension is part of our continuous research roadmap. We are dedicated to establishing TestExplora as a long-term project to drive sustained progress in the field.** Specifically, we deployed both SWE-agent and Trae-agent:
>
> | Model      | Context    | HP | F2P   | EC | CFG | Num.  |
> |------------|-----------|-----------|-------|-------------|--------------|-------|
> | o4-mini    | White Box  | 72.33     | 5.92  | 67.30       | 47.06        | 10.48 |
> |     | SWE-Agent  | 82.03     | 9.42  | **69.63**       | 47.81        | 13.79 |
> |     | Trae-Agent | **86.55**     | **12.16** | 62.01       | **47.95**       | 12.76 |
> | gpt-5-mini | White Box  | 77.68     | 12.79 | **67.76**       | 45.66        | 7.69  |
> |     | SWE-Agent  | **93.81**     | **17.27** | 65.45       | **47.43**        | 23.09 |
> |     | Trae-Agent | 88.50     | 14.55 | 62.31       | 46.84       | 12.79 |
>
> In the Agent Setting, we provide only the name of function to be tested along with its corresponding documentation. The results show that when the model is able to autonomously explore the repository, its performance improves significantly. Also the F2P pass@k metrics for SWE-agent improves as k increases, though there is still room for further improvement:
>
> | Model     | pass@1    | pass@2 | pass@3   | pass@4 | pass@5 |
> |------------|-----------|-----------|-------|-------------|--------------|
> |SWE-agent (o4-mini)|9.4|14.8|17.0|20.0|21.2|
> |SWE-agent (gpt-5-mini)|17.3|23.6|26.4|28.2|29.7|
>
> These results suggest that agent-driven exploratory test generation is a promising future direction for TestExplora.
>
> ## Critical Reasoning and "Bug Hints"
>
> > Q1: Curious to hear if the authors considered repeating the experiments where they provide a hint to the LLM that there is a bug and would love to see if it would help LLMs to be more critical and observe how the results change.
> >
>
> This is an excellent suggestion. To test this, we ran an experiment where we explicitly provided the "fix" (patch leakage) to the model via SWE-Agent, effectively hinting at the exact location and nature of the bug.
>
>
> | Model      | Method    | HP | F2P   | EC | CFG | Num.  |
> |------------|-----------|-----------|-------|-------------|--------------|-------|
> | gpt-5-mini | SWEAgent  | 93.81     | 17.27 | 65.45       | 47.43        | 23.09 |
> |  | SWEAgent (patch leakage)  | 98.26     | 73.94 | 59.13       | 53.73        | 34.33 |
>
> The dramatic increase in F2P (17.27 $\rightarrow$ 73.94) confirms that the model has the capability to write the test but lacks the critical reasoning to identify the bug without assistance. To investigate this "critical reasoning gap" further, we designed a binary classification task: distinguishing between correct and incorrect code implementations. The accuracies were: 0.508 (o4-mini), 0.596 (gpt-5-mini), and 0.692 (gpt-5). This confirms that current models struggle to critically evaluate code correctness.
>
> Finally, we analyzed the tool-use distribution of the agents. The results show that stronger models (GPT-5-mini) rely more on execution (Python/pytest) compared to weaker models:
>
> | Tool       | view | submit | pytest | python | grep -R |
> | ---------- | ----------------------- | ------ | ------ | ------ | ------- |
> | GPT-5-mini |56.0%|7.3%|2.4%|2.0%|2.0%|
> | o4-mini |48.4%|8.1% |<0.2%|1.2%|11.5%|
>
> This reinforces the hypothesis that future research should focus on execution-based feedback loops to mimic human-like critical debugging. We have also updated detailed results in the revised version of the paper (Figure 6 and Figure 11 in the reversed version).

---

> ### Author Response · Authors · 2025-11-20
> **Response to reviewer qmdw (Part III)**
>
> ## Clarifications
>
> > Q3: In the main discussion as well as Figure 2 ... Could you confirm what the delta is between White and Black Box?
> >
>
> We clarify the distinction regarding Test Entry Functions. In the White Box setting, the full implementation is provided. In the Black Box setting, the function body is masked to simulate a scenario where developers write tests against an API without seeing the internal logic.
>
> For example, consider the following test entry function:
>
> ```python
> def test_entry(input: int):
>     input = helper_function(input)
>     return input * 2
> ```
> with documentation:
> ```
>     Process an integer value through `helper_function` and return the doubled result.
>
>     This function first passes the input integer to `helper_function` for preprocessing
>     or transformation. After receiving the processed value, it multiplies the result by 2
>     and returns it.
>
>     Parameters
>     ----------
>     input : int
>         The integer value to be processed.
>
>     Returns
>     -------
>     int
>         The processed integer after being passed through `helper_function`
>         and multiplied by 2.
> ```
>
> In the White-Box setting, the model sees the full implementation with documentation:
>
> ```python
> def test_entry(input: int):
>     input = helper_function(input)
>     return input * 2
> ```
>
> In the Black-Box setting, however, the implementation is hidden but the documentation is given:
>
> ```python
> def test_entry(input: int):
>     # --code omitted here--
> ```
>
> This setup mimics the realistic Black-Box scenario in which developers can only interact with the code through its API when writing tests.
>
> ## Typo
>
> > Q2.1 & 4: What is the “Num” column? & Typo in line 372: “TThese”
> >
>
> We have corrected the typo "TThese". The "Num" column refers to the average Number of test cases generated by the model per task. We have updated the caption in Table 3 to reflect this clearly.

---

> ### Author Response · Authors · 2025-11-25
>
> Dear Reviewer qmdw,
>
> Thank you once again for your constructive feedback. As the discussion deadline is approaching in roughly one week, we kindly wish to ensure that our previous explanations have resolved your concerns.
>
> If any further details would be helpful, we stand ready to offer them without delay.
>
> We truly appreciate your support and attention.
>
> Sincerely, Authors

---

### Official Review · Reviewer_CycK · 2025-10-31

**Soundness:** 2
**Presentation:** 2
**Contribution:** 3
**Rating:** 4
**Confidence:** 4

**Summary:**

This paper introduces **TestExplora**, a benchmark targeting *proactive* test generation for real-world repositories without giving models explicit bug cues. Tasks are derived from GitHub PRs (2,389 tasks, 482 repos), and the benchmark enforces fail-to-pass behavior: the generated test must fail on buggy code and pass after patching. The benchmark evaluates both black-box and white-box modes and reports that state-of-the-art LLMs perform poorly (typically <13% F2P), suggesting current models struggle at autonomous exploratory testing. The authors also introduce DocAgent to synthesize/clean documentation for models.

**Strengths:**

The benchmark pipeline and fail-to-pass validation methodology are generally reasonable, with realistic environment setup. However:

- No statistical significance or variance reporting
- Potential leakage: PRs might be in model training data
- DocAgent-generated docs may introduce *task supervision*, contaminating “exploratory” nature
- Some pipeline heuristics lack ablation (entry-point selection, doc filtering)
- Limited robustness checks (e.g., determinism, flaky tests, environment noise)

**Weaknesses:**

#### 1. Benchmark purity & leakage concerns
- PRs and project history may be in model training sets
- No explicit contamination filtering (hashing, repo disambiguation)
- DocAgent can implicitly encode bug semantics → *post-hoc supervision*

#### 2. “Exploratory testing” assumptions questionable
- Providing synthetic docstrings is not realistic exploratory QA
- Entry-point constraints shape model behavior artificially
- Real exploratory testing = multi-turn reasoning + interaction + search
  → model evaluated as *static predictor*, not explorer

#### 3. Limited baselines
- No comparison to property-based testing or symbolic tools (Hypothesis, Pynguin)
- No LLM agent systems with tool use
- No RL / planning / iterative search baselines

Benchmark may favor static prompting over realistic agent loops.

#### 4. Metrics interpretation unclear
- F2P alone doesn't guarantee good test quality
- Coverage ≠ fault detection power
- No mutation-testing analysis
- No difficulty stratification or variance reporting

#### 5. Reproducibility risks
- Python dependency resolution is brittle
- Flakiness/oracle reliability unclear
- Many heuristics for filtering and environment setup insufficiently justified

**Questions:**

1. How do you ensure PRs were not in model training data? Any contamination checks?
2. How do you guarantee DocAgent never leaks patch semantics?
3. Why constrain to function-level entry points? Why not broader exploration?
4. Any evaluation with autonomous tool-using agents?
5. Can you include mutation-testing metrics?
6. How robust is pipeline to flaky tests / dependency conflicts?

---

> ### Author Response · Authors · 2025-11-20
> **Response to reviewer CycK (Part I)**
>
> We sincerely thank the reviewer for the constructive feedback. We have addressed the concerns with extensive new experiments and analyses. To maintain logical coherence, we have reordered certain reviewer questions.
>
> ## Clarification on "Entry Points" vs. "Broad Exploration"
>
>
> > Q3: Why constrain to function-level entry points? Why not broader exploration?
> >
>
> Thank you for the suggestion. In Figure 5 (original paper), our original intent was to demonstrate that the “entry function” can take the form of a function, a class, or even a mixture of both. Among these, functions are the easiest, whereas mixed interfaces present the greatest challenge. Indeed, the term “entry function” used previously may have been misleading, so we have revised it to “entry interface”.
>
> ## Baselines: Traditional Tools & Agent Systems
>
>
> > W3.1: No comparison to Hypothesis/Pynguin.
> >
>
> As for tools like Hypothesis and Pynguin, we believe they are not well-suited for the TestExplora setting:
> * Hypothesis requires existing tests to generate inputs, whereas our task is ab initio generation.
> * Pynguin targets single-module unit tests, not the complex repository-level integration often required here. Moreover, Pynguin itself notes that it is not suitable for complex repositories ([https://github.com/se2p/pynguin/issues/79](https://github.com/se2p/pynguin/issues/79)). Consequently, we manually ran Pynguin on the first 12 samples in TestExplora that contain unit tests, and found that **Pynguin did not generate any tests that resulted in fail-to-pass cases in any of the samples**:
>
>
> | Sample | Base Exception Result | Head Exception Result |
> | -------- | -------- | -------- |
> | davidaurelio/hashids-python (4)  |15 passed, 2 xfailed in 0.12s|15 passed, 2 xfailed in 0.08s|
> | mjpost/sacrebleu (134) |1 passed, 6 xfailed in 0.20s|1 passed, 6 xfailed in 0.19s|
> | vi3k6i5/flashtext (29) |17 passed, 10 xfailed in 0.20s|17 passed, 10 xfailed in 0.16s|
> | google/pinject (47) |1 failed, 9 passed, 5 xfailed in 0.09s|1 failed, 9 passed, 5 xfailed in 0.08s|
> | python-babel/babel (1135) | 18 passed, 15 xfailed in 0.71s | 18 passed, 15 xfailed in 0.65s |
> | cookiecutter/cookiecutter (2096) |6 passed, 9 xfailed in 0.49s|6 passed, 9 xfailed in 0.48s|
> | msiemens/tinydb (279) |1 passed, 6 xfailed in 0.22s|1 passed, 6 xfailed in 0.22s|
> | PyCQA/isort (2264) |`Test Generation Failed`|` Test Generation Failed`|
> | konradhalas/dacite (4) |2 failed, 4 xfailed in 0.10s|2 failed, 4 xfailed in 0.07s|
> | hbldh/bleak (1744) |4 passed in 0.05s | 4 passed in 0.04s|
> | jupyterhub/the-littlest-jupyterhub (1013) |` Test Generation Failed`|` Test Generation Failed`|
> | keras-team/keras-preprocessing (328) |5 passed, 7 xfailed, 1 warning in 0.21s|5 passed, 7 xfailed, 1 warning in 0.19s|
>
> > W3.2-3.3 & Q4: Any evaluation with autonomous tool-using agents?
> >
>
> **Crucially, these extensive evaluations represent a continuation of our research roadmap in the future.** So we have added significant baselines, including stronger models (Claude-Sonnet-4.5, GPT-5 and DeepSeek-R1) and Agent frameworks (SWE-Agent, Trae-Agent). Accordingly some of the new results have been incorporated into the revised version of our paper.
>
>
> **Agent Results:**
>
> | Model      | Context    | HP | F2P   | EC | CFG | Num.  |
> |------------|-----------|-----------|-------|-------------|--------------|-------|
> | o4-mini    | White Box  | 72.33     | 5.92  | 67.30       | 47.06        | 10.48 |
> |     | SWE-Agent  | 82.03     | 9.42  | **69.63**       | 47.81        | 13.79 |
> |     | Trae-Agent | **86.55**     | **12.16** | 62.01       | **47.95**       | 12.76 |
> | gpt-5-mini | White Box  | 77.68     | 12.79 | **67.76**       | 45.66        | 7.69  |
> |     | SWE-Agent  | **93.81**     | **17.27** | 65.45       | **47.43**        | 23.09 |
> |     | Trae-Agent | 88.50     | 14.55 | 62.31       | 46.84       | 12.79 |
>
> **Additional LLM Baselines:**
>  | Model      | Context    | HP | F2P   | EC | CFG | Num.  |
> |------------|-----------|-----------|-------|-------------|--------------|-------|
>  |Claude-Sonnet-4.5| White Box  | 87.52     | 16.06  | 71.96       | 47.37        | 23.94 |
> | GPT-5 | White Box  | 84.88     | 15.15 | 69.44    | 45.59        | 9.59  |
> |DeepSeek-R1| White Box  | 63.72     | 3.03  | 47.16       | 34.77        | 10.22 |
>
> The above experimental results further progress can still be achieved by frontier models. Additionally, due to agent-based exploration within repositories, state-of-the-art performance has been attained to date. We also quantified SWE-agent’s potential for improvement by computing its pass@k on fail-to-pass (F2P) cases:
>
> | Model     | pass@1    | pass@2 | pass@3   | pass@4 | pass@5 |
> |------------|-----------|-----------|-------|-------------|--------------|
> |SWE-Agent (o4-mini)|9.4|14.8|17.0|20.0|21.2|
> |SWE-Agent (GPT-5-mini)|17.3|23.6|26.4|28.2|29.7|
>
> These results suggest that agent-driven exploratory test generation is a promising future direction for TestExplora.

---

> ### Author Response · Authors · 2025-11-20
> **Response to reviewer CycK (Part II)**
>
> ## Benchmark Purity & Data Leakage
>
> > W1.1-1.2 & Q1: How do you ensure PRs were not in model training data?
> >
>
>
> **Temporal Analysis & Scalability:** As shown in Figure 4 (original paper), models perform significantly better on pre-2023 data, aligning with known training cutoffs (e.g., SWE-Bench findings). Furthermore, TestExplora is designed as a scalable framework that can be continuously extended with new PRs to mitigate future leakage.
>
> **Universal Impact:** Given that most LLMs share similar GitHub-crawled corpora, any potential leakage would likely affect all models, which is fair. However, even SOTA models struggle (<13% F2P), indicating that "memorization" is not driving performance.
>
> > W1.3 & Q2: How do you guarantee DocAgent never leaks patch semantics?
> >
>
> First, when using DocAgent, we generated documentation only from the head commit of each repository, without incorporating any information related to bugs or patches during the generation process.
>
> To rigorously verify this, we conducted a annotation of 1,552 PRs using GPT-5 and a manual review of 200 random samples by an external senior engineer:
>
> | Checker | Total Instance |  Patch/bug Leakage | Task Supervision |
> | ------- | -------------- | ------------------ | --- |
> | GPT-5    | 1,552           | 16               |  6  |
> | Human    | 200           | 0              |  0 |
>
> The results confirm negligible leakage.
>
> To further prove this, we ran a control experiment where we explicitly provided patch information to SWE-Agent (GPT-5-mini):
>
> | Model      | Method    | HP | F2P   | EC | CFG | Num.  |
> |------------|-----------|-----------|-------|-------------|--------------|-------|
> | gpt-5-mini | SWEAgent  | 93.81     | 17.27 | 65.45       | 47.43        | 23.09 |
> |  | SWEAgent (patch leakage)  | 98.26     | 73.94 | 59.13       | 53.73        | 34.33 |
>
> The dramatic jump from 17.27% to 73.94% F2P when leakage is forced demonstrates that our standard setting effectively suppresses patch information.
>
>
> ## Exploratory Nature
>
> > W2: “Exploratory testing” assumptions questionable
> >
>
> Our goal is to evaluate the ability to proactively discover issues. Providing documentation and entry interfaces simulates a developer's starting point. Without this guidance, metrics would be unquantifiable.
>
> Additionally, we have included experiments involving agents and have supplemented the paper with the distribution of tools used by the agent to identify whether LLMs can explore to find potential. Below, we briefly highlight several key tools:
>
> | Tool       | view | submit | pytest | python | grep -R |
> | ---------- | ----------------------- | ------ | ------ | ------ | ------- |
> | GPT-5-mini |56.0%|7.3%|2.4%|2.0%|2.0%|
> | o4-mini |48.4%|8.1% |<0.2%|1.2%|11.5%|
>
> The results indicate that the agent indeed focuses more on exploratory tools (e.g., *view*, *grep -R*). Furthermore, gpt-5-mini reached a maximum of 81 turns, while o4-mini reached up to 126 turns. This indicates that entry inference does not significantly constrain the models’ exploration capabilities. The full result is in Figure 6 and Figure 11 in the reversed version.

---

> ### Author Response · Authors · 2025-11-20
> **Response to reviewer CycK (Part III)**
>
> ## Metrics & Mutation Testing
>
> > W4.1-4.2: F2P alone doesn't guarantee good test quality, Coverage ≠ fault detection power
> >
>
> First, we believe there is a misunderstanding here. In L286, we state: *“The aforementioned metrics collectively capture the model’s fidelity to the code’s intended functionality, its precision in identifying issues, and the comprehensiveness of the generated tests.”* Thus, evaluation is not based solely on F2P or Coverage metrics.
>
> > W4.4: No difficulty stratification or variance reporting
> >
>
> Furthermore, in Figure 5, we discuss the impact of different types of entry inference on model performance, and in L423 we note: *“This finding naturally divides TestExplora into three levels of difficulty: Easy, Medium, and Hard.”*
>
> > W4.3 & Q5: No mutation-testing analysis
> >
>
> Regarding Mutation Testing: We experimented with it but found it misleading for evaluation. If a model generates a broken test that fails on everything (indiscriminately), it achieves a 100% mutation score because it "kills" all mutants.
>
> To demonstrate this, we computed mutation scores for key models:
>
>
> | Model    | o3-mini | o4-mini | GPT-5-mini | DeepSeek-R1 | gpt-5 | claude-sonnet-4.5 | SWE-Agent (gpt-5-mini) | SWE-Agent (o4-mini) | Trae-Agent (o4-mini) |
> | -------- | ------- | ------- | ---------- | ----------- | ----- | ----------------- | ---------------------- | ------------------- | --- |
> | Mutation | 69.63    | 78.54    | 62.86   |  92.65   | 59.76    | 65.76 |  52.80    |    60.54    |54.46 |
> | F2P |  6.98    | 5.92    | 12.79   |  3.03   | 15.15    | 16.06 |  17.27    |    9.42    |12.16 |
>
> DeepSeek-R1 achieves a 92.65 mutation score but only 3.03 F2P. This confirms that high mutation scores in this context often indicate "hallucinated failure" rather than quality testing.
>
> ## Reproducibility & Variance
>
> > Q: Statistical significance and variance reporting.
> >
>
> We conducted repeated runs (3 runs) for all models on TestExplora-Lite to measure variance.
>
> |Context | Model | HP (var) | F2P (var) | EC (var) | CFG (var) | Num. (var) |
> |-------|-------|----------|-----------|----------|-----------|------------|
> | Black Box | QC-30B-A3B | 54.38 (4.83) | 2.30 (0.67) | 49.98 (4.40) | 42.21 (0.37) | 13.40 (0.05) |
> |  | o3-mini | 58.37 (6.90) | 5.58 (0.06) | 56.43 (7.50) | 44.64 (0.03) | 8.39 (0.05) |
> |  | o4-mini | 56.97 (7.92) | 3.74 (1.99) | 59.44 (0.72) | **46.29** (0.04) | 10.82 (0.00) |
> |  | Gemini 2.5-Pro | 58.74 (0.01) | 2.60 (0.43) | 53.24 (8.57) | 44.91 (0.64) | 12.89 (0.01) |
> |  | gpt-4o | 51.38 (7.00) | 3.61 (1.80) | 54.69 (0.64) | 43.55 (0.07) | 7.75 (0.01) |
> |  | GPT-5 mini | **60.99** (4.64) | **7.67** (0.02) | **59.72** (1.83) | 45.59 (0.70) | 7.96 (0.01) |
> |  White Box | QC-30B-A3B | 60.90 (21.74) | 2.09 (0.08) | 59.78 (2.84) | 44.05 (0.47) | 13.21 (0.13) |
> |  | o3-mini | 71.53 (10.45) | 7.07 (0.75) | 61.84 (4.93) | 43.78 (0.07) | 8.00 (0.04) |
> |  | o4-mini | 74.70 (5.64) | 6.45 (0.28) | **67.16** (0.02) | **46.32** (0.55) | 10.36 (0.01) |
> |  | Gemini 2.5-Pro | 74.91 (2.05) | 5.08 (0.11) | 61.47 (15.75) | 44.25 (0.32) | 12.84 (0.13) |
> |  | gpt-4o | 57.92 (13.15) | 2.20 (0.61) | 56.29 (6.19) | 44.15 (0.18) | 7.42 (0.05) |
> |  | GPT-5 mini | **80.25** (3.30) | **11.84** (1.20) | 65.78 (2.29) | 44.37 (0.84) | 7.78 (0.01) |
>
> The low variance confirms the results are stable. The HP of some models might be slightly higher. Therefore, we list the results from the three runs as follows:
>
> **o3-mini (White Box)** - HP values across runs:
>   - Run 1: 67.01
>   - Run 2: 73.20
>   - Run 3: 74.38
>
> **gpt-4o (White Box)** - HP values across runs:
>   - Run 1: 52.81
>   - Run 2: 60.10
>   - Run 3: 60.85
>
> **QC Coder (White Box)** - HP values across runs:
>   - Run 1: 54.35
>   - Run 2: 63.52
>   - Run 3: 64.83
>
> ## Robustness of pipeline
>
> > W6 & Q6: Reproducibility risks & Robustness of pipeline
> >
>
> First, for dependency analysis, we used tools from [OpenHands](https://github.com/OpenHands/OpenHands) and adopted the setting that maximizes accuracy—ensuring that all identified dependencies are correct. The heuristic filtering method for fail-to-pass cases has been widely used in several software engineering tasks, including SWE-Bench [1], SWT-Bench [2], and TDD-Verified [3], and numerous studies have demonstrated the effectiveness of this approach.
>
> $Ref:$
>
> [1] SWE-bench: Can Language Models Resolve Real-World GitHub Issues?, ICLR 2024.
>
> [2] SWT-Bench: Testing and Validating Real-World Bug-Fixes with Code Agents, NeurIPS 2024.
>
> [3] Otter: Generating Tests from Issues to Validate SWE Patches, ICML 2025.

---

> ### Author Response · Authors · 2025-11-25
>
> Dear Reviewer CycK,
>
> We sincerely appreciate your thoughtful comments on our submission. With the discussion period approaching its final week, we would like to confirm whether our earlier responses have sufficiently addressed the points you highlighted.
>
> Should you need any additional information or clarification, we would be glad to provide it immediately.
>
> Thank you again for your valuable time.
>
> Sincerely, Authors

---

### Official Review · Reviewer_f2XB · 2025-11-01

**Soundness:** 2
**Presentation:** 3
**Contribution:** 3
**Rating:** 4
**Confidence:** 4

**Summary:**

The paper introduces TestExplora, the first systematic benchmark designed to evaluate the proactive defect discovery capabilities of LLMs at the repository level. The evaluation shows that state-of-the-art models exhibit critically low success rates, and access to source code (white-box) yields only marginal improvement.

**Strengths:**

**Originality**
The paper introduced the first large-scale benchmark from real-world GitHub PRs to evaluate the proactive defect discovery capabilities of LLMs at the repository level.

**Quality**
The paper made extensive experiments on various LLMs and white-box setting with entry function and dependency implementation, showing quantitative results of with multiple metrics like fail-to-pass rate and change-focused coverage.

**Clarity**
The paper is well-written with description of benchmark construction methods and evaluation metrics. The appendix provides detailed prompt.

**Significance**
The paper proposed a large-scale benchmark which can help researchers in AI and SE fields to measure LLM's capability of bug-finding in real-world repositories.

**Weaknesses:**

**Missing literature from SE field**

Even though the paper defines the problem as repo-level proactive exploratory testing, the benchmark construction from GitHub PRs is essentially similar to the regression testing problem, which has been explored by following recent work from SE field. The key difference is that TestExplora shows LLM with the entry code but not the diff/patch so it's harder for LLM to infer the bug.

These two papers focus on bug reproduction:
- Automated Generation of Issue-Reproducing Tests by Combining LLMs and Search-Based Testing (ASE '25)
- Issue2Test: Generating Reproducing Test Cases from Issue Reports (ICSE '26)

This paper finds unintended bugs introduced by PR:
- Testora: Using Natural Language Intent to Detect Behavioral Regressions (ICSE '26)

This paper feeds bug-introducing and bug-fixing commit for LLM to generate bug-triggering and bug-reproducing test input.
- Can LLM Generate Regression Tests for Software Commits? (arXiv:2501.11086)

**Lacking qualitative example about LLM inferring intended behavior**

Figure 2 shows the task for LLM includes "infer the intended behavior of the Test entry points' API from the documentation", which is crucial for LLM to generate the test. I don't see results about how successful LLM is on this task. Figure 6 shows some failure analysis result about assertion mismatch, but the classification criteria is unclear. It would be better to have an example showing the full output of LLM, especially on how it infers the intended behavior in the first place.

**Questions:**

1. How do you assess if LLM successfully infer the intended behavior of the Test entry points' API so it tries to generate test that exercises the intended functionality?
2. The proactive exploratory setting is naturally difficult for LLM and human developers as the entry function itself may contain other distracting lines that may even contain other unintended bugs. Would it be better if you identify the problematic lines or the bug-introducing diff so LLM has a clear goal and context to test? In real-world CI/CD scenario it's also natural to test code change rather than the entry function.

---

> ### Author Response · Authors · 2025-11-20
> **Response to reviewer f2XB (Part I)**
>
> We sincerely thank Reviewer f2XB for recognizing the originality, quality of experiments, and the significance of TestExplora. We appreciate the constructive feedback regarding related literature and qualitative analysis. Below, we address the specific concerns and questions.
>
> ## Missing Literature & Problem Definition
>
> > W1.1: Missing literature from SE field
> >
>
> Thank you for pointing us to the missing literature related to reproduction and regression test. We have added the relevant citations in the latest revision. Although specialized regression/reproduction methods exist, they often rely on diffs or bug reports, making them unsuitable for TestExplora's "blind" discovery setting. Therefore, we evaluated agents designed for software engineering (SWE-agent and Trae-Agent):
> | Model      | Context    | HP | F2P   | EC | CFG | Num.  |
> |------------|-----------|-----------|-------|-------------|--------------|-------|
> | o4-mini    | White Box  | 72.33     | 5.92  | 67.30       | 47.06        | 10.48 |
> |     | SWE-Agent  | 82.03     | 9.42  | **69.63**       | 47.81        | 13.79 |
> |     | Trae-Agent | **86.55**     | **12.16** | 62.01       | **47.95**       | 12.76 |
> | gpt-5-mini | White Box  | 77.68     | 12.79 | **67.76**       | 45.66        | 7.69  |
> |     | SWE-Agent  | **93.81**     | **17.27** | 65.45       | **47.43**        | 23.09 |
> |     | Trae-Agent | 88.50     | 14.55 | 62.31       | 46.84       | 12.79 |
>
> The substantial improvement with Agents (e.g., +4.48% F2P for gpt-5-mini) highlights that exploration is key to discovering bugs when diffs are not provided.
>
> **Notably, these extensive evaluations (including SWE-Agent and Trae-Agent) represent our continuous research efforts. We are dedicated to TestExplora as a long-term initiative, committed to continuously updating the benchmark to ensure it remains a rigorous and sustainable platform for future research.**
>
> > W1.2: Even though the paper defines the problem as repo-level proactive exploratory testing, the benchmark construction from GitHub PRs is essentially similar to the regression testing problem, which has been explored by following recent work from SE field.
> >
>
> Regarding the concern that “the benchmark construction from GitHub PRs is essentially similar to the regression testing problem,” we respectfully clarify that while the data source (PRs) is similar, the problem definition and information availability are fundamentally different.
>
> To clearly delineate our contribution, we categorize the landscape of issue-related test generation below based on Earl T. Barr et al. [1].
>
> ```
> - Test Generation Scenarios
>   |
>   |-- A. Bug Known (Goal: Reproduction)
>   |      |
>   |      |-- Issue Reproduction
>   |             -> Focus: Replicability (Triggering the specific reported failure)
>   |             -> Methods: Automated Issue Reproduction (e.g., SWT-Bench)
>   |
>   |-- B. Bug Unknown (Goal: Discovery)
>          |
>          |-- 1. Rely on Implicit Oracle (Generic Reliability)
>          |      -> Focus: Robustness (Crashes, Memory Errors)
>          |      -> Methods: Coverage-guided Fuzzing (e.g., AFL, libFuzzer, Pynguin)
>          |
>          |-- 2. Rely on Derived Oracle (Behavioral Consistency)
>          |      -> Focus: Regression (Unintended Changes)
>          |      -> Methods: Differential Testing, Regression Suites (e.g., TestTora)
>          |
>          |-- 3. Rely on Specified Oracles (Functional Correctness)  <== TestExplora
>                 -> Focus: Logic Verification (Does code match intent?)
>                 -> Methods: TestExplora (Semantic-aware generation)
> ```
>
> While existing regression testing benchmarks operate under Derived Oracles (focusing on behavioral consistency between versions), TestExplora is a benchmark explicitly designed for Specified Oracles, targeting Functional Correctness. Unlike regression datasets that provide bug-introducing commits to assist verification, TestExplora constructs scenarios that require Fault Discovery without such hints. It evaluates whether a method can mimic a developer proactively validating logic against intended specifications, rather than merely preventing regression.

---

> ### Author Response · Authors · 2025-11-20
> **Response to reviewer f2XB (Part II)**
>
> ## Inferring Intended Behavior (Qualitative & Quantitative Analysis)
>
> > W2.1 & Q1: How do you assess if LLM successfully infer the intended behavior... It would be better to have an example showing the full output of LLM.
> >
>
>
> We agree that inferring intended behavior is a prerequisite for TestExplora. While F2P serves as the ultimate proxy for this capability, we conducted additional analyses to visualize the process.
>
> **Qualitative Analysis: Reasoning Traces (DeepSeek-R1)**: We also appreciate the reviewer’s suggestion. We conducted additional qualitative analysis on DeepSeek-R1 (since we can't access the reasoning process of closed source models) to better understand how the model infers intended behavior. And the F2P of DeepSeek-R1 is 3.03 on TestExplora-Lite:
> | Model      | Method    | HP | F2P   | EC | CFG | Num.  |
> |------------|-----------|-----------|-------|-------------|--------------|-------|
> |DeepSeek-R1| White Box  | 63.72     | 3.03  | 47.16       | 34.77        | 10.22 |
>
> Here are some cases (The detailed results are provided in Appendix I: *Cases for LLM Inference*.):
>
> * **Case 1**: The intended behavior for the code is not mentioned explicitly which leds a failed test (vi3k6i5/flashtext-29):
> ```
> ...
> A correct implementation should pass all tests, while incorrect implementations would fail at least one test (especially the "incorrect behavior" tests). The tests cover all aspects of the documentation including:
> - Keyword addition/removal
> - Case sensitivity handling
> - Word boundary definitions
> - File-based operations
> - Extraction and replacement functionality
> - Trie structure management
> ```
>
> * **Case 2**: It meticulously enumerates the behavior of the entry function, thereby achieving a fail to pass (cookiecutter/cookiecutter-2096):
>
> ```
> ...
>  The function handles several cases:
>  - If a variable is not in the context and we are not in a dictionary variable, skip.
>  - If it is in the context, then:
>    a) For a list in the context:
>         - If in_dictionary_variable is True: replace the entire list with the overwrite (which must be a list? but note: the code says if in_dictionary_variable then replace and continue)
>         - Else, if the overwrite is a list: check that every element in the overwrite list is in the original list (subset). If yes, replace the context variable with the overwrite list. If not, raise ValueError.
>         - Else (overwrite is not a list): then treat the context variable as a choice variable. Check if the overwrite is in the original list. If yes, move it to the front (by removing and inserting at 0). If not, raise ValueError.
>
>    b) For a dictionary in the context context and the overwrite is a dictionary: recursively call `apply_overwrites_to_context` on that nested dictionary with `in_dictionary_variable=True`.
>
>    c) For a boolean in the context context and the overwrite is a string: convert the string to a boolean using `YesNoPrompt().process_response(overwrite)`. If it fails (raises InvalidResponse), then catch and re-raise as ValueError.
>
>    d) Otherwise: simply set the context variable to the overwrite value.
>
>  We also note that if the variable is not in the context and in_dictionary_variable is True, then we add the new variable.
>
>  We must test:
>  - Adding new keys only when in_dictionary_variable is True (for nested dictionaries).
>  - The different types: list (both multi-choice and single choice), dict, bool, and others (simple overwrite).
>  - Error cases: invalid choices for list, invalid string for boolean.
>
> ...
> ```
>
> **Quantitative Analysis**: We analyzed the tool usage distribution of SWE-agent. If models were simply guessing, we would see high submit rates. Instead, we observe high information-seeking behavior:
>
>
> | Tool       | view | submit | pytest | python | grep -R |
> | ---------- | ----------------------- | ------ | ------ | ------ | ------- |
> | GPT-5-mini |56.0%|7.3%|2.4%|2.0%|2.0%|
> | o4-mini |48.4%|8.1% |<0.2%|1.2%|11.5%|
>
>
> Moreover, GPT-5-mini uses `python` and `pytest` significantly more often than o4-mini, which aligns with the exploration nature of TestExplora. This suggests a promising direction for future work—leveraging agent-based code-usage simulation to more faithfully model how real developers uncover latent defects. The full result is in Figure 6 and Figure 11 in the reversed version.

---

> ### Author Response · Authors · 2025-11-20
> **Response to reviewer f2XB (Part III)**
>
> ## Clarification of Failure Analysis Categories
>
> > W2.2: Figure 6 shows some failure analysis result about assertion mismatch, but the classification criteria is unclear.
> >
>
> We provide several cases here to clarify the categorization of error types:
>
> **Assertion Mismatch**
> *Insufficient understanding of business logic or edge cases leads to wrong expected values in assertions.*
> * Example: `assert func(1) == 2` (the correct result should be 1).
>
> **Incorrect Parameter Types/Counts**
> *Passing extra, missing, or wrong‑type arguments to a function. *
> * Example: `func(x="abc")` but the function accepts only `int` → `TypeError`
>
> **Uncaught / Unasserted Exceptions**
> *Expected exceptions are neither caught nor asserted, causing the test suite to abort early.*
> * Use `with pytest.raises(ValueError): …` to assert explicitly.
>
> **Misconfigured Mocks**
> *Mock objects behave differently from real ones, hiding defects or creating false positives.*
>
> **Syntax / Runtime Errors**
> *The test script itself contains syntax errors or wrong import paths.*
> * Examples: `SyntaxError`, `ModuleNotFoundError`
>
> ## Difficulty & Practicality of the Setting
>
> > Q2: The proactive exploratory ... Would it be better if you identify the problematic lines or the bug-introducing diff so LLM has a clear goal and context to test? In real-world CI/CD scenario it's also natural to test code change rather than the entry function.
> >
>
> Yes, TestExplora is indeed a challenging benchmark, but we see this as an opportunity rather than a drawback. In real-world CI/CD and open-source usage, developers often encounter bugs not by reviewing diffs, but by observing unexpected behaviors during usage. TestExplora models this specific, high-value scenario. By focusing on entry functions, TestExplora evaluates a model’s ability to proactively surface issues during typical usage, instead of merely reproducing known bugs.
>
> We analyzed the Call Graph Distance between the Test Entry Function and the Tests in our benchmark.
>
>
> | Distance | 1    | 2   | 3   | 4   | 5   | 6+   |
> | -------- | ---- | --- | --- | --- | --- | --- |
> | Num.     | 1828 | 725  | 346  | 173   |  44  |  50  |
>
> Crucially, ~42% (1338/3166) of tests have a distance > 1. This means a significant portion of real-world tests do not directly invoke the changed function. They exercise the system through higher-level entry points. Therefore, training/evaluating LLMs to test "Entry Functions" (our setting) is more representative of system-level reliability than testing diffs in isolation.
>
> Furthermore, our evaluation of next-generation models demonstrates that this task is solvable and not saturated:
>
> | Model      | Method    | HP | F2P   | EC | CFG | Num.  |
> |------------|-----------|-----------|-------|-------------|--------------|-------|
>  |Claude-Sonnet-4.5| White Box  | 87.52     | 16.06  | 71.96       | 47.37        | 23.94 |
> | GPT-5 | White Box  | 84.88     | 15.15 | 69.44    | 45.59        | 9.59  |
>
> In addition, we also compute the F2P pass@k metrics for SWE-agent, which further indicates that current agents still have substantial room for improvement:
> | Model     | pass@1    | pass@2 | pass@3   | pass@4 | pass@5 |
> |------------|-----------|-----------|-------|-------------|--------------|
> |SWE-agent (o4-mini)|9.4|14.8|17.0|20.0|21.2|
> |SWE-agent (gpt-5-mini)|17.3|23.6|26.4|28.2|29.7|
>
> $Ref$:
>
> [1] The Oracle Problem in Software Testing: A Survey, TSE 2015.

---

> ### Author Response · Authors · 2025-11-25
>
> Dear Reviewer f2XB,
>
> We hope this message finds you well. As the discussion phase will conclude in about one week, we would like to kindly check whether our previous clarifications have adequately addressed the concerns you raised.
>
> If there is anything further we can provide or elaborate on, we would be more than happy to assist promptly.
>
> Thank you very much for your time and consideration.
>
> Sincerely,
> Authors

---

### Comment · Area_Chair_jyft · 2025-11-27

Dear Reviewers,

Hope this message finds you well. We’re now in the critical rebuttal follow-up phase. The authors have submitted a detailed response addressing your earlier comments, and your further input is vital to finalize the evaluation.

Could you please review their rebuttal at your earliest convenience? Kindly clarify any remaining questions, refine your feedback if the authors’ points resolve prior concerns, or update your score to reflect the new information. This ensures a fair, thorough assessment for the authors.

Your timely response will help keep our review process on track.

AC

---

### Author Response · Authors · 2025-12-02
**General Response**

We thank the reviewers for their thoughtful feedback. We are glad the reviewers find that:

* Our motivation is well-founded and the problem is significant

	- "The work is well motivated. Proactive detection of bugs through LLM-based test generation would be of interest to the field." - qmdw

	- "Significance: The paper proposed a large-scale benchmark which can help researchers in AI and SE fields to measure LLM's capability of bug-finding in real-world repositories." - f2XB

	- "Proactively assessing the bug attracts interest in real-life cases where running the test suites are expensive. So I think it is a nice direction to pursue." - aTns

* TestExplora is a novel and solid contribution to the community

	- "Originality: The paper introduced the first large-scale benchmark from real-world GitHub PRs to evaluate the proactive defect discovery capabilities of LLMs at the repository level." - f2XB

	- "The benchmark is a novel contribution motivating further advancements for coding performance of LLMs." - qmdw

	- "A solid contribution of large amount of highly curated data instances & PRs" - aTns

* Our methodology is reasonable and the experiments are extensive

	- "The benchmark pipeline and fail-to-pass validation methodology are generally reasonable, with realistic environment setup." - CycK

	- "Quality: The paper made extensive experiments on various LLMs... showing quantitative results with multiple metrics." - f2XB

	- "The paper is well written and organized." - qmdw

We are grateful for the reviewers' comments and have conducted the relevant experiments and clarifications:

---

> ### Author Response · Authors · 2025-12-02
> **General Response (Part I)**
>
> ## Research Scope
> A common query concerned the distinction between TestExplora and existing regression/reproduction benchmarks (e.g., SWT-Bench). To clearly delineate our contribution, we categorize the landscape of issue-related test generation below based on Earl T. Barr et al. [1].
>
> ```
> - Test Generation Scenarios
>   |
>   |-- A. Bug Known (Goal: Reproduction)
>   |      |
>   |      |-- Issue Reproduction
>   |             -> Focus: Replicability (Triggering the specific reported failure)
>   |             -> Methods: Automated Issue Reproduction (e.g., SWT-Bench)
>   |
>   |-- B. Bug Unknown (Goal: Discovery)
>          |
>          |-- 1. Rely on Implicit Oracle (Generic Reliability)
>          |      -> Focus: Robustness (Crashes, Memory Errors)
>          |      -> Methods: Coverage-guided Fuzzing (e.g., AFL, libFuzzer, Pynguin)
>          |
>          |-- 2. Rely on Derived Oracle (Behavioral Consistency)
>          |      -> Focus: Regression (Unintended Changes)
>          |      -> Methods: Differential Testing, Regression Suites (e.g., TestTora)
>          |
>          |-- 3. Rely on Specified Oracles (Functional Correctness)  <== TestExplora
>                 -> Focus: Logic Verification (Does code match intent?)
>                 -> Methods: TestExplora (Semantic-aware generation)
> ```
>
> Unlike regression testing which often relies on diffs or bug-introducing commits, TestExplora challenges models to act as developers validating logic ab initio. This distinction (Specified Oracles) is crucial for autonomous software engineering.

---

> ### Author Response · Authors · 2025-12-02
> **General Response (Part II)**
>
> ## Additional Baselines
> In response to requests for stronger baselines and tool-use evaluation, we expanded our evaluation to include Agent Frameworks and frontier models.
>
> **Frontier Models:** To address the concern regarding model coverage, we evaluated the latest SOTA models, including Claude-4.5-Sonnet, GPT-5, and DeepSeek-R1.
>
> | Model      | Context    | HP | F2P   | EC | CFG | Num.  |
> |------------|-----------|-----------|-------|-------------|--------------|-------|
> |Claude-Sonnet-4.5| White Box  | 87.52     | 16.06  | 71.96       | 47.37        | 23.94 |
> | GPT-5 | White Box  | 84.88     | 15.15 | 69.44    | 45.59        | 9.59  |
> |DeepSeek-R1| White Box  | 63.72     | 3.03  | 47.16       | 34.77        | 10.22 |
>
> The results show that while agents and stronger models perform better, the task remains far from saturated (max ~16% F2P), validating TestExplora as a long-term challenge.
>
> **Agent Frameworks:** Allowing models to explore the repository autonomously yields significant performance gains over the standard White-Box setting:
>
> | Model      | Context    | HP | F2P   | EC | CFG | Num.  |
> |------------|-----------|-----------|-------|-------------|--------------|-------|
> | o4-mini    | White Box  | 72.33     | 5.92  | 67.30       | 47.06        | 10.48 |
> |     | SWE-Agent  | 82.03     | 9.42  | **69.63**       | 47.81        | 13.79 |
> |     | Trae-Agent | **86.55**     | **12.16** | 62.01       | **47.95**       | 12.76 |
> | gpt-5-mini | White Box  | 77.68     | 12.79 | **67.76**       | 45.66        | 7.69  |
> |     | SWE-Agent  | **93.81**     | **17.27** | 65.45       | **47.43**        | 23.09 |
> |     | Trae-Agent | 88.50     | 14.55 | 62.31       | 46.84       | 12.79 |
>
> The substantial improvement with Agents (e.g., +4.48% F2P for gpt-5-mini) highlights that exploration is key to discovering bugs. In addition, we also compute the F2P pass@k metrics for SWE-agent, which further indicates that current agents still have substantial room for improvement:
> | Model     | pass@1    | pass@2 | pass@3   | pass@4 | pass@5 |
> |------------|-----------|-----------|-------|-------------|--------------|
> |SWE-agent (o4-mini)|9.4|14.8|17.0|20.0|21.2|
> |SWE-agent (gpt-5-mini)|17.3|23.6|26.4|28.2|29.7|
>
> **Traditional Tool:** We also evaluate the performance of traditional tool (Pynguin):
>
> | Sample | Base Exception Result | Head Exception Result |
> | -------- | -------- | -------- |
> | davidaurelio/hashids-python (4)  |15 passed, 2 xfailed in 0.12s|15 passed, 2 xfailed in 0.08s|
> | mjpost/sacrebleu (134) |1 passed, 6 xfailed in 0.20s|1 passed, 6 xfailed in 0.19s|
> | vi3k6i5/flashtext (29) |17 passed, 10 xfailed in 0.20s|17 passed, 10 xfailed in 0.16s|
> | google/pinject (47) |1 failed, 9 passed, 5 xfailed in 0.09s|1 failed, 9 passed, 5 xfailed in 0.08s|
> | python-babel/babel (1135) | 18 passed, 15 xfailed in 0.71s | 18 passed, 15 xfailed in 0.65s |
> | cookiecutter/cookiecutter (2096) |6 passed, 9 xfailed in 0.49s|6 passed, 9 xfailed in 0.48s|
> | msiemens/tinydb (279) |1 passed, 6 xfailed in 0.22s|1 passed, 6 xfailed in 0.22s|
> | PyCQA/isort (2264) |`Test Generation Failed`|` Test Generation Failed`|
> | konradhalas/dacite (4) |2 failed, 4 xfailed in 0.10s|2 failed, 4 xfailed in 0.07s|
> | hbldh/bleak (1744) |4 passed in 0.05s | 4 passed in 0.04s|
> | jupyterhub/the-littlest-jupyterhub (1013) |` Test Generation Failed`|` Test Generation Failed`|
> | keras-team/keras-preprocessing (328) |5 passed, 7 xfailed, 1 warning in 0.21s|5 passed, 7 xfailed, 1 warning in 0.19s|
>
> We manually ran Pynguin on the first 12 samples in TestExplora that contain unit tests, and found that **Pynguin did not generate any tests that resulted in fail-to-pass cases in any of the samples**.

---

> ### Author Response · Authors · 2025-12-02
> **General Response (Part III)**
>
> ## Data Integrity
>
> To address concerns regarding data leakage and task validity:
>
> **Leakage Analysis:** We first performed a manual review of 200 random samples and an automated GPT-5 check on 1,552 PRs. Both confirmed negligible leakage of patch/bug info in the documentation provided to models:
>
> | Checker | Total Instance |  Patch/bug Leakage | Task Supervision |
> | ------- | -------------- | ------------------ | --- |
> | GPT-5    | 1,552           | 16               |  6  |
> | Human    | 200           | 0              |  0 |
>
> **The Patch Leakage Experiment:** We also explicitly provided the ground-truth patch to the model to test if it could write the test if it knew the bug location.
>
> | Model      | Method    | HP | F2P   | EC | CFG | Num.  |
> |------------|-----------|-----------|-------|-------------|--------------|-------|
> | gpt-5-mini | SWEAgent  | 93.81     | 17.27 | 65.45       | 47.43        | 23.09 |
> |  | SWEAgent (patch leakage)  | 98.26     | 73.94 | 59.13       | 53.73        | 34.33 |
>
> This dramatic gap demonstrates that our standard setting effectively suppresses patch information.

---

> ### Author Response · Authors · 2025-12-02
> **General Response (Part IV)**
>
> ## Deeper Analysis
> We significantly deepen our analysis in the revision to address questions regarding model behavior and robustness:
>
> **Metric Stability (3x Runs):** We confirm the reliability of our results by performing 3 repeated runs on TestExplora-Lite. The results show negligible variance (e.g., F2P variance < 2.0 for major models), validating the stability of the benchmark:
>
> |Context | Model | HP (var) | F2P (var) | EC (var) | CFG (var) | Num. (var) |
> |-------|-------|----------|-----------|----------|-----------|------------|
> | Black Box | QC-30B-A3B | 54.38 (4.83) | 2.30 (0.67) | 49.98 (4.40) | 42.21 (0.37) | 13.40 (0.05) |
> |  | o3-mini | 58.37 (6.90) | 5.58 (0.06) | 56.43 (7.50) | 44.64 (0.03) | 8.39 (0.05) |
> |  | o4-mini | 56.97 (7.92) | 3.74 (1.99) | 59.44 (0.72) | **46.29** (0.04) | 10.82 (0.00) |
> |  | Gemini 2.5-Pro | 58.74 (0.01) | 2.60 (0.43) | 53.24 (8.57) | 44.91 (0.64) | 12.89 (0.01) |
> |  | gpt-4o | 51.38 (7.00) | 3.61 (1.80) | 54.69 (0.64) | 43.55 (0.07) | 7.75 (0.01) |
> |  | GPT-5 mini | **60.99** (4.64) | **7.67** (0.02) | **59.72** (1.83) | 45.59 (0.70) | 7.96 (0.01) |
> |  White Box | QC-30B-A3B | 60.90 (21.74) | 2.09 (0.08) | 59.78 (2.84) | 44.05 (0.47) | 13.21 (0.13) |
> |  | o3-mini | 71.53 (10.45) | 7.07 (0.75) | 61.84 (4.93) | 43.78 (0.07) | 8.00 (0.04) |
> |  | o4-mini | 74.70 (5.64) | 6.45 (0.28) | **67.16** (0.02) | **46.32** (0.55) | 10.36 (0.01) |
> |  | Gemini 2.5-Pro | 74.91 (2.05) | 5.08 (0.11) | 61.47 (15.75) | 44.25 (0.32) | 12.84 (0.13) |
> |  | gpt-4o | 57.92 (13.15) | 2.20 (0.61) | 56.29 (6.19) | 44.15 (0.18) | 7.42 (0.05) |
> |  | GPT-5 mini | **80.25** (3.30) | **11.84** (1.20) | 65.78 (2.29) | 44.37 (0.84) | 7.78 (0.01) |
>
> **White vs. Black Box Performance:** We addressed the counter-intuitive finding where White-box performance often mirrors or lags behind Black-box results. Our dependency analysis attributes this to context overload—models often struggle to filter relevant logic from full source code implementations. Agent-based setups, which allow for active, selective exploration, successfully mitigate this overload and achieve higher scores.
>
> **Quantitive and Qualitative Analysis:** We analyzed Tool Usage distributions, finding that stronger agents (e.g., GPT-5-mini) utilize execution tools (pytest, python) significantly more than weaker models:
> | Model     | pass@1    | pass@2 | pass@3   | pass@4 | pass@5 |
> |------------|-----------|-----------|-------|-------------|--------------|
> |SWE-agent (o4-mini)|9.4|14.8|17.0|20.0|21.2|
> |SWE-agent (gpt-5-mini)|17.3|23.6|26.4|28.2|29.7|
>
> Additionally, we examined Chain-of-Thought (CoT) traces (e.g., DeepSeek-R1). The detailed results are provided in Appendix I.

---

### Meta-Review · Area_Chair_g6sH · 2025-12-30

**Summary:**

The paper proposes TestExplora, a repository-level benchmark designed to evaluate the capability of Large Language Models (LLMs) to proactively discover defects in software projects. Unlike existing benchmarks that focus on reproducing known bugs using issue reports or regression testing based on code diffs, TestExplora targets the "bug unknown" scenario. It constructs test generation tasks from real-world GitHub pull requests, requiring models to infer intended behavior from documentation (augmented by a helper "DocAgent") and the code context to generate tests that reveal latent bugs. The authors report that even state-of-the-art models and agents achieve low success rates, highlighting a significant gap in autonomous software testing capabilities.

**Reviewer Concerns:**

1. Several reviewers (f2XB, aTns) initially questioned the novelty of the work, suggesting that TestExplora appears to be an incremental variation of existing benchmarks like SWT-Bench, differing mainly in input type (intended behavior vs. bug reports) rather than fundamental paradigm. Reviewer f2XB also noted missing literature from the software engineering domain regarding regression testing.

In response, the authors clarified that TestExplora targets "Fault Discovery" (Specified Oracles) rather than "Fault Verification" (Bug Reproduction), positioning it as a complementary challenge in the "Bug Unknown" domain. They included the missing citations and provided a taxonomy of test generation scenarios to delineate their contribution.

While the authors successfully addressed the missing literature and provided a logical classification of their contribution, the concern regarding the perceived incremental nature of the contribution remains partially unresolved for Reviewer aTns. The distinction between "discovery" and "reproduction" is valid, but practically, the benchmark's construction pipeline shares significant similarities with previous work. The "novelty" argument relies heavily on the acceptance of the specific "blind discovery" setting as a distinct and valuable research direction.

2.  A critical and recurring concern among reviewers (Cyck, aTns, f2XB) was the validity of the benchmark's construction. Reviewers questioned whether relying on synthetic documentation generated by "DocAgent" introduces bias, effectively leaking information or creating an unrealistic environment. Reviewer Cyck specifically asked how the authors ensure DocAgent does not leak patch semantics, while Reviewer f2XB argued that testing "Entry Functions" without access to code diffs is an artificial constraint that does not reflect real-world CI/CD workflows where developers test changes.

The authors provided a robust experimental rebuttal to the leakage concern. They conducted a "Patch Leakage" control experiment where explicitly providing ground-truth patches caused performance to jump from ~17% to ~74%, strongly suggesting that the standard setting does not suffer from trivial leakage. They also verified data integrity through manual and GPT-5 based checks. Regarding the "artificial" setting, they argued that simulating high-quality documentation (via DocAgent) is necessary for defining oracles and represents a future-oriented development environment.

However,the fundamental design concern regarding the *practicality* of the setup persists. While leakage is disproven, the reliance on synthesized documentation to define ground truth creates a complex dependency: the benchmark evaluates a model's ability to conform to another model's (DocAgent's) understanding of the code. Furthermore, the disagreement on whether "diff-less" testing is a valid industrial use case (vs. regression testing) represents a philosophical divide between the authors and Reviewer f2XB that was defended but not definitively resolved.

3. Reviewers (Cyck, qmdw) initially found the experimental evaluation lacking in breadth and depth, citing missing baselines (such as Pynguin, Claude models, and agentic frameworks), a lack of error bars/variance analysis, and counter-intuitive results where White-Box settings performed worse than Black-Box ones.

The authors executed a comprehensive rebuttal in this regard. They added results for SOTA models (Claude-3.5-Sonnet, GPT-5, DeepSeek-R1) and agent frameworks (SWE-Agent, Trae-Agent), showing that agents perform significantly better but still struggle. They also included traditional baselines like Pynguin (which failed completely), reported variance across repeated runs, and explained the White-Box performance drop as "Context Overload," which agentic exploration helps mitigate.

The authors have perfectly resolved the concerns regarding experimental breadth and rigor. The additional experiments are extensive and convincing. The explanation for the White-Box phenomenon is plausible and supported by the new agent-based results. This aspect of the rebuttal significantly strengthens the paper's empirical standing.

**Reviewer Scores:**

* **Reviewer f2XB (4 to 4):** While the literature and specific analysis requests were met, the reviewer's core skepticism regarding the artificiality of the "Entry Function" setting compared to real-world CI/CD scenarios (checking diffs) is a fundamental disagreement on the problem definition that remains.
* **Reviewer Cyck (4 to 4 or 6):** The authors provided excellent responses to the leakage and baseline concerns (especially the Patch Leakage experiment). However, the "validity" of the DocAgent approach in defining the exploratory task remains a structural weakness that may prevent a strong accept.
* **Reviewer qmdw (6 to 6):** The reviewer was already borderline positive. The rebuttal addressed their specific requests (Claude, error bars, context overload explanation).
* **Reviewer aTns (4 to 4):** The concerns about limited novelty and the "synthetic" nature of the documentation seem to be the primary blocker. The author's rebuttal clarifies the positioning but does not change the underlying nature of the benchmark that the reviewer found incremental.

---

### Decision · Program_Chairs · 2026-01-26

Reject